# LIFT: Language-Interfaced Fine-Tuning for Non-Language Machine Learning Tasks

**Tuan Dinh,**\* **Yuchen Zeng,**\* **Ruisu Zhang, Ziqian Lin, Michael Gira,**
**Shashank Rajput, Jy-yong Sohn, Dimitris Papailiopoulos, Kangwook Lee**

University of Wisconsin-Madison, USA

## Abstract

Fine-tuning pretrained language models (LMs) without making any architectural changes has become a norm for learning various language downstream tasks. However, for *non*-language downstream tasks, a common practice is to employ task-specific designs for input, output layers, and loss functions. For instance, it is possible to fine-tune an LM into an MNIST classifier by replacing the word embedding layer with an image patch embedding layer, the word token output layer with a 10-way output layer, and the word prediction loss with a 10-way classification loss, respectively. A natural question arises: Can LM fine-tuning solve non-language downstream tasks *without* changing the model architecture or loss function? To answer this, we propose **Language-Interfaced Fine-Tuning (LIFT)** and study its efficacy and limitations by conducting an extensive empirical study on a suite of non-language classification and regression tasks. LIFT does not make *any* changes to the model architecture or loss function, and it solely relies on the natural language interface, enabling "no-code machine learning with LMs." We find that LIFT performs comparably well across a wide range of low-dimensional classification and regression tasks, matching the performances of the best baselines in many cases, especially for the classification tasks. We also report experimental results on the fundamental properties of LIFT, including inductive bias, robustness, and sample complexity. We also analyze the effect of pretraining on LIFT and a few properties/techniques specific to LIFT, *e.g.,* context-aware learning via appropriate prompting, calibrated predictions, data generation, and two-stage fine-tuning. Our code is available at `https://github.com/UW-Madison-Lee-Lab/LanguageInterfacedFineTuning`.

## 1 Introduction

Deep neural networks have been highly successful across a multitude of domains, from computer vision [1, 2] and natural language processing [3, 4], to game playing [5, 6]. Most advances in deep learning have come with a variety of domain-specific designs for network architectures, such as convolutional filters [7, 8, 9] for vision tasks, or recurrent modules [10, 11] and attention mechanisms [12, 13] in the context of natural language processing. A domain-and-modality agnostic model that can be adapted to solve tasks across different modalities and domains has become a desideratum [14], motivating great efforts in transfer learning [15] and multi-modal learning [16]. Recently, transformer-based language models (LMs) [13, 17, 18, 19] exhibited impressive versatility across different domains and modalities. They have shown great performances for various language-based tasks [20] such as question answering [21, 22], or commonsense reasoning [23]. They have also been applied to non-language modalities [18]. For instance, GPT-2 [17] pretrained on language data can be efficiently fine-tuned to perform image classification and numerical computation [18].

When downstream tasks are language-based tasks, adapting pretrained LMs can be achieved without modifying the models' architecture. Typically, this adaptation is enabled via simple fine-tuning [24,

---

\*Equal contribution. Emails: Tuan Dinh (`tuan.dinh@wisc.edu`), Yuchen Zeng (`yzeng58@wisc.edu`)

36th Conference on Neural Information Processing Systems (NeurIPS 2022).

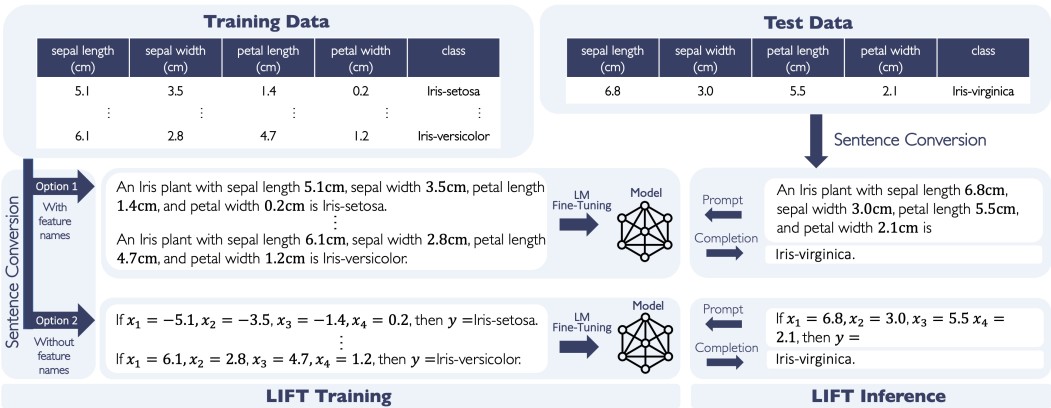

Figure 1: **A high-level illustration of the Language-Interfaced Fine-Tuning (LIFT) framework.** LIFT has a two-phase procedure: (1) converting the dataset into sentences and (2) fine-tuning the pretrained language model (*e.g.*, GPT) on the obtained sentences. This figure visualizes how LIFT can be applied to the Iris classification task. We first convert the Iris dataset into plain English sentences (left). Since feature names and the task description are available for this task, one could incorporate them as part of the prompt (as option 1 in the figure). (In Sec. 4.1, we show that adding such contextual information to prompts helps LIFT achieve higher predictive accuracy.) One may also choose to use a simpler prompt with a generic naming convention $(x_1, x_2, \ldots, x_d)$ for $p$ features (as option 2 in the figure). After the sentence conversion step, LIFT fine-tunes a pretrained LM with the sentence set without making any changes to model architecture or loss. At inference time, we convert the test samples to a sentence form using the same prompt, excluding the label part. LIFT performs surprisingly well in various non-language regression/classification tasks, and we summarize our main findings in Table 3. Note that to obtain a model for a given task, all we need here is to design proper sentence templates for LIFT and no changes to architecture or loss functions are needed.

25, 26, 27] or in-context few-shot learning methods [28, 29]. However, not altering the architecture may pose a limitation for transferring to non-language tasks. As their input and output formats are not in some language form, adapting LMs to these domains may seem to require architectural changes. Indeed, it has been a common practice to re-design the input/output layers and loss functions to accommodate a different predictive task. For instance, to adapt GPT-2 [21] to other modalities, the frozen pretrained transformer [18] adds new input/output layers to handle different types of input/output. To make such changes, one must have a good understanding of the underlying principles of LMs and an ability to make proper modifications at the code level.

A natural question that arises is whether such changes are necessary. In other words,

> Does language model fine-tuning work for non-language tasks
> **without** changing the architecture or loss function at all?

To answer this, we consider a simple fine-tuning procedure for LMs, referred to as **Language-Interfaced Fine-Tuning (LIFT)**. This procedure can be used to learn predictors for any classification or regression task. LIFT runs in two phases: (1) converting labeled samples into sentences, and (2) fine-tuning pretrained LMs on the sentence dataset without altering the architecture or loss function.

Fig. 1 illustrates how we fine-tune GPT with LIFT to solve the Iris classification task [30]. LIFT first converts each labeled sample into a sentence with two options. The first option is to incorporate feature names and the task description into the sentence template. In this example, we could convert a training sample **r** into "An Iris plant with sepal length **r.sepal_length**, sepal width **r.sepal_width**, petal length **r.petal_length**, and petal width **r.petal_width** is **r.class**." Here, we use the dot notation, *i.e.,* **r.**⋆ denotes the string conversion of the corresponding attribute of sample **r**. One may also adopt a simpler (and more generic) sentence template, such as "If x1=**r.x1**, x2=**r.x2**, ..., xp=**r.xp**, then y=**r.y**," if there are $p$ features. We then fine-tune LMs without changing either architecture or loss function. Then, we perform inference as follows. LIFT first converts test samples into sentences using the same template while leaving the prediction part empty. It then feeds the converted sentences as prompts to the fine-tuned model. The output tokens are parsed to provide the final predictions.

Our work empirically shows that LIFT can provide high-accuracy solutions for a variety of non-language tasks. Fig. 2 shows examples of real functions learned by GPT-J models [31] fine-tuned using LIFT given 1000 samples. Recall that LIFT does not require any changes in the architecture

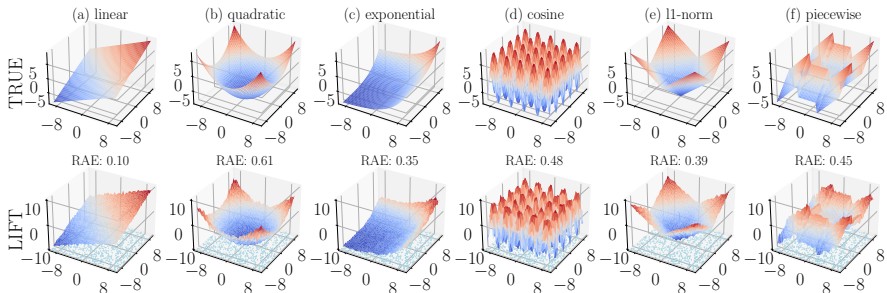

Figure 2: **Approximating various functions with LIFT using GPT-J.** We visualize the target functions (first row) and the predictor functions learned by LIFT on GPT-J (second row). Blue dots show the 1000 training samples. One can observe that LIFT well approximates the target functions.

or loss function, Thus, our findings show that such changes to architecture/loss function might *not* be necessary, even when the target predictive task is not a language task. Thus, LIFT can be almost perceived as a "no-code machine learning" framework as the data-to-sentence conversion is extremely straightforward even without extensive programming skills and machine learning backgrounds.

Motivated by these intriguing properties, we investigate the efficacy and limitations of LIFT on non-language tasks by conducting an extensive empirical study on a suite of classification and regression tasks. *First*, we observe that LIFT performs well across a wide range of low-dimensional classification and regression tasks. In most cases, it nearly matches (or slightly outperforms) the best baselines' performance. To further understand LIFT, we conduct experiments testing the fundamental learning properties, *e.g.,* its inductive bias, sample efficiency, ability to extrapolate, worst- and average-case noise robustness, and how the pretraining of LMs affects LIFT. *Third*, we study a few unique properties specific to LIFT, *e.g.,* context-aware learning with task-specific prompting, prediction calibration, and the additional use of LIFT for data generation. *Lastly*, to improve upon the basic fine-tuning, we employ a few techniques: two-stage fine-tuning with synthetic pretext tasks and data augmentation. Both techniques improve the performance of LIFT. We finally provide discussions on limitations and future investigations of LIFT.

**Scope of the study.** Our work proposes the use of natural language interface for learning with LMs via LIFT. We emphasize that our goal is *not* to achieve the state-of-the-art performance, but to investigate thoroughly: (i) what LIFT can and cannot do, (ii) properties of models fine-tuned via LIFT, and (iii) whether we can improve LIFT with advanced techniques.

## 2 Methodology and Experimental Setup

**LIFT training.** To fine-tune a pretrained LM with LIFT on a target supervised learning task, we apply two steps: (1) convert each sample into a sentence with a fixed template, and (2) fine-tune LMs with sentence datasets. We use the default cross-entropy loss for token prediction in LMs. Our generic template (without feature names and task description) for sample **r** is

$$\underbrace{\text{When we have x1=}\mathbf{r.x1}\text{, x2=}\mathbf{r.x2}\text{, } \ldots \text{, xp=}\mathbf{r.xp}\text{, what should be y?}}_{\text{question}} \underbrace{\#\#\#}_{\text{q/a separator}} \underbrace{y = \mathbf{r.y}}_{\text{answer}} \underbrace{@@@}_{\text{end of answer}} ,$$

if **r** has $p$ attributes. Here, we use the separator convention recommended by OpenAI [32] – "###" for question/answer separation, and "@@@" for end of generation. When attributes names and task descriptions are available, one can use a more informative prompt (shown in Fig. 1) with all actual prompts are provided in Sec. 4.1. We report learning curves of LIFT on several tasks in Appendix E.5.

**LIFT inference.** For inference, we use the same prompt template except for the answer and end-of-answer parts. Once the fine-tuned LM completes the test prompt, we simply parse the output tokens. For classification, we simply compare the generated text with the string representation of the class names. For regression, we convert the generated string into a number. For instance, with the output being "$y$=10.35@@@extratokens", we split the output sentence by the stop separator "@@@" into two parts. Taking the first part "$y$=10.35", we parse the value "10.35" as our final prediction.

The generated output might be *invalid*. For classification tasks, the output string may not match any actual class, which we declare as misclassification. Note that one may obtain better accuracy by returning its closest class using string metrics. For regression tasks, we consider output invalid if the string-to-number parsing fails. In these cases, we adjust the generation randomness by increasing the decoding temperature [33, 34, 35] from 0 (deterministic mode) to 0.75 (random mode). We repeat

the inference up to five times, then return the average value of the training set if all attempts fail. Note that invalid output occurs very rarely (less than or around 1% in most tested cases).

For evaluation metrics, we use accuracy for classification tasks, and RMSE, RAE errors for regression tasks, where $\text{RAE} = \sum_{i=1}^{n} |\hat{y}_i - y_i| / \sum_{i=1}^{n} |\frac{1}{n} \sum_{j=1}^{n} y_j - y_i|$ and $\text{RMSE} = \sqrt{\sum_{i=1}^{n} (\hat{y}_i - y_i)^2 / n}$ on each dataset $\mathcal{D} = \{(\boldsymbol{x}_i, y_i)\}_{i=1}^{n}$ with $n$ samples, features $\boldsymbol{x} \in \mathcal{X} \subset \mathbb{R}^p$, and outcome $y$.

**Pretrained LMs.** We apply LIFT on two pretrained LMs: GPT-J [31] and GPT-3 [19]. To fine-tune GPT-J, we use LoRA [24], a parameter-efficient method that constrains weight matrix updates to be low-rank. For experiments on GPT-J, we used `p3.8xlarge` and `p3.2xlarge` instances from AWS and RTX3090 GPUs in the local server. Since GPT-3 is not fully publicly available, we use the API provided by OpenAI to perform black-box GPT-3 fine-tuning. More details are in Appendix C.2.1.

**Datasets.** We design and select a wide range of datasets to better understand the behavior of LIFT. For *classification*, we use three types of non-language data: low-dimensional synthetic datasets, real tabular datasets in OpenML [36], and vision datasets (MNIST [37], Fashion-MNIST [38] and their permuted variants [39]). For *regression*, we use both synthetic and real datasets. For synthetic datasets, we defined samples $(\boldsymbol{x}_i, y_i)$ of input-output pair as $\text{y} \sim f(\text{x}) + \mathcal{N}(0, \sigma^2)$, where $\sigma^2 \geq 0$ is the noise level. Unless otherwise stated, we sample the feature x uniformly from a hypercube $[L, U]^p$, where $L$ and $U$ are minimum/maximum feature values, and $p$ is the number of features. Following the suggestion by [40], we consider various functions $f$ for regression tasks: (i) linear function, (ii) quadratic function, (iii) exponential function, (iv) cosine function, (v) $\ell 1$-norm function, and (vi) piece-wise linear function. Their 2D visualizations are provided in the first row of Fig. 2. We also use four real datasets: Medical Insurance (Insurance) [41], Combined Cycle Power Plant (CCPP) [42], Servo [43], and Student Performance (Student) [44]. More details are included in Appendix C.1.

**Baselines.** We consider standard learning algorithms [45, 46]. For classification, we use logistic regression (*LogReg*), decision tree (*DT*), k-nearest neighbor (*KNN*), support vector machine with Gaussian kernel (*SVM*), a four-layer ReLU neural network (*MLP*) with 200 neurons per hidden layer, random forest (*RF*), and XGBoost (*XG*). We also use the majority class classifier (*MCC*) that outputs the most dominant class. For regression, we use polynomial regression (*PR*), kernel ridge regression (*KR*) with radial basis function kernel, $k$-nearest neighbors (*KNN*), a three-layer ReLU neural network (*MLP*) with 50 hidden neurons per each layer, Gradient Boosting Trees (*GBT*), random forest (*RF*), and Gaussian process (*GP*). For hyperparameter selection, we apply the grid search on a set of parameters' values and use cross-validation on the training set (see details in Appendix C.2).

## 3 Basic Findings of LIFT

Table 3 summarizes our main findings. We also study sample complexity (Sec. 3.2), comparison with in-context learning (Sec. 3.3), models' decision boundaries (Sec. 3.4), and the effect of LMs' pretraining on LIFT (Sec. 3.6). Appendix E provides additional results, including the effect of input and output layers (E.1), model size (E.2), and LIFT for Ridge regression (E.4).

### 3.1 How Well Does LIFT Perform on Standard ML Tasks?

**Classification.** Table 4 compares classification accuracies between algorithms on a wide range of tasks. We observe that LIFT achieves comparable performance to most baselines. In most cases, LIFT/GPT ranks highly in the top three best-performing methods. We find that LIFT can learn non-linear relationships between features and the responses: LIFT/GPT-3 achieves 81.17% accuracy on the circle dataset, while logistic regression failed to perform better than the MCC (50%). As the difficulty of tasks varies, which can be estimated by the average performance of baselines, LIFT also suffers from performance degradation. LIFT can perform comparably well even when the number of features is as large as hundreds, though the limited number of tokens as inputs to LMs restricts the number of features LIFT can input. However, when the number of classes is large (say 100s), both LIFT/GPT models have lower accuracies than many baselines, though they manage to be better than MCC. For instance, on the 100-class Margin dataset, the accuracy gap between LIFT/GPT-J and the best algorithm (RBF-SVM) is nearly 30%. Note that LIFT can directly use raw data while most baselines require feature scaling and normalization for good performance. More results are provided in Appendix D.1.1, including comparisons with methods leveraging larger models.

**Regression.** Tables 19 and 20 present our *function approximation* comparison. For the low-dimensional cases, LIFT is comparable to baselines. Still, it fails to beat the strongest baselines, such

Table 3: **Summary of the main findings.**

| Topic | Findings |
|---|---|
| Overall performance | On various classification tasks, LIFT achieves accuracies comparable to strong baselines (Table 4). For regression, LIFT well approximates different types of low-dimensional functions (Fig. 2) but does not perform well for high-dimensional cases (Table 19). |
| Robustness | For regression, LIFT is robust to outliers in training data (Fig. 28). For classification, LIFT is comparable to baselines under label corruption on training data (Fig. 29) but more vulnerable to feature corruption on test data (Table. 33). |
| Context-aware learning | We can improve LIFT on classification tasks by designing prompts to specify feature names and the target task. The improvement is significant when the description of the feature names and the target task can be interpreted with common knowledge (Table 9). |
| Two-stage training | Warming up LIFT with pretext tasks using synthetic data improves the prediction performance, especially in the low-data regime (Fig. 40). |
| Data augmentation | For classification tasks, training with augmented data significantly improves the tolerance of LIFT against perturbed test data (Table 12). |

Table 4: **Accuracies ($\uparrow$) on classification datasets.** We evaluate LIFT/GPTs on 2D synthetic data, tabular data in OpenML [36], and image data, varying number of features ($p$) and data classes ($c$). Overall, LIFT/GPTs perform comparably well across tasks, adapting to non-linear datasets (circles, two circles) beyond the capacity of logistic regression. For OpenML datasets, they achieve competitive performances with the best methods, *e.g.*, XGBoost). The performance degrades when more classes are given, *e.g.*, $c$=100. They achieve competitive accuracies on both MNIST and Fashion MNIST. Note that MNIST's classes are not fully balanced; thus, MCC achieves 11.35% instead of 10%. Table 17 provides the full comparison with all baselines (KNN, MLP, Random Forest).

| Dataset (ID) | $p$ / $c$ | MCC | LogReg | DT | RBF-SVM | XG | LIFT/GPT-J | LIFT/GPT-3 |
|---|---|---|---|---|---|---|---|---|
| **Synthetic Data** | | | | | | | | |
| circles (3) | 2 / 2 | 50.00 | 48.58±1.94 | 77.42±0.24 | **83.08±0.59** | 81.42±0.31 | 79.95±1.53 | 81.17±0.42 |
| two circles (6) | 2 / 2 | 50.00 | 49.83±4.18 | 75.50±0.20 | 80.00±0.54 | 79.25±0.35 | 75.92±1.65 | **81.42±0.82** |
| blobs (2) | 2 / 4 | 25.00 | **96.75±0.00** | 96.08±0.82 | **96.75±0.00** | 96.17±0.12 | 96.17±0.59 | 96.67±0.24 |
| moons (4) | 2 / 4 | 50.00 | 88.58±0.12 | 99.25±0.41 | **100.00±0.00** | 99.83±0.12 | 99.58±0.42 | **100.00±0.00** |
| 9Clusters (1) | 2 / 9 | 11.25 | 100.00±0.00 | 100.00±0.00 | 100.00±0.00 | 100.00±0.00 | 99.75±0.00 | **100.00±0.00** |
| **Tabular Data (OpenML)** | | | | | | | | |
| Customers (1511) | 8 / 2 | 68.18 | **87.12±0.54** | 85.98±0.53 | 86.36±0.00 | 85.23±0.00 | 85.23±1.61 | 84.85±1.42 |
| Pollution (882) | 15 / 2 | 50.00 | 58.33±11.79 | **77.78±3.93** | 58.33±6.81 | 63.89±7.86 | 63.89±3.93 | 63.89±7.86 |
| Spambase (44) | 57 / 2 | 60.59 | 93.27±0.00 | 90.7±0.14 | 93.70±0.00 | **95.87±0.00** | 94.03±0.54 | 94.90±0.36 |
| Hill-Valley (1479) | 100 / 2 | 49.79 | 77.78±0.00 | 56.38±0.89 | 68.72±0.00 | 59.26±0.00 | **100.00±0.20** | 99.73±0.19 |
| IRIS (61) | 4 / 3 | 33.33 | 96.67±0.00 | 97.77±3.85 | **100.00±0.00** | **100.00±0.00** | 96.67±0.00 | 97.0±0.00 |
| TAE (48) | 5 / 3 | 35.48 | 45.16±4.56 | 65.59±5.49 | 53.76±6.63 | **66.67±8.05** | 61.29±6.97 | 65.59±6.63 |
| CMC (23) | 9 / 3 | 42.71 | 49.49±0.83 | 56.72±0.32 | 56.50±0.97 | 52.43±0.42 | 49.83±0.28 | **57.74±0.89** |
| Wine (187) | 13 / 3 | 38.89 | **100.00±0.00** | 93.52±2.62 | **100.00±0.00** | 97.22±0.00 | 93.52±1.31 | 92.59±1.31 |
| Vehicle (54) | 18 / 4 | 25.88 | 80.39±1.00 | 63.92±2.37 | **81.18±0.48** | 73.14±0.28 | 64.31±2.37 | 70.20±2.73 |
| LED (40496) | 7 / 10 | 11.00 | 68.67±0.94 | 66.33±2.87 | 68.00±0.82 | 66.00±0.82 | 65.33±0.47 | **69.33±2.05** |
| OPT (28) | 64 / 10 | 10.14 | 96.53±0.22 | 89.8±1.09 | 97.95±0.00 | 97.48±0.17 | 98.22±0.11 | **98.99±0.30** |
| Mfeat (12) | 216 / 10 | 10.00 | 97.67±0.12 | 87.67±1.05 | **98.83±0.24** | 96.75±0.00 | 94.17±1.75 | 93.08±0.24 |
| Margin (1491) | 64 / 100 | 0.94 | 81.35±0.15 | 43.86±1.21 | **81.98±0.30** | 70.21±0.29 | 50.23±1.33 | 59.37±0.92 |
| Texture (1493) | 64 / 100 | 0.94 | 81.67±0.97 | 46.88±1.93 | **83.44±0.89** | 70.73±1.41 | 50.32±2.18 | 67.50±1.42 |
| **Image Data** | | | | | | | | |
| MNIST | | 11.35 | 91.95±0.69 | 87.42±0.64 | 97.70±0.97 | 97.69±0.04 | 97.01±1.15 | **98.15±0.67** |
| Permuted MNIST | 784 / 10 | 11.35 | 92.58±0.04 | 87.87±0.69 | **98.06±0.31** | 97.62±0.09 | 95.80± 0.07 | 96.25±0.35 |
| Fashion MNIST | | 10.00 | 85.59±0.09 | 80.52±0.40 | **90.59±0.02** | 90.19±0.04 | 85.10 ±0.19 | 90.18 ±0.12 |
| Permuted F-MNIST | | 10.00 | 84.95±0.84 | 79.91±0.93 | 88.04±1.69 | **89.93±0.14** | 82.25±0.27 | 88.92±0.71 |

as GP, as GPT models measure the error by comparing tokens instead of measuring how close the prediction values are to true values. We conjecture that we can improve our performance by level encoding, *i.e.*, representing numerical values as binary values. We also investigate the *interpolation and extrapolation* of LIFT and defer the details to Sec. D.1.1. All methods fail to extrapolate and interpolate well for all functions, and the interpolation performance of LIFT is only good in the linear regression case. Interestingly, LIFT tends to output seen values (from training data) for extrapolation.

## 3.2 How Many Samples Does LIFT Need?

We investigate whether LIFT is sample efficient. Fig. 25 in Appendix shows the sample complexity evaluation on classification and regression tasks. We find that GPT models can be quickly fine-tuned to learn new tasks with LIFT. For *classification*, as the number of classes increases (left to right columns in Fig. 25a), LIFT does need more samples for adaptation, probably because the data input

Table 5: **Comparison of accuracies (↑) between ICL and fine-tuning with LIFT on OpenML datasets.** "LIFT/Full-Data" and "LIFT/Subset" represent LIFT on the full dataset and and its subset used correspondingly in the ICL setting (number of prompts). Here, the size of subset is chosen to satisfy the LMs' context length. Overall, LIFT/GPTs on full data achieve the best performances. However, when using the same number of samples, LIFT and ICL are more comparable in most cases. Note that both methods may be worse than MCC due to the limited training data in some cases.

| Dataset (ID) | #Prompts | MCC | GPT-J | | | GPT-3 | | |
|---|---|---|---|---|---|---|---|---|
| | | | In-Context | LIFT/Subset | LIFT/Full-data | In-Context | LIFT/Subset | LIFT/Full-data |
| Breast (13) | 35 | 70.69 | 56.90±19.51 | **58.62±2.44** | 64.94±11.97 | 62.07±1.41 | **70.69±0.00** | 71.26±1.62 |
| TAE (48) | 50 | 35.48 | **34.33±1.47** | 32.26±9.50 | 61.29±4.56 | **37.64±4.02** | 33.33±1.52 | 65.59±6.63 |
| Vehicle (54) | 14 | 25.88 | **25.49±0.55** | 26.04±1.69 | 64.31±2.37 | **28.82±2.10** | 23.73±2.27 | 70.20±2.73 |
| Hamster (893) | 43 | 53.33 | 48.89±3.14 | **60.00±10.88** | 55.55±16.63 | **57.78±6.29** | 53.33±0.00 | 53.33±0.00 |
| Customers (1511) | 29 | 68.18 | 56.06±17.14 | **59.85±2.84** | 85.23±1.61 | 60.61±1.42 | **63.26±6.96** | 84.85±1.42 |
| LED (40496) | 33 | 68.67 | 10.00±0.82 | **13.04±3.27** | 65.33±0.47 | 8.00±1.63 | **11.33±2.62** | 69.33±2.05 |

and output spaces are more complex to learn. For *regression* tasks, we find that 1000 samples are sufficient for LIFT to have a small RMSE, similar to other baselines. There exist some functions (*e.g.*, cosine and piecewise) where LIFT has lower sample complexity than popular baselines.

### 3.3 Language-Interfaced Learning: LIFT versus In-Context Learning (ICL)

Beyond fine-tuning (with LIFT), our language-interfaced learning framework can be used for other learning methods for LMs, including in-context learning (ICL) [47, 48, 19] that performs inference on new tasks without fine-tuning by conditioning on a few training examples. Table 5 compares the classification performances between (a) ICL, (b) LIFT trained on a subset with $n$ samples, and (c) LIFT trained on the full dataset. Note that the number of training samples ($n$) used for ICL depends on the context length of given LMs. As we can see, LIFT using the full dataset always achieves the best performances. However, LIFT/Subset and ICL are more comparable in most cases when they use the same number of training samples, which are sufficiently small for ICL methods to fit in LMs.

**Remark.** *One can replace fine-tuning with ICL in our language-interfaced procedure when the target tasks require fewer training samples.*

### 3.4 Can We Understand the Inductive Biases of Language Models via LIFT?

To better understand LIFT/GPTs' inductive biases, we investigate their classification decision boundaries varying the boundaries' complexity, as shown in Fig. 6. We first train a binary-class neural network and use its snapshots at different training epochs to construct datasets having decision boundaries at different complexity levels (first column in Fig. 6). We observe that LIFT/GPT models adapt well to three boundaries and capture their rough shapes. Furthermore, their boundary shapes are axis-parallel, similar to the boundary of tree-based classifiers. They also show a lot of fractals similar to the observations on some convolution neural networks [49]. See Appendix D.1.3 for results of 3-class and 5-class datasets and quantitative measurements.

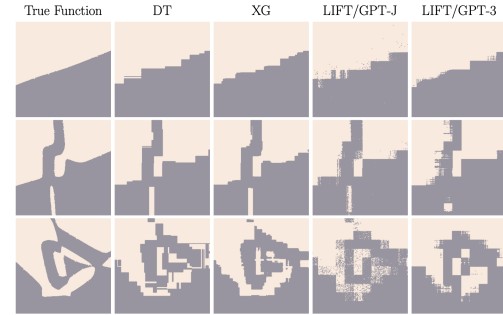

Figure 6: **Decision boundary visualization.** We use three snapshots of a trained network to construct datasets having labels as their predictions (the first column). Top to bottom: snapshots with more training epochs, corresponding to more complex boundaries. LIFT/GPTs adapt well on different boundaries.

### 3.5 How Robust is LIFT?

We investigate the robustness of LIFT against the outlier samples in training data and the feature corruption on test data. Appendix D.1.4 provides additional experimental results, including the robustness for the case of label corruption on training data and class-imbalanced data.

**Robustness to outliers in training data.** We consider regression tasks where we have outliers whose outcome $y$ is not consistent with the majority of samples in terms of fitting $(\boldsymbol{x}, y)$. Fig. 28a compares RAE values of methods with and without outliers (2% outliers in the training set). LIFT/GPT models are among the most robust ones: their performances are almost unaffected, while baselines suffer

Table 8: **Accuracies (↑) of LIFT with different LMs.** We compare variants of LIFT with different LMs: LIFT/GPTs using GPTs pretrained on natural language data (our models), LIFT/Rand-GPT-J using a randomly initialized GPT-J, LIFT/CodeGen and LIFT/CodeParrot using LMs pretrained on programming language data, and LIFT/Gibberish using GPT-J fine-tuned on gibberish data.

| Dataset (ID) | MCC | LIFT/GPT-3 | LIFT/GPT-J | LIFT/Rand-GPT-J | LIFT/Gibberish | LIFT/CodeGen | LIFT/CodeParrot |
|---|---|---|---|---|---|---|---|
| Blobs (2) | 25.00 | 96.67± 0.24 | 96.17± 0.59 | 25.65± 1.58 | 96.42± 0.24 | 93.67± 0.72 | 93.39± 1.82 |
| Two Circles (6) | 50.00 | 81.42± 0.82 | 75.92± 1.65 | 49.88± 5.01 | 68.67± 1.50 | 53.02± 0.66 | 50.08± 2.47 |
| Iris (61) | 33.33 | 97.0± 0.00 | 96.67± 0.00 | 27.78± 20.79 | 94.44± 1.57 | 43.31± 6.67 | 60.00± 8.82 |
| Customers (1511) | 68.18 | 84.85± 1.42 | 85.23± 1.61 | 52.47± 7.15 | 67.43± 1.42 | 45.96± 8.96 | 43.11± 3.34 |
| Wine (187) | 38.89 | 92.59± 1.31 | 93.52± 1.31 | 22.22± 15.71 | 84.26± 3.46 | 77.78± 0.00 | 33.88± 3.87 |
| LED (40496) | 11.0 | 69.33± 2.05 | 65.33± 0.47 | 11.68± 4.44 | 72.67± 1.25 | 11.00± 4.00 | 23.46± 13.85 |

huge performance drops. Furthermore, we evaluate models under various percentages of outliers (1%, 2%, 5%, 10%, 20%), as shown in Fig. 28b. Compared to the robust baselines (median-3NN and median-5NN) [50], LIFT/GPT-3 is comparably robust, while LIFT/GPT-J is more vulnerable when more outliers are present.

**Robustness to feature corruption on test data.** Given a clean test data $(x, y)$ having feature $x$ and label $y$, we explore whether adding small perturbation $\delta$ on the feature changes the performance; we measure the accuracy of LIFT on perturbed data $(x + \delta, y)$. We apply transfer attack [51] since we do not have full access to the GPT-3 model, and finding adversarial examples in the discrete input space is complex [52]. Table 7 reports robustness results on MNIST classification under PGD attacks transferred from LeNet-5. The perturbation radius is set to $\varepsilon \in [0, 0.01, 0.1, 0.3]$ where MNIST pixel value is within [0,1]. We compare three networks: LeNet-5, MLP (2 hidden layers with 300 and 100 neurons), and LIFT/GPT-3. When

Table 7: **Accuracies (↑) under the perturbation on the input feature of MNIST data.** See the full results in Table 33 in Appendix D.1.4.

| Source | PGD attack on LeNet-5 | | |
|---|---|---|---|
| Target | LeNet-5 | MLP | LIFT/GPT-3 |
| $\varepsilon = 0$ | 99.22 | 98.09 | 98.15 |
| $\varepsilon = 0.01$ | 97.27 | 97.77 | 44.88 |
| $\varepsilon = 0.1$ | 26.80 | 93.99 | 33.66 |
| $\varepsilon = 0.3$ | 0.00 | 36.62 | 20.31 |

$\epsilon \in \{0.01, 0.1\}$, LIFT/GPT-3 tolerates random noise (as in Table 33) but cannot tolerate transferred adversarial attack, implying that the adversarial attack on LeNet-5 is transferred to LIFT/GPT-3.

## 3.6 Does LIFT Need Large-Scale Models Pretrained on Natural Language Data?

We investigate the requirement of pretrained LMs for which LIFT performs well. We compare variants of LIFT under different types of LMs: GPTs pretrained on natural language data (our models), a large LM without pretraining (Rand-GPT-J), and LMs pre-trained on non-human language data, including CodeParrot [53] and CodeGen-2B-mono [54] trained mainly on programming language data, and a GPT-J fine-tuned on Gibberish data [55]. See Appendix D.1.5 for the detailed setting.

**Does LIFT only need a large pretrained model?** To answer this question, we compare performances of LIFT when GPTs are pretrained (LIFT/GPTs) and when GPT-J have weights being randomly initialized (LIFT/Rand-GPT-J). More specifically, for LIFT/Rand-GPT-J, we randomly initialized a GPT-J model and fine-tuned the whole model (instead of LoRA). As shown in Table 8, accuracies of LIFT/Rand-GPT-J are much lower than those of our models (LIFT/GPTs), across all datasets. These results indicate that LIFT benefits from pretraining, not just from the large-scale design of LMs.

**Does LIFT need a model trained on natural language data?** As shown in Table 8, LIFT/GPTs perform much better than LIFT/CodeGen and LIFT/CodeParrot for all datasets. This implies that LIFT may perform better with LMs pretrained on natural language data. When the pretrained GPT-J is fine-tuned on gibberish data [55], the accuracies drop for a few tasks and are lower than LIFT/GPTs overall. However, LIFT/Gibberish still achieves comparably good performance and its small performance gaps to LIFT/GPT-J can be attributed to the relatively light impact of fine-tuning on large pretrained LMs. Thus pretraining on natural language data is necessary for LIFT.

## 4 Evaluation of LIFT-Specific Learning Properties

In this section, we study the behavior of LIFT in a more fine-grained manner.

### 4.1 Does LIFT Benefit from Incorporating Feature Names?

Unlike standard machine learning algorithms, LIFT can be provided context information by incorporating the feature names and task descriptions in the prompts. Intuitively, this incorporation may improve the sample complexity of LIFT as the prior knowledge already learned in the pretraining

Table 9: **The effect of using feature names on LIFT**. We compare classification accuracy (↑) of LIFT/GPT-3 when feature names provided in the target dataset *are and are not* incorporated into the prompts. We provide four versions of LIFT when feature names are correctly incorporated (Correct-Names columns) and when feature names are randomly shuffled (Shuffled-Names columns). We evaluate models on three OpenML datasets, including CMC (23), TAE (48), Vehicle (54), and German. We also compare our models with two baselines: the majority class classifier (MCC) and XGBoost. As a result, all LIFT models achieve better performance than MCC. Among the evaluated models, LIFTs with correct feature names achieve the best accuracies on both TAE, Vehicle, and German datasets while achieving the comparable accuracies to the best model on the CMC dataset. *Two designs of the prompt format result in the same template for the Vehicle dataset.*

| Dataset (ID) | MCC | LIFT | | | | | |
| --- | --- | --- | --- | --- | --- | --- | --- |
| | | W/o Names I | W/o Names II | Shuffled-Names I | Shuffled-Names II | Correct-Names I | Correct-Names II |
| CMC (23) | 42.71 | **57.74±0.89** | 57.40±1.37 | 56.27±2.06 | 57.06±4.24 | 57.40±1.09 | 56.27±2.22 |
| TAE (48) | 35.48 | 65.59±6.63 | 66.67±5.48 | 60.22±6.72 | 64.52±8.53 | **69.89±9.31** | **69.89±6.72** |
| Vehicle (54) | 25.88 | 70.20±2.73 | 71.96±3.09 | 70.20±5.34 | 69.22±2.72 | **75.29±2.04*** | |
| German | 70.00 | 71.33 ± 5.20 | 67.83 ± 2.72 | 73.00 ± 1.87 | 71.67 ± 0.94 | 72.33 ± 1.70 | **74.17± 1.25** |

phase may help LIFT predict better. We design seven prompt templates to assess how incorporating feature names affects the performance of LIFT (see more details in Appendix D.2.1). We empirically verify this intuition and show our results in Table 9 for several classification tasks using pretrained GPT-3 models. We first observe that all LIFT models outperform MCC with significant accuracy gaps, indicating that they are all properly trained. Second, we observe that correctly incorporating feature names helps boost the performances of LIFT for datasets except for CMC. Third, if we use similar prompts with shuffled feature names (Shuffled-Names I, II), then the performance of LIFT drops by a significant margin. These results imply that the aforementioned performance improvements are indeed due to proper prompting with correct feature/value association.

## 4.2 Is LIFT Well Calibrated?

We investigate whether LIFT is *calibrated*, *i.e.*, the prediction reflects the confidence, by exploring how LIFT performs under various noise levels, as shown in Fig. 36 in Appendix. We conduct experiments on six synthetic regression datasets, each consisting of 1,000 noisy training samples shown as blue markers in the first row. To be specific, we generate (1) the input $x$ following the guideline in Sec. C.1 for regression tasks and (2) the noisy outcome $y$ where the standard deviation of noise $\sigma(x) = (x + 10)/10$ increases along the $x$-axis (from $x = -10$ to $x = 10$), and study how different noise level affects the predictive behavior of LIFT. In the inference phase, we set the decoding temperature $T = 1$ for LIFT to make random predictions. For visualization purposes, we generate an additional 103 samples uniformly in $[-10, 10]$ for each task and plot the standard deviation of 20 LIFT/GPT-J predictions on each sample in the bottom row of Fig. 36. Note that the bottom row of Fig. 36 shows that the standard deviation of LIFT/GPT-J's prediction nearly matches that of noisy training samples (observations) across different tasks. These results imply that LIFT/GPT-J is calibrated. Similarly, Fig. 37 of Sec. D.2.2 shows that LIFT/GPT-3 is calibrated.

## 4.3 Can We Use LIFT for Data Generation?

Generative models have been widely used in computer vision [56, 57, 58]. Beyond classification and regression tasks, we study whether LIFT can be used for generative tasks, *i.e.*, learning the underlying data distribution and generating realistic data samples. In particular, we consider two image generation tasks on MNIST dataset: (a) generating an image given a digit number, and (b) completing an image given a digit number and its pixels on the top half of the image. Fig. 10a and Fig. 10b show our generated images for the two tasks respectively. We observe that the generated images have the correct digit shape and reasonably high quality in most cases, especially for the image completion (Fig. 10b). See Appendix D.2.3 for more details.

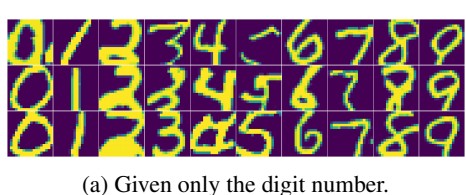
(a) Given only the digit number.

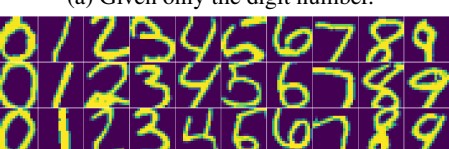
(b) Given the digit number and a half of image pixels.

Figure 10: **Generating MNIST images using LIFT/GPT-J.** We observe that LIFT/GPT-J can generate images of comparably high quality. The temperature is set to 1.

# 5 Improving LIFT with Existing Techniques

We improve LIFT with advanced techniques: two-stage fine-tuning and data augmentation.

## 5.1 Two-Stage Fine-Tuning for LIFT with Synthetic Pretext Tasks

In Sec. 3.2, we observe that LMs need a sufficient number of samples to start adapting. We suspect that LMs' adaptation to non-language tasks contains two phases: (1) learn the task description, *i.e.,* input space, label space, and sentence templates [47, 59], and (2) learn the target task. Thus, we consider utilizing synthetic data to describe the task for LMs in the first phase, thus reducing the sample complexity. This results in a new two-stage training procedure for LIFT.[1] In particular, for any given dataset, we first generate two pretext tasks with simple synthetic Gaussian datasets (discussed in C.1) sharing the same number of features and the label space

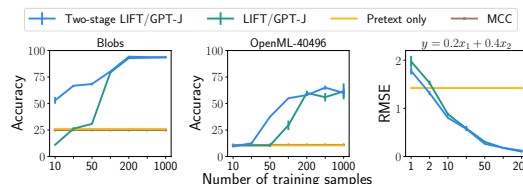

Figure 11: **Two-stage fine-tuning.** The two-stage method (blue) applies LIFT first on synthetic pretext data before the real datasets, outperforming fine-tuning (green) when training data is small. The full experiment results are presented in Fig. 40.

(for classification tasks) or the range of responses' values (for the regression tasks) to the actual data. We apply LIFT on pretext tasks for a few (2 or 3) epochs, then continue LIFT with the target (given) dataset. For GPT-3, it is unclear how to keep the order of samples not shuffled with the current black-box API during the fine-tuning stage. Hence, we only provide the experimental results of GPT-J. Fig. 11 shows that two-stage fine-tuning improves LIFT over the original fine-tuning when the number of training samples is small on both classification and regression tasks.

## 5.2 Data Augmentation

Data augmentation [61] is a simple tool for improving the generalization performance for various classification problems. Here, we investigate whether data augmentation benefits LIFT. Table 12 shows the effect of adding random noise in the training data on the performance of LIFT/GPT-J for the MNIST classification problem. Here, we test each model on three settings: (1) clean data, (2) Gaussian noise, and (3) signed constant noise. We allow each noise can perturb up to the magnitude of $\epsilon \in [0, 1]$ at each dimension (*i.e.*, each pixel) when the black/white pixel of MNIST is represented in the $[0, 1]$ range. We defer the generation procedure of random Gaussian noise to Sec. D.1.4 in Appendix.

Table 12: **Accuracies (↑) of LIFT with/without data augmentation (DA), as well as baselines (LeNet-5, MLP) on MNIST.** Each row represents different ways of training, and each column means different test data. Data augmentation (DA) means that we are using a noisy version of MNIST training data by adding Gaussian noise. Given an MNIST image having range [0,1], the noise is added in the $L_\infty$ ball with radius $\epsilon$. One can confirm that the data augmentation significantly improves the tolerance of LIFT/GPT-J against perturbed test data in both Gaussian and signed constant noise. For each column, we boldfaced the highest value among baselines and the highest value among LIFT/GPT-J.

| | Clean | Gaussian noise | | Signed const. noise | |
|---|---|---|---|---|---|
| | $\epsilon = 0$ | $\epsilon = 0.01$ | $\epsilon = 0.1$ | $\epsilon = 0.01$ | $\epsilon = 0.1$ |
| LeNet-5 | **99.22** | **99.25** | **99.20** | **99.26** | **99.06** |
| MLP | 98.09 | 98.05 | 97.70 | 98.08 | 97.39 |
| LIFT/GPT-J | **96.88** | **95.27** | 56.14 | 55.83 | 27.73 |
| LIFT/GPT-J, DA (Gaussian, $\epsilon = 0.05$) | 93.80 | 94.39 | 93.40 | 93.46 | 61.24 |
| LIFT/GPT-J, DA (Gaussian, $\epsilon = 0.1$) | 93.78 | 94.31 | **94.98** | **94.12** | **75.25** |

One can observe that LIFT/GPT-J without any data augmentation (DA) is vulnerable to random noise, unlike existing baselines (LeNet-5 and MLP). However, when we apply data augmentation, *i.e.*, train LIFT/GPT-J with noisy training data, the accuracy improves significantly for the perturbed (either adding Gaussian noise or Signed constant noise) test data. This shows the effectiveness of simple data augmentation in LIFT. Exploring the effect of other data augmentation schemes, *e.g.*, mixup [62] and its variants [63, 64, 65], is remained an interesting future work.

# 6 Related Works

**Fine-tuning for adapting LMs to non-language tasks.** Fine-tuning [66] pretrained LMs is the standard practice for learning downstream tasks, which may involve simple architecture modifi-

---

[1]The recent work [60] demonstrates the usefulness of the intermediate fine-tuning method for LMs. However, they focus on self-supervised objectives for fine-tuning pretrained LMs for few-shot in-context learning.

cations, such as adding linear layers [67, 68] or freezing layers [18, 69, 70]. The recent progress focuses on *parameter-efficient* techniques for reducing trainable parameters, including adapter-based fine-tuning [25, 26, 27] that trains additional small residual blocks between layers, freezing-based fine-tuning [71, 18, 72] that freezes most of the pretrained parameters, and distillation-based fine-tuning [73]. Our LIFT/GPT-J is fine-tuned with LoRA [24], a parameter-efficient method approximating the weight updates using low-rank matrices.

To directly adopt existing fine-tuning methods of LMs for non-language tasks, it is common practice to modify the input/output layers and the loss functions, which may cause undesired behaviors like catastrophic forgetting [66, 74]. Our work is highly motivated by Frozen Pretrained Transformer (FPT) [18] that directly fine-tunes GPT-2 [21] pretrained on language tasks for other modalities by freezing most pretrained parameters and adding only input and output layers for the modality adaptation. Unlike FPT, our method requires no such changes in the architecture and objective function. Several works also extend the existing LMs to handle different input data types, such as images [75, 76], audio [77], tabular data [78], and knowledge base [79] by updating the pretraining phase with these data and their corresponding tasks or using general-purpose architecture [80]. Our work is based on GPT language models trained *only* on textual data.

**Analyzing the adaptability of LMs.** Similar to ours, recent works [81, 82, 83, 82] attempt to understand and quantify the adaptability [20] and capacity of large LMs, such as Big-Bench [82] with a new benchmark of more than 200 tasks on a diverse set of topics.

**General-purpose models.** A primary goal of our work is to push the limit of the existing generalist language models (*e.g.,* GPT-3 [19]) to other modalities and domains, supporting the idea of building a domain-and-modality agnostic generalist model [19, 84, 3, 85, 86, 87, 88, 89]. Note that LIFT can be applied to any generalist model with LM-like architectures, such as GATO [89]. Furthermore, our work shares the general goal with automated machine learning (AutoML) [90, 91] in improving the usability of machine learning, though LIFT uses only a single pretrained LMs for all tasks while AutoML automates the standard machine learning pipeline from a set of existing algorithms.

# 7  Discussion and Conclusion

We propose the use of **language-interfaced** framework, via **Language-Interfaced Fine-Tuning (LIFT)**, for using LMs to solve non-language downstream tasks without changing the models' architecture or loss function. LIFT first converts labeled samples into sentences and then fine-tunes pretrained LMs on the sentence dataset using the standard fine-tuning method and loss function. Via an extensive empirical study, we show that LIFT/GPT performs relatively well on low-dimensional classification and regression non-language tasks. Furthermore, LIFT/GPTs are robust in several practical settings, and can properly calibrate the predictions and generate realistic data samples. LIFT can be improved using in-context feature names, two-stage fine-tuning, and data augmentation. Moreover, our work is arguably one of the *first to thoroughly study the efficacy of language-interfaced learning framework* with pretrained language models on standard regression and classification tasks, paving the way for enabling "no-code machine learning with language models."

**Limitations and open questions.** Despite promising performances on various tasks and settings, we observe some limitations of LIFT to basic learning tasks. LIFT/GPT do not perform well if the features have high dimensions (for regression) or when the number of classes is large (for classification). In addition, the context length of LIFT is restricted to the context length of LMs and LIFT/GPT is memory-inefficient. One can combine LIFT with memory-efficient LMs such as LinTransformer [92] to address this issue. Besides, our works open some interesting questions for future works. First, do LMs and LIFT/GPT have behaviors similar to ensemble methods or decision tree since they have similar decision boundaries? Secondly, are LMs universal models that can adapt well to any modalities and domains? Lastly, can LIFT/GPTs adapt better for regression tasks using more sophisticated encoding schemes for numeric features?

**Social impacts.** Future research should also investigate potential fairness issues of applying LIFT. Based on large language models, LIFT might have embedded bias targeting certain social groups. Especially when feature names are included in the training prompts, the models may be more sensitive to social biases and thus might make unfair and harmful predictions. We leave measuring the embedded bias in LIFT as one of the interesting future directions.

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
