# Appendix

Due to the space limit, many interesting findings, discussions, and further details are included in Appendix. We start by discussing the detailed motivation of our work in (A) and introducing related work in (B). We then describe our datasets in (C.1) and the implementation of all models in (C.2). For the experimental results, we provide detailed and extended results for all findings presented in the main paper in (D), with more visualizations, score tables, and additional evaluations on more functions, datasets, and properties. In (E), we present additional interesting findings and visualizations which have not been discussed so far in the main paper. Lastly, we provide a more profound discussion in (F).

# A   Further Motivations of LIFT

While we mainly emphasized the "no-code-ML" property of LIFT, indeed, it has a lot of potentials to be more useful and powerful than many of the current ML models. Particularly, LIFT can bring an entirely novel approach to enable (i) explainability and (ii) updatability via information retrieval.

## A.1   Explainability

Most ML models cannot interpret their predictions. While specific algorithms are developed to enable such models' explainability, their efficacy is still in question. On the other hand, LMs have been shown to be able to explain their predictions [93, 94]. This ability has been used in many fields such as human moral judgements [95] and mathematical reasoning [96]. Therefore, LIFT, based on a large pretrained LM, can be made to explain its prediction using its reasoning capabilities.

Consider the German-credit dataset [97], which aims to predict whether the bank should approve or decline loan applications. After asking GPT-3 if one should approve/decline a loan application via LIFT, one can also ask GPT-3 to explain its own prediction result. This can be implemented by making two consecutive inference calls, as shown in Table 13. Here, we provide five different responses generated with different random seeds.

Note that these explanations are generated without seeing *any* reasoning examples, *i.e.*, these are solely based on the zero-shot reasoning capability of language models. While these explanations are not even close to perfect, they are still reasonable. Improving the few-/zero-shot reasoning capabilities of language models is a fast-growing research field, so incorporating new techniques can further improve the explainability of LIFT.

## A.2   Updatability via Information Retrieval

Another drawback of current ML models is that it is difficult to "update" models when a distribution shift occurs. Handling these distribution shifts in a compute-efficient manner has recently become one of the most active research areas in the field. Recently, augmenting LMs [98, 99, 100, 101] with a retrieval mechanism has shown to be an efficient way of updating LMs. With such a retrieval mechanism equipped, LMs can be efficiently updated as one can simply update its associated database, or even connect LMs to the Internet [101].

While we only used LMs not equipped with a retrieval mechanism in this paper, it is straightforward to apply our framework to other LMs that can retrieve information from databases or the Internet, *i.e.*, LIFT can support a compute-efficient update mechanism.

# B   Detailed Related Works

This section provides detailed related work.

**Pretraining and adapting language models (LMs).**    Our work uses modern large LMs, which promoted significant advances in the field of natural language processing (NLP) [4]. Most popular LMs use transformer architectures [12, 13] as the backbone, from early models like BERT [102] built on Transformer encoders to GPT variants [103, 21] built on Transformer decoders. Multiple modern large LMs have been proposed, including RoBERTa [104], ALBERT [105], XLNet [106], and the latest models with billions or trillions of parameters, such as GPT-3 [107], Switch-Transformers [108], and PALM [109].

LMs are trained to encode a large amount of linguistic knowledge from multiple sources in their contextual representations [110], which are helpful and can be easily adapted to various other tasks. Thus, starting with BERT [102], it has become a standard practice to pretrain and then fine-tunes a large LM for plenty of downstream tasks in lieu of training a model from scratch for a specific task [102, 19, 4, 104, 111]. This technique dramatically impacts a wide range of NLP tasks, such as language modeling [19, 109], question answering systems [112, 113], text summarization [114], neural machine translation [115], and reasoning [116, 117].

However, the excellent performance of fine-tuned LMs without architecture changes has been mainly limited to NLP tasks so far. This work, instead, investigates whether we can leverage LM fine-tuning

Table 13: **An example illustrating the reasoning capability of large pretrained language models.** We convert a data sample from German-credit dataset [97] into text and use GPT-3 to predict whether the bank should approve or decline the individual's loan application. After receiving the first response of "accept", we add "The bank will accept this loan application. Let's think logically. This is because" to the original prompt and use it as the new input of GPT-3. The five different responses generated by GPT-3 with different random seeds demonstrate the reasoning ability of language models. Note that these explanations are generated without seeing any reasoning examples, *i.e.*, these are solely based on the zero-shot reasoning capability of language models.

| | **Prompt or Generated Text** |
|---|---|
| Raw input | Checking Account: less than 200 DM; Loan Duration: 48 months; Credit History: no credits/paid back duly; Loan Purpose: education; Number of applying for credits: 18424; Savings Account: less than 100 DM; Employment Time: 1 4 years; Installment Rate: 1%; Sex: Female; Other Debtors: No; Property: building society savings agreements/life insurance; Age: 32; Housing: self-owned house; Job: highly qualified employee/ self-employed; Foreigner: no |
| First prompt (soliciting predic-tion) | A 32-year-old female is applying for a loan of 18424 credits for 48 months for other purposes. She has a checking account with less than 200 Deutsche Mark and a savings account with less than 100 Deutsche Mark. She had no credits or paid back all credits duly. The installment rate of this applicant in percentage is 1%. She is employed for more than one year but less than four years and is recognized as a highly qualified employee or is self-employed. She has lived in a self-owned house. She owns some building society savings agreements or life insurance. Given such information, will the bank accept or decline this loan application? |
| First re-sponse | accept |
| Second prompt (soliciting explana-tion) | A 32-year-old female is applying for a loan of 18424 credits for 48 months for other purposes. She has a checking account with less than 200 Deutsche Mark and a savings account with less than 100 Deutsche Mark. She had no credits or paid back all credits duly. The installment rate of this applicant in percentage is 1%. She is employed for more than one year but less than four years and is recognized as a highly qualified employee or is self-employed. She has lived in a self-owned house. She owns some building society savings agreements or life insurance. Given such information, will the bank accept or decline this loan application? **The bank will accept this loan application. Let's think logically. This is because** |
| Second response (random seed 1) | the applicant is employed and is qualified in a high level. Since she is employed, she is able to make all the payments. Furthermore, she has a self-owned house and owns some building society savings agreements or life insurance. She has less than 2000 Deutsche Mark in checking and savings account. She had no credits or paid back all credits duly. She is able to pay the loans. All of these are positive factors to this loan. The bank will accept this loan. |
| Second response (random seed 2) | the applicant has a high income. She has lived in a self-owned house. She has some savings agreements or life insurance. She is a highly qualified employee. She has no negative credits. All of these facts show that she is not in danger of not being able to pay back the loan. Therefore, the bank will accept this loan application. |
| Second response (random seed 3) | the applicant has a checking account with less than 200 Deutsche Mark and savings account with less than 100 Deutsche Mark. Hence she is a low-risk customer. Further-more, she has no credits or paid back all credits duly. And the installment rate of this applicant is 1%. She is employed for less than four years and is recognized as a highly qualified employee. She has lived in a self-owned house. She owns all the building society savings agreements or life insurance. Hence she is considered a good customer. So the bank will accept her loan application |
| Second response (random seed 4) | the applicant is employed with a high salary and is economically stable. She is not likely to file for bankruptcy. She can pay back the loan. All in all, we can see that the bank will accept this loan application. |
| Second response (random seed 5) | the bank will accept all applications with a very low risk of default. There is a very low probability that she will be unemployed in the near future. She can afford to pay off this loan. Furthermore, this applicant has a high annual income. |

for non-language tasks across different modalities. Our work is highly motivated by Frozen Pretrained Transformer (FPT) [18], which directly adapts GPT-2 [21] pretrained on language tasks and textual data to other modalities. FPT freezes most pretrained parameters except the layer normalization layers and adds input and output layers for fine-tuning. The authors empirically show that GPT-2 can be efficiently fine-tuned for different modalities and domains, including vision and numeric computation. Nevertheless, FPT requires changes in the architecture and objective function to adapt to different data representations, while our method LIFT does not. Furthermore, we mainly focus on basic machine learning tasks, such as function approximation or tabular data classification.

Several works have attempted to extend the existing LMs to handle different types of input data, such as images [75, 76], audio [77], tabular data [78], and knowledge bases [79] by updating the pretraining phase with these data and their corresponding tasks. For instance, XGPT [118] takes images as the input and uses the image captioning task as the primary task in the pretraining stage for GPT to generate images' captions. Similarly, multiple works utilize pretrained LMs for generating text descriptions of images or videos in image captioning (VisualGPT [119]) and video captioning (VideoBERT [120], Unified VLP [121], UniVL [122]). SpeechBERT [77] also integrates LMs for speech recognition in the weakly-supervised setting to reduce the need for scarce supervised data. LMs can also adapt to numeric tasks [123] or other domains such as protein folding [124] or symbolic math solvers [125]. Recent works [126, 127, 78] pretrain LMs with large tabular datasets, improving the question answering systems by reasoning over the tables. Compared with these existing works, our work is unique in that it is based on GPT language models trained *only* on textual data.

**Analyzing the adaptability of LMs.**   Similar to our work, recent works [20, 81] have made efforts to understand and quantify the feasibility and limitations of the adaptability of large LMs for upstream performance and downstream tasks. For instance, the recent work [20] built a benchmark of 500 small language tasks for testing the adaptability of LMs, observing that the LMs [128] can adapt well to an extensive range of complex tasks rather than just memorizing the learned patterns. BIG-bench [82] is recently introduced as a new benchmark for quantifying the capacity of LMs, consisting of more than 200 tasks on a diverse set of topics. Another relevant work [83] attempts to understand the effect of LM pretraining by studying how the transformer architecture, the backbone of LMs, succeeds at a designed synthetic task. Similar efforts in this line of work are to analyze the behaviors, representations, and inductive bias of pretrained LMs [129, 130, 131] or investigate different aspects of LMs [132, 133, 134]. For instance, a recent work [132], in the investigation of the difficulty of numeracy in LMs observes that transformer-based language models do not work well on complex numeric tasks and are sensitive to different formats of numeracy. Note that these existing works focus primarily on downstream language tasks, while we focus on adapting LMs on non-language tasks without any modification of the loss or architecture.

**Methods for adapting LMs.**   The most common method for adapting LMs is *fine-tuning* [66] which aims to slightly adjust pretrained parameters for learning the downstream tasks [135, 136]. Fine-tuning can involve simple architecture modifications, such as adding linear layers [67, 68] or freezing parts of the network [18, 69, 70]. Fine-tuning can be improved with advanced techniques, such as multi-stage methods [137], intermediate fine-tuning [138, 139, 140], or self-supervised training [141]. The recent progress in fine-tuning LMs focuses on the *parameter-efficient* techniques for minimizing the number of fine-tunable parameters, including adapter-based fine-tuning [25, 26, 27] that adds and trains small residual blocks between transformer layers, freezing-based fine-tuning [71, 18, 72] that freezes most of the pretrained parameters and fine-tunes only tiny parts of the networks, and distillation-based fine-tuning [73]. In this line of work, LoRA [24] further reduces the number of trainable parameters in large LMs by approximating the weight updates using low-rank matrices without changing pretrained parameters. LoRA is used as the fine-tuning method for GPT-J in our LIFT/GPT-J framework. To directly adopt these fine-tuning methods used in LMs for non-language tasks, it is common practice to modify the input/output layers and the loss functions. However, these modifications might lead to undesired behaviors like catastrophic forgetting [66, 74]. On the other hand, our method LIFT uses the language interface for fine-tuning without making any changes to the architecture or the loss function.

In-context few-shot learning paradigm [142, 143, 28, 29, 144] suggests modifying only the inputs of LMs by adding a few examples of the downstream task. A critical part of these methods is reformulating the downstream task samples to the language modeling inputs [47, 59], resulting in multiple efforts in generating [145], searching [48], and properly tuning the prompts [146, 147, 148].

While these methods have shown great effectiveness for multiple NLP downstream tasks, it is unclear how to apply them to downstream tasks of other modalities. On the other hand, our work successfully adapts LMs for non-language tasks, further pushing the application boundaries of large-scale LMs.

**Deep learning for tabular datasets.** While deep neural networks have been successfully applied to various data types, such as images or text, they still face difficulties with a few classification and regression tasks on tabular data [149, 150], one of the most popular data types in practice. This may be due to the heterogeneous nature of tabular data, with their features being sparse, type-mixed, and weaker in correlation than natural image-language data [151, 149]. Multiple deep learning methods and architectures have been proposed for tabular datasets, from making discrete decision trees more differentiable [152, 153], regularizing neural networks' weights [154, 155], to recent attempts using attention-based modules [156, 157, 158]. Though these transformer-based models are the closest works to us in this line of work, their works focus on designing and improving architecture designs for specifically learning the tabular data rather than adapting the LMs. To the best of our knowledge, we are the first to thoroughly study large LM adaptation for tabular learning without architecture changes. Our work shows promising results of LMs in closing the gap to the best-performing methods, including tree-based ensemble algorithms (Random Forest [159] or XGBoost [160]).

**General-purpose models (generalist models).** A primary goal of our work is to push the limit of the existing generalist language models (*e.g.,* GPT-3 [19]) to other modalities and domains, supporting the idea of building a domain-and-modality agnostic generalist model. Early works [161, 14, 162] explored this idea by developing and training multi-task and multi-modal models on a wide range of diverse tasks to obtain better generalization and adaptation. The development of large-scale LMs has significantly contributed to the area of generalist models for languages [19, 84, 3], vision [85], visual language [86, 87, 88], and control problems [89]. These generalist models are usually trained with the scale on an extensive range of corpora, probably containing multiple modalities and domains. In this line of work, a general-purpose architecture [80] has also been studied to handle different input and output data types. Although LIFT primarily focuses on the LMs, it can be applied to other generalist models with LM-like architectures, such as GATO [89]. Furthermore, it is worth noticing that our work shares the general goal with automated machine learning (AutoML) [90, 91] in improving the usability of machine learning. AutoML automates the standard machine learning pipeline for model selection and hyperparameter tuning from a set of existing algorithms. At the same time, LIFT uses only a single pretrained LM for solving all tasks.

## C  Experimental Setup

### C.1  Datasets

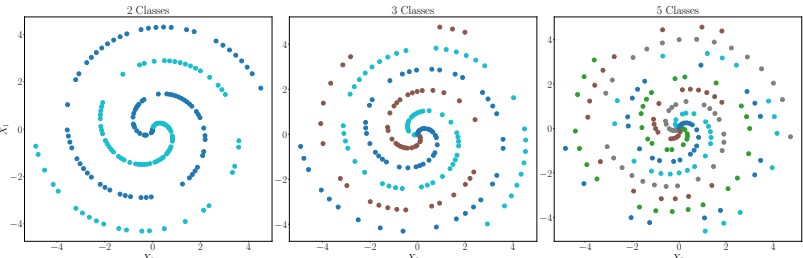

Figure 14: **Rolls dataset of 2, 3, 5 classes with 300 samples per dataset (all classes are balanced).**

**Classification datasets.** Table 16 summarizes the datasets used for classification tasks. We use two additional types of synthetic datasets: **neural-net-based datasets** used for understanding the inductive biases of algorithms (in Sec. 3.4 and Appendix D.1.3) and **Gaussian pretext datasets** for the two-stage fine-tuning experiments (in Sec. 5.1). The *neural-net-based datasets* are generated as follows. For binary classification, we train a 2-layer neural network with `tanh` activation functions using the *Rolls* dataset shown in the leftmost figure in Fig. 14, and take six snapshots of the decision boundary of the trained neural network shown in Fig. 15; we took snapshots at training epochs 10, 40, 80, 210, 320 and 490. Then, for each snapshot, we define a synthetic dataset (what we call a neural-net-based dataset) having labels as the neural network's prediction for randomly chosen 2000 samples. For 3-class and 5-class classifications, we also use 2000 samples to train a 2-layer neural

network using the *Rolls* dataset shown as the second and the third figure in Fig. 14. The decision boundaries of networks trained on more epochs are visually more complex. Hence, the corresponding classification tasks are becoming more complicated, from the left column to the right column in Fig. 15. In the manuscript, we tested on three out of six datasets, obtained by snapshots at 10, 80, and 490 epochs, respectively. Given a target dataset of $n$ classes and $d$ features, *Gaussian pretext datasets* are generated as follows: using `scikit-learn`[2], we randomly generate datasets of $n$ clusters, where each cluster has 100 normally distributed samples in the $d$-dimensional space.

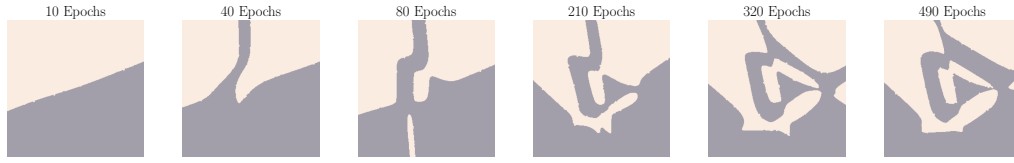

Figure 15: **Neural-net-based datasets.** Given `Rolls` dataset in Fig. 14, we train a 2-layer neural network for 10, 40, 80, 210, 320, 490 epochs, and get six decision boundaries at each column. We define six neural-net-based datasets from here: each decision boundary is used as a labeling function of each neural-net-based dataset. In the main manuscript, we used three out of six datasets, obtained by snapshots at 10, 80, and 490 epochs.

(**Image datasets**) For MNIST-based datasets, as the context length of LMs is limited, we (center) scale and crop the size of images from $28 \times 28$ into $18 \times 18$ and use the integral format of pixel values $[0, 255]$. We convert each image into a sequence of pixels in the order of left to right and top to bottom. Each pixel sequence is then converted into the sentence input as of our language-interfaced framework.

**Regression datasets.** We test LIFT/GPT on regression problems for both synthetic/real datasets. (**Synthetic datasets**) To assess the regression performance of LIFT in different datasets, we generate synthetic datasets based on six different functions types, including smooth functions, non-smooth functions, and non-continuous functions:

1. Linear functions $y = f(\boldsymbol{x}) = \boldsymbol{x}^\top \mathbf{1}/p$
2. Quadratic functions $y = f(\boldsymbol{x}) = \boldsymbol{x}^T \boldsymbol{I} \boldsymbol{x}/p$, where $\boldsymbol{I}$ is the identity matrix
3. Continuous exponential function $y = f(\boldsymbol{x}) = \sum_{i=1}^{p} e^{0.2x_i}/p$
4. Cosine functions $y = f(\boldsymbol{x}) = \sum_{i=1}^{p} \cos(0.5\pi x_i)/p$
5. (Non-smooth) $\ell 1$-norm function $y = f(\boldsymbol{x}) = \|\boldsymbol{x}\|_1 /p$
6. (Non-continuous) Piecewise linear function

$$f(\mathbf{x}) = \frac{1}{p}\sum_{i=1}^{p} \tilde{f}(x_i) := \left\{ \begin{array}{ll} x_i - 1 & -10 \leq x_i < -3, \\ 0 & -3 \leq x_i < 3, \\ x_i + 1 & 3 \leq x_i \leq 10. \end{array} \right\}.$$

We let $\mathrm{x}_i \sim \mathrm{Unif}(-10, 10)$ for each coordinate $i$, and the noise level $\sigma = 0.1$ by default. We normalize all the functions above for fair comparison among different functions so that $y \in [-9, 9]$ when $\boldsymbol{x} \in [-10, 10]^p$. In particular, to assess whether LIFT is better at dealing with positive numbers or integers, we generate additional datasets by further manipulating the $(\mathbf{x}, \mathbf{y})$ distribution of linear and piecewise functions. For datasets with real numbers, we generate the 1D dataset $\mathrm{x} \sim \mathrm{Unif}(-150, -150)$. For datasets with only positive numbers, we generate the dataset $\mathrm{x}_i \sim \mathrm{Unif}(0, 300)$. To generate the datasets with all integer prompts, we round down all the features to integers. For visualization, in addition to the training, validation, and test datasets, we generated grid datasets. Unless otherwise stated, we generate uniformly spaced 200 samples for 1D visualizations and 2,500 samples for 2D visualizations, with each coordinate $\mathrm{x}_i \in [-10, 10]$ for $i = 1, \ldots, p$. To visualize the extrapolation performance, we let the $\mathrm{x}_i \in [-15, 15]$.

(**Real datasets**) We consider four real datasets: Medical Insurance dataset [41] with 1,338 samples and 6 features, Combined Cycle Power Plant (CCPP) dataset [42] with 9,568 samples and 4 features, Servo [43] dataset with 167 samples and 4 features, and Student [44] dataset with 649 samples and 33

---

[2]https://scikit-learn.org/stable/modules/generated/sklearn.datasets.make_classification.html

Table 16: **Classification datasets.** We have three non-language types of data: synthetic data, real tabular data, and vision data. We use five synthetic datasets. For the real tabular data, we select datasets from OpenML with a wide range of number of features, types of features, number of classes, and number of training samples. We use MNIST, Fashion MNIST, and their permuted variants for the vision datasets.

| Data Type | Dataset | ID | Abbreviation | No. Features | No. Classes | No. Instances | Note |
|---|---|---|---|---|---|---|---|
| Synthetic | 9Gaussians | 1 | - | 2 | 9 | 2000 | - |
| | Blobs | 2 | - | 2 | 4 | 2000 | - |
| | Circle | 3 | - | 2 | 2 | 2000 | non-linear boundary |
| | TwoCircles | 6 | - | 2 | 2 | 2000 | non-linear boundary |
| | Moons | 4 | - | 2 | 2 | 2000 | - |
| Tabular (OpenML) | wholesale-customers | 1511 | Customers | 8 | 2 | 440 | Imbalance |
| | pollution | 882 | Pollution | 15 | 2 | 60 | 1 symbolic feature |
| | spambase | 44 | Spambase | 57 | 2 | 4601 | 1 symbolic feature |
| | hill-valley | 1479 | Hill-Valley | 100 | 2 | 1212 | 1 symbolic feature |
| | tae | 48 | TAE | 5 | 3 | 151 | Categorical data |
| | cmc | 23 | CMC | 9 | 3 | 1473 | Meaningful feature Names |
| | wine | 187 | Wine | 13 | 3 | 178 | Integral features |
| | vehicle | 54 | Vehicle | 18 | 4 | 846 | Meaningful feature Names |
| | LED-display-domain-7digit | 40496 | LED | 7 | 10 | 500 | 1 symbolic feature |
| | optdigits | 28 | OPT | 64 | 10 | 5620 | 1 symbolic feature |
| | mfeat-factors | 12 | Mfeat | 216 | 10 | 2000 | 1 symbolic feature |
| | pollen | 871 | Pollen | 5 | 2 | 3848 | - |
| | climate-model-simulation-crashes | 1467 | Climate | 20 | 2 | 540 | |
| | one-hundred-plants-margin | 1491 | Margin | 64 | 100 | 1600 | 1 symbolic feature |
| | one-hundred-plants-shape | 1492 | Shape | 64 | 100 | 1600 | 1 symbolic feature |
| | one-hundred-plants-texture | 1493 | Texture | 64 | 100 | 1599 | 1 symbolic feature |
| | breast-cancer | 13 | Breast | 9 | 2 | 286 | - |
| | iris | 61 | Iris | 4 | 3 | 150 | - |
| | visualizing_hamster | 893 | Hamster | 5 | 2 | 73 | - |
| | PizzaCutter3 | 1444 | Pizza | 37 | 2 | 1043 | - |
| Vision | MNIST | - | - | 784 | 10 | 70k | - |
| | Permuted MNIST | - | P-MNIST | 784 | 10 | 70k | - |
| | Fashion MNIST | - | FMNIST | 784 | 10 | 70k | - |
| | Permuted Fashion MNIST | - | P-FMNIST | 784 | 10 | 70k | - |

features. The Medical Insurance and Student datasets contain feature names that can be interpreted using common knowledge (see feature lists in Table 35), while CCPP and Servo do not.

## C.2 LIFT and Baseline Implementation

This section provides details of our models and implementation. We describe our pretrained language models and the baseline implementations (C.2.1), the computing resources used for running experiments (C.2.2), and how to fine-tune and select the hyperparameters (C.2.3).

### C.2.1 Pretrained Language Models and Baselines

**Pretrained language models.** Our main results are with two pretrained language models: GPT-J [31] and GPT-3 [19]. We mainly focus on GPT-J for the reproducibility purpose and provide additional results on GPT-3 as reference. For GPT-J, we use a quantized version[3] of 6 billion parameters with 8-bit weights. For GPT-3, we use the GPT-3 OpenAI API[4] and employ the `Ada` version by default. In Sec. E.2, we compare two previously mentioned models with three bigger versions of GPT-3 (`Baggage`, `Curie`, `Davinci`). The largest one is `Davinci`-GPT-3 containing approximately 175 billions of parameters.

Since GPTs can output any language token, the output might not be appropriate for the desired task. For example, GPT might output non-numerical words for regression task approximating a function $f : \mathbb{R}^n \to \mathbb{R}$. In such a case, we categorize this as *invalid output*.

**Baselines.** For XGBoost, we use the open-source `XGBoost` module[5]. For other baselines, we implement them using `scikit-learn` module[6].

---

[3]https://huggingface.co/hivemind/gpt-j-6B-8bit

[4]https://openai.com/api/

[5]https://xgboost.readthedocs.io/

[6]https://scikit-learn.org/

### C.2.2 Computing Resources

For experiments on GPT-J, we use GPU computing from two 24GB-RTX3090 GPUs and AWS EC2 instances[7] (p3.8xlarge, p3.2xlarge). For other models, we run experiments on CPU instances.

### C.2.3 Hyperparameter Selection

In fine-tuning GPT-J, we use Adam-8bit optimizer implementation[8] with weight decay of $0.01$ and 6 warm-up steps. The learning rate is chosen from 1e-4 and 2e-4 for the synthetic/OpenML datasets, and 1e-5 for the vision datasets. We use a linear learning scheduler for the optimizer. For classification, the batch size depends on the number of features of the datasets. We set the batch size to be 128, 32, 16, and 2 for datasets with the number of features being no greater than 2, between 2 and 6, between 6 and 20, and greater than 20, respectively. For regression, we set batch size as 4 by default and reduce it to 1 to avoid the memory issue when the number of features increases. For GPT-3, we use the API provided by OpenAI to perform black-box GPT-3 fine-tuning with the default setting. Our implementation is with PyTorch framework[9].

We perform hyperparameter selection based on validation results for all methods for a fair comparison. The hyperparameter tuning scheme for all methods is detailed as follows:

**Classification methods.**

- LIFT/GPT-J: number of epochs $\in \{5, 10, 15\}$
- LIFT/GPT-3: learning rate multiplier $\in \{0.05, 0.1, 0.2\}$
- Random Forest (*RF*): maximum depth $\in \{3, 5, 10\}$, minimum number of samples required to split an internal node $\in \{2, 5, 10\}$
- Decision Tree (*DT*): maximum depth of the tree $\in \{3, 5, 20\}$ and criterion $\in \{$Gini impurity, Shannon information gain$\}$
- Support Vector Machine (*SVM*): kernel $\in \{$linear kernel, radial basis function$\}$, regularization parameter $\in \{1, 10, 100\}$
- Multilayer Perceptron (*MLP*): initial learning rate $\in \{0.001, 0.01, 0.1\}$
- Logistic regression (*LogReg*): inverse of regularization strength $\in \{1, 10, 100\}$
- K-Nearest Neighbor (*KNN*): number of neighbors to use $\in \{1, 3, 5\}$, power parameter for the Minkowski metric $\in \{1, 2\}$
- XGBoost (*XG*): maximum depth $\in \{3, 5, 10\}$

**Regression methods**

- LIFT/GPT-J: number of epochs $\in \{2, 6, 10\}$
- LIFT/GPT-3: learning rate multiplier $\in \{0.05, 0.1, 0.2\}$
- Polynomial Regression (*PR*): no hyperparameter selection but fixed the degree at 3 since higher-order polynomial regression introduces out-of-memory error, especially for high-dimensional datasets
- K-Nearest Neighbor (*KNN*): number of neighbors $\in \{2, 5, 8\}$
- Kernel Regression (*KR*): Gamma parameter of Radial Basis Kernel $\in \{0.01, 0.1, 1\}$
- Multilayer Perceptron (*MLP*): initial learning rate $\in \{0.0001, 0.001, 0.01\}$
- Gradient Boosting Tree (*GBT*): learning rate $\in \{0.001, 0.01, 0.1\}$
- Random Forest (*RF*): maximum depth $\in \{4, 6\}$
- Gaussian Process (*GP*): the number of optimizer restarts used to find parameters of the kernel that maximize the log marginal likelihood $\in \{5, 10\}$.

Note that we perform model selection based on validation RAE, instead of validation loss.

---

[7]https://aws.amazon.com/ec2/

[8]https://huggingface.co/hivemind/gpt-j-6B-8bit

[9]https://pytorch.org/

# D   Detailed and Extended Results of Experiments in the Main Paper

In this section, we provide the extended version of experimental results presented in the main paper, including the primary findings of LIFT (D.1), the specific learning properties of LIFT (D.2), and improving techniques for LIFT (D.3).

## D.1   Results for Basic Findings of LIFT (Section 3)

### D.1.1   How Well Does LIFT Perform on Standard ML Tasks?

Table 17: **Accuracies (↑) on various classification datasets.** We evaluate LIFT/GPTs on different classification datasets: 2D synthetic datasets, tabular datasets in OpenML [36] , and image datasets, varying number of features ($p$) and number of classes ($c$). Overall, LIFT/GPTs perform comparably well across all tasks. LIFT/GPTs can be adapted well to non-linear datasets (circles, two circles), beyond the capacity of logistic regression. On the OpenML data, LIFT/GPTs achieves competitive performances with the best methods, such as XGBoost or RBF-SVM. The performance of LIFT/GPTs degrades as the number of classes is large, *e.g.*, when the number of classes $c$=100. On the vision data, LIFT/GPTs perform comparably well, achieving highly competitive accuracies on both MNIST and Fashion MNIST. We note that the classes of MNIST are not fully balanced. Thus MCC gets 11.35% instead of 10% as MCC returns the optimal class learned from the training dataset.

| Type | Dataset (ID) | p/c | MCC | LogReg | KNN | DT | MLP | RBF-SVM | RF | XG | LIFT/GPT-J | LIFT/GPT-3 |
|---|---|---|---|---|---|---|---|---|---|---|---|---|
| Synthetic | circles (3) | 2 / 2 | 50.00 | 48.58±1.94 | 81.25±0.20 | 77.42±0.24 | 82.00±0.54 | **83.08±0.59** | 82.42±1.33 | 81.42±0.31 | 79.95±1.53 | 81.17±0.42 |
| | two circles (6) | 2 / 2 | 50.00 | 49.83±4.18 | **81.83±0.62** | 75.50±0.20 | 68.42±3.86 | 80.00±0.54 | 76.08±0.59 | 79.25±0.35 | 75.92±1.65 | 81.42±0.82 |
| | blobs (2) | 2 / 4 | 25.00 | 96.75±0.00 | 95.50±0.20 | 96.08±0.82 | 96.58±0.42 | 96.75±0.00 | **97.17±0.24** | 96.17±0.12 | 96.17±0.59 | 96.67±0.24 |
| | moons (4) | 2 / 4 | 50.00 | 88.58±0.12 | **100.00±0.00** | 99.25±0.41 | 98.75±1.08 | **100.00±0.00** | 99.75±0.00 | 99.83±0.12 | 99.58±0.42 | **100.00±0.00** |
| | 9Clusters (1) | 2 / 9 | 11.25 | 100.00±0.00 | 100.00±0.00 | 100.00±0.00 | 100.00±0.00 | 100.00±0.00 | **100.00±0.00** | **100.00±0.00** | 99.75±0.00 | **100.00±0.00** |
| Tabular (OpenML) | Customers (1511) | 9 / 2 | 68.18 | 87.12±0.54 | **88.64±0.00** | 85.98±0.53 | 86.36±1.86 | 86.36±0.00 | 85.23±0.00 | 85.23±0.00 | 85.23±1.61 | 84.85±1.42 |
| | Pollution (882) | 16 / 2 | 50.00 | 58.33±11.79 | 66.67±6.81 | **77.78±3.93** | 66.67±0.00 | 58.33±6.81 | **77.78±3.93** | 63.89±7.86 | 63.89±3.93 | 63.89±7.86 |
| | Spambase (44) | 58 / 2 | 60.59 | 93.27±0.00 | 90.77±0.00 | 90.7±0.14 | **94.35±0.00** | 93.70±0.00 | 95.01±0.00 | **95.87±0.00** | 94.03±0.54 | 94.90±0.36 |
| | Hill-Valley (1479) | 101 / 2 | 49.79 | 77.78±0.00 | 56.38±0.00 | 56.38±0.89 | 50.21±0.00 | 68.72±0.00 | 51.44±0.00 | 59.26±0.00 | **100.00±0.20** | **99.73±0.19** |
| | TAE (48) | 6 / 3 | 35.48 | 45.16±4.56 | 60.22±4.02 | 65.59±5.49 | 54.84±2.63 | 53.76±6.63 | **67.74±7.90** | 66.67±8.05 | 61.29±6.97 | 65.59±6.63 |
| | CMC (23) | 10 / 3 | 42.71 | 49.49±0.83 | 50.85±1.91 | 56.72±0.32 | 57.29±0.73 | 56.50±0.97 | 53.45±1.05 | 52.43±0.42 | 49.83±0.28 | **57.74±0.89** |
| | Wine (187) | 14 / 3 | 38.89 | **100.00±0.00** | 96.29±1.31 | 93.52±2.62 | 98.15±2.62 | **100.00±0.00** | **100.00±0.00** | 97.22±0.00 | 93.52±1.31 | 92.59±1.31 |
| | Vehicle (54) | 19 / 4 | 25.88 | 80.39±1.00 | 69.61±0.74 | 63.92±2.37 | 79.21±0.28 | **81.18±0.48** | 75.88±1.27 | 73.14±0.28 | 64.31±2.37 | 70.20±2.73 |
| | LED (40496) | 8 / 10 | 11.00 | 68.67±0.94 | 63.67±6.13 | 66.33±2.87 | **72.00±0.82** | 68.00±0.82 | 64.33±0.94 | 66.00±0.82 | 65.33±0.47 | 69.33±2.05 |
| | OPT (28) | 65 / 10 | 10.14 | 96.53±0.22 | 96.92±0.16 | 89.8±1.09 | 97.36±0.27 | 97.95±0.00 | 97.69±0.14 | 97.48±0.17 | 98.22±0.11 | **98.99±0.30** |
| | Mfeat (12) | 217 / 10 | 10.00 | 97.67±0.12 | 97.67±0.31 | 87.67±1.05 | 96.5±0.35 | **98.83±0.24** | 97.75±0.35 | 96.75±0.00 | 94.17±1.75 | 93.08±0.24 |
| | Margin (1491) | 65 / 100 | 0.94 | 81.35±0.15 | 77.60±0.97 | 43.86±1.21 | 77.71±1.91 | **81.98±0.30** | 76.21±0.29 | 70.21±0.29 | 50.23±1.33 | 59.37±0.92 |
| | Texture (1493) | 65 / 100 | 0.94 | 81.67±0.97 | 80.62±0.76 | 46.88±1.93 | 76.88±2.44 | **83.44±0.89** | 73.12±0.76 | 70.73±1.41 | 50.32±2.18 | 67.50±1.42 |
| Images | MNIST | | 11.35 | 91.95±0.69 | 96.71±0.11 | 87.42±0.64 | 97.30±0.16 | 97.70±0.97 | 94.91±0.18 | 97.69±0.04 | 97.01±1.15 | **98.15±0.67** |
| | P-MNIST | 784 / 10 | 11.35 | 92.58±0.04 | 96.74±0.08 | 87.87±0.69 | 97.39±0.14 | **98.06±0.31** | 94.59±0.18 | 97.62±0.09 | 95.80±0.07 | 96.25±0.35 |
| | FMNIST | | 10.00 | 85.59±0.09 | 85.59±0.03 | 80.52±0.40 | 88.86±0.02 | **90.59±0.02** | 85.25±0.13 | 90.19±0.04 | 85.10 ± 0.19 | 90.18±0.12 |
| | P-FMNIST | | 10.00 | 84.95±0.84 | 85.15±0.61 | 79.91±0.93 | 88.86±0.61 | 88.04±1.69 | 84.93±0.59 | **89.93±0.14** | 82.25±0.27 | 88.92±0.71 |

We now provide full results of classification and regression performances with all baselines, including the investigation of the interpolation and extrapolation performance of LIFT for regression tasks.

**Classification.** Table 17 presents the classification performance with other baselines, including KNN, MLP, and Random Forest. We further consider two additional baselines here, which has larger model sizes compared to the baselines we discussed in Sec. 2. TabNet [157] and TabTransformer [156] are deep neural network models based on architectures specifically designed for tabular data. The results are presented in Table 18. We observe that LIFT achieves comparable performance to TabNet and TabTransformer. This further highlight the good performance of LIFT.

**Regression.** Table 19 provides the regression evaluation with all regression baselines on synthetic datasets, and Table 20 provides the results for real datasets. Since experiments with LIFT/GPT-J are conduced on AWS and local server and due to this limitation of memory resources, we fail to run experiments of LIFT/GPT-J on high-dimensional datasets. Therefore, for 50D and 120D synthetic datasets, only results of LIFT/GPT-3 are reported.

We further provide the visualization of regression models. Fig. 21 and 22 visualize the 2D predictions for various functions with 200 and 1000 samples training datasets, respectively. Each coordinate of the training sample is drawn uniformly from $[-10, 10]$. Specifically, the prediction is performed on the interval $[-12, 12]$.

Fig. 23 and Fig. 24 visualize the interpolation and extrapolation of various methods. All methods fail to extrapolate and interpolate well for all functions. It turns out that LIFT is not having good interpolation performance except in the linear regression case. An interesting observation is that LIFT tends to output seen values (from training data) for extrapolation. For example, in Fig. 24b, the outputs of LIFTs for $x \notin [-10, 10]$ (extrapolation) lie in the range of outputs for $x \in [-10, 10]$ (trained data), and similar behaviors are observed for other functions as well.

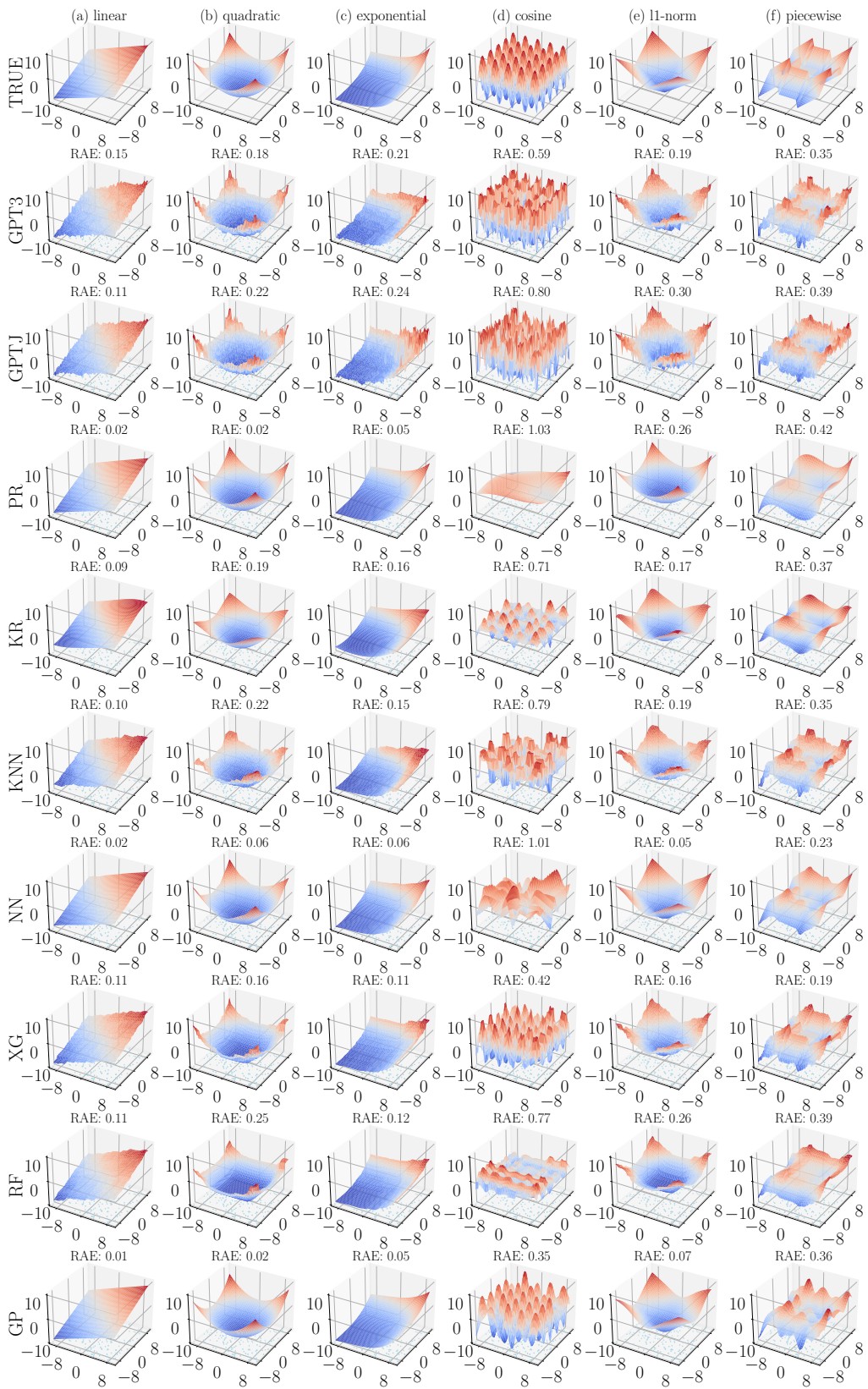

Figure 21: **Performance of LIFT/GPTs and baselines in approximating various functions.** The first row visualizes the true values of the functions, and the second & third rows visualize the predicted values of LIFT/GPTs after fine-tuning for the corresponding regression tasks with **200 training samples**. We compared with other baselines with the same training samples.

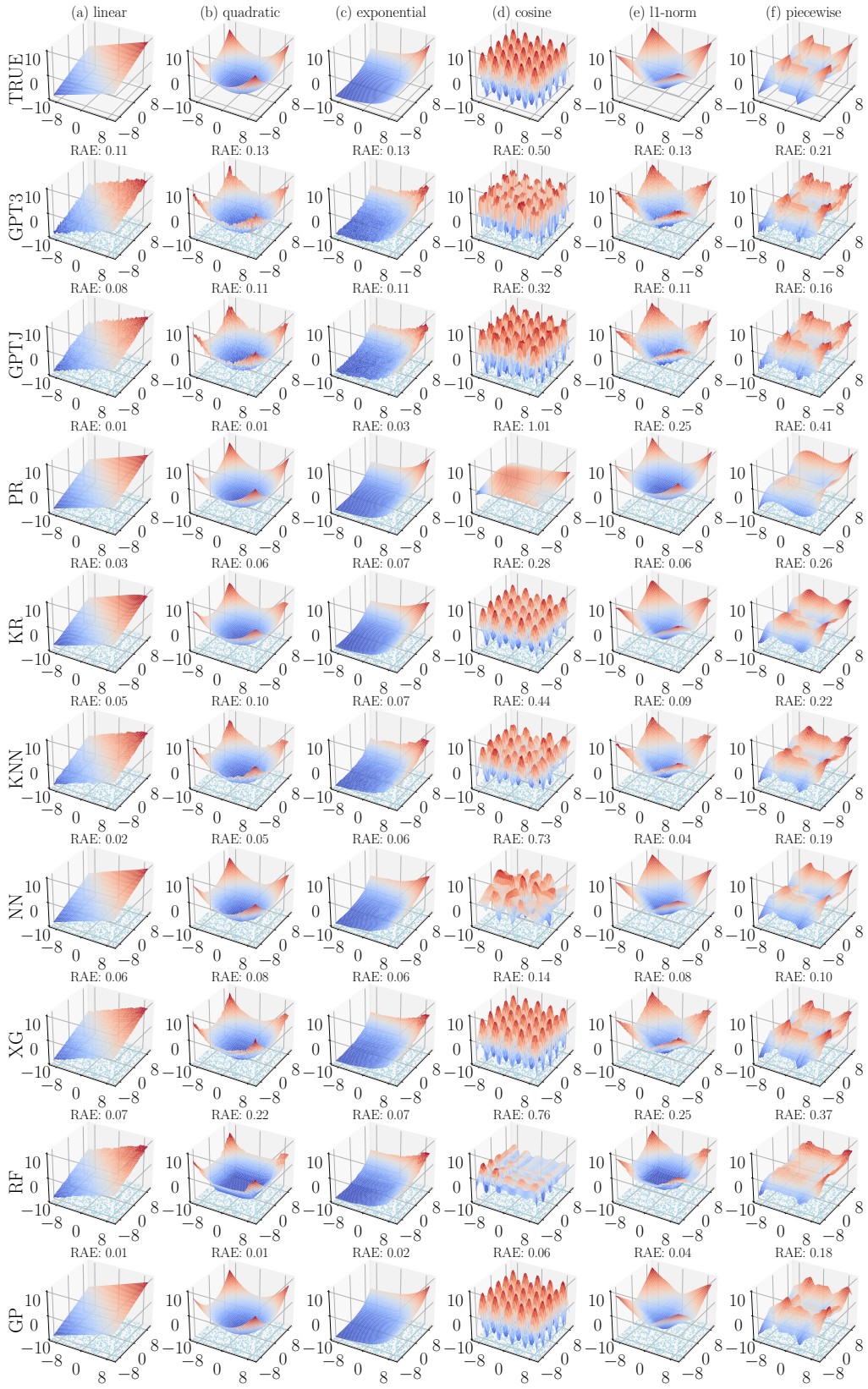

Figure 22: **Performance of LIFT/GPTs in approximating various functions.** The first row visualizes the true values of the functions, and the second & third rows visualize the predicted values of LIFT/GPTs after fine-tuning for the corresponding regression tasks with **1000 training samples**. We compared with other baselines with the same training samples.

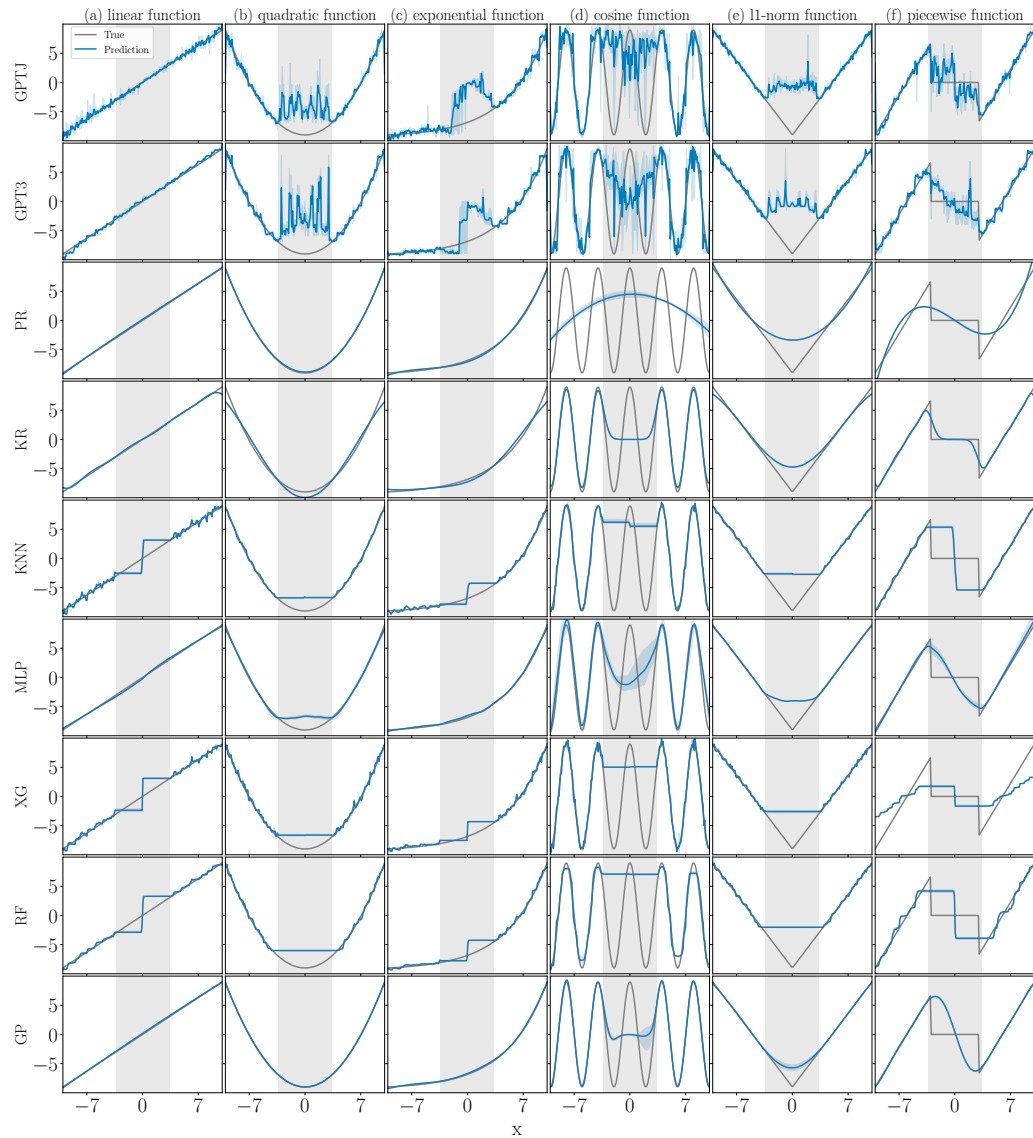

Figure 23: **Interpolation performance on synthetic regression tasks.** Each algorithm is trained with samples in the white background region ($3 \leq |x| \leq 10$), and tested on the interpolation area $|x| \leq 3$. LIFT/GPTs are having worse interpolation performances compared with existing methods.

Table 18: **Comparison of accuracies(↑) between LIFT and deep neural network models designed for tabular datasets.** We consider two baselines here: TabNet [157] and TabTransformer [156]. We observe that LIFT achieves comparable performance to TabNet and TabTransformer, which is more evidence of the good performance of LIFT.

| Dataset (ID) | MCC | LIFT/GPT-3 | LIFT/GPT-J | TabNet | TabTransformer |
|---|---|---|---|---|---|
| Blobs (2) | 25.00 | 96.67± 0.24 | 96.17± 0.59 | 96.75± 0.00 | 50.00± 0.00 |
| Two Circles (6) | 50.00 | 81.42± 0.82 | 75.92± 1.65 | 74.25± 12.39 | 49.25± 1.29 |
| Iris (61) | 33.33 | 97.0± 0.00 | 96.67± 0.00 | 97.78± 1.92 | 72.22± 5.09 |
| Customers (1511) | 68.18 | 84.85± 1.42 | 85.23± 1.61 | 85.22± 3.93 | 87.12± 0.66 |
| Wine (187) | 38.89 | 92.59± 1.31 | 93.52± 1.31 | 94.44± 5.56 | 90.74± 13.70 |
| LED (40496) | 11.0 | 69.33± 2.05 | 65.33± 0.47 | 67.00± 2.46 | 41.00± 12.49 |

Table 19: **Comparison of regression methods in approximating various functions.** The regression performance is measured by RAE(↓), and we tested on six functions with various $p$, the number of features. LIFT can approximate different types of functions in low-dimensional cases ($p = 1, 2$), although it fails to achieve performance comparable to that of strong baselines. We observed that LIFT fails to achieve satisfying regression performance in high-dimensional cases ($p = 50, 100$). *Results of* LIFT*/GPT-J on high-dimensional datasets are not available due to the resource limitation.*

| Dataset | Method | PR | KR | KNN | MLP | GBT | RF | GP | LIFT/GPT-J | LIFT/GPT-3 |
|---|---|---|---|---|---|---|---|---|---|---|
| Linear | $p = 1$ | 0.01 ± 0.0 | 0.05 ± 0.0 | 0.04 ± 0.0 | 0.03 ± 0.0 | 0.05 ± 0.0 | 0.04 ± 0.0 | 0.01 ± 0.0 | 0.08 ± 0.0 | 0.06 ± 0.0 |
| | $p = 2$ | 0.03 ± 0.0 | 0.09 ± 0.0 | 0.12 ± 0.0 | 0.04 ± 0.0 | 0.12 ± 0.0 | 0.12 ± 0.0 | 0.01 ± 0.0 | 0.12 ± 0.0 | 0.19 ± 0.0 |
| | $p = 50$ | 0.71 ± 0.0 | 1.02 ± 0.0 | 0.78 ± 0.0 | 1.85 ± 0.1 | 0.97 ± 0.0 | 0.87 ± 0.0 | 0.13 ± 0.0 | - | 1.18 ± 0.2 |
| | $p = 100$ | 0.95 ± 0.0 | 1.02 ± 0.0 | 0.88 ± 0.0 | 3.02 ± 0.0 | 0.99 ± 0.0 | 0.94 ± 0.0 | 0.64 ± 0.0 | - | 2.14 ± 0.5 |
| Quadratic | $p = 1$ | 0.01 ± 0.0 | 0.05 ± 0.0 | 0.05 ± 0.0 | 0.03 ± 0.0 | 0.06 ± 0.0 | 0.05 ± 0.0 | 0.01 ± 0.0 | 0.11 ± 0.0 | 0.13 ± 0.0 |
| | $p = 2$ | 0.03 ± 0.0 | 0.16 ± 0.0 | 0.17 ± 0.0 | 0.06 ± 0.0 | 0.15 ± 0.0 | 0.25 ± 0.0 | 0.02 ± 0.0 | 0.28 ± 0.1 | 0.22 ± 0.0 |
| | $p = 50$ | 1.12 ± 0.0 | 5.19 ± 0.0 | 1.33 ± 0.0 | 2.28 ± 0.0 | 0.98 ± 0.0 | 0.96 ± 0.0 | 0.69 ± 0.0 | - | 0.99 ± 0.2 |
| | $p = 100$ | 1.02 ± 0.0 | 7.30 ± 0.0 | 1.29 ± 0.0 | 2.89 ± 0.0 | 1.01 ± 0.0 | 0.98 ± 0.0 | 0.89 ± 0.0 | - | 1.06 ± 0.1 |
| Exponential | $p = 1$ | 0.04 ± 0.0 | 0.07 ± 0.0 | 0.05 ± 0.0 | 0.02 ± 0.0 | 0.05 ± 0.0 | 0.04 ± 0.0 | 0.01 ± 0.0 | 0.11 ± 0.0 | 0.09 ± 0.0 |
| | $p = 2$ | 0.04 ± 0.0 | 0.15 ± 0.0 | 0.13 ± 0.0 | 0.07 ± 0.0 | 0.09 ± 0.0 | 0.11 ± 0.0 | 0.04 ± 0.0 | 0.19 ± 0.0 | 0.20 ± 0.0 |
| | $p = 50$ | 0.94 ± 0.0 | 10.23 ± 0.0 | 1.04 ± 0.0 | 3.18 ± 0.2 | 1.05 ± 0.0 | 0.96 ± 0.0 | 0.53 ± 0.0 | - | 1.15 ± 0.0 |
| | $p = 100$ | 0.96 ± 0.0 | 14.12 ± 0.0 | 1.03 ± 0.0 | 4.14 ± 0.0 | 0.97 ± 0.0 | 0.93 ± 0.0 | 0.79 ± 0.0 | - | 1.03 ± 0.0 |
| Cosine | $p = 1$ | 1.05 ± 0.0 | 0.12 ± 0.0 | 0.14 ± 0.0 | 0.38 ± 0.1 | 0.15 ± 0.0 | 0.35 ± 0.0 | 0.04 ± 0.0 | 0.38 ± 0.1 | 0.44 ± 0.1 |
| | $p = 2$ | 1.04 ± 0.0 | 0.74 ± 0.0 | 0.83 ± 0.1 | 1.06 ± 0.0 | 0.41 ± 0.0 | 0.80 ± 0.0 | 0.31 ± 0.0 | 0.82 ± 0.2 | 0.65 ± 0.1 |
| | $p = 50$ | 1.01 ± 0.0 | 1.01 ± 0.0 | 1.00 ± 0.0 | 1.59 ± 0.0 | 1.00 ± 0.0 | 0.99 ± 0.0 | 1.01 ± 0.0 | - | 1.25 ± 0.1 |
| | $p = 100$ | 1.02 ± 0.0 | 1.00 ± 0.0 | 1.09 ± 0.0 | 2.43 ± 0.1 | 1.04 ± 0.0 | 1.06 ± 0.0 | 1.00 ± 0.0 | - | 1.20 ± 0.3 |
| L1norm | $p = 1$ | 0.23 ± 0.0 | 0.06 ± 0.0 | 0.05 ± 0.0 | 0.03 ± 0.0 | 0.06 ± 0.0 | 0.06 ± 0.0 | 0.03 ± 0.0 | 0.10 ± 0.0 | 0.09 ± 0.0 |
| | $p = 2$ | 0.24 ± 0.0 | 0.17 ± 0.0 | 0.19 ± 0.0 | 0.06 ± 0.0 | 0.15 ± 0.0 | 0.29 ± 0.0 | 0.07 ± 0.0 | 0.24 ± 0.0 | 0.20 ± 0.0 |
| | $p = 50$ | 1.09 ± 0.0 | 1.01 ± 0.0 | 1.28 ± 0.0 | 1.97 ± 0.1 | 0.98 ± 0.0 | 0.94 ± 0.0 | 0.96 ± 0.0 | - | 1.12 ± 0.1 |
| | $p = 100$ | 1.01 ± 0.0 | 1.01 ± 0.0 | 1.22 ± 0.0 | 2.80 ± 0.1 | 1.03 ± 0.0 | 1.01 ± 0.0 | 0.99 ± 0.0 | - | 1.27 ± 0.2 |
| Piecewise | $p = 1$ | 0.45 ± 0.0 | 0.17 ± 0.0 | 0.08 ± 0.0 | 0.08 ± 0.0 | 0.06 ± 0.0 | 0.07 ± 0.0 | 0.10 ± 0.0 | 0.15 ± 0.0 | 0.17 ± 0.0 |
| | $p = 2$ | 0.39 ± 0.0 | 0.34 ± 0.0 | 0.33 ± 0.0 | 0.20 ± 0.0 | 0.19 ± 0.0 | 0.38 ± 0.0 | 0.29 ± 0.0 | 0.40 ± 0.1 | 0.40 ± 0.1 |
| | $p = 50$ | 0.93 ± 0.0 | 1.00 ± 0.0 | 0.97 ± 0.0 | 2.11 ± 0.0 | 1.00 ± 0.0 | 0.94 ± 0.0 | 0.93 ± 0.0 | - | 1.35 ± 0.1 |
| | $p = 100$ | 1.01 ± 0.0 | 1.00 ± 0.0 | 1.08 ± 0.0 | 4.20 ± 0.1 | 1.02 ± 0.0 | 1.01 ± 0.0 | 1.01 ± 0.0 | - | 1.11 ± 0.0 |

Table 20: **Comparison of regression methods in real datasets.** The regression performance is measured by RAE(↓). We observe that LIFT/GPT-3 achieves the top 2 regression performance among all the real datasets.

| Dataset | Method | PR | KR | KNN | MLP | GBT | RF | GP | LIFT/GPT-J | LIFT/GPT-3 |
|---|---|---|---|---|---|---|---|---|---|---|
| ccpp | | 0.22 ± 0.00 | 21.60 ± 0.00 | 0.45 ± 0.00 | 0.30 ± 0.00 | 0.17 ± 0.00 | 0.21 ± 0.00 | 0.69 ± 0.00 | 0.24 ± 0.01 | 0.18 ± 0.01 |
| servo | | 0.92 ± 0.00 | 0.95 ± 0.00 | 0.86 ± 0.00 | 0.82 ± 0.00 | 0.25 ± 0.00 | 0.25 ± 0.00 | 1.03 ± 0.00 | 1.17 ± 0.16 | 0.29 ± 0.02 |
| insurance | | 0.48 ± 0.00 | 1.48 ± 0.00 | 1.03 ± 0.00 | 0.44 ± 0.00 | 0.25 ± 0.00 | 0.26 ± 0.00 | 1.30 ± 0.00 | 0.53 ± 0.11 | 0.14 ± 0.05 |
| student | | 0.47 ± 0.00 | 1.56 ± 0.00 | 0.66 ± 0.00 | 0.37 ± 0.00 | 0.39 ± 0.00 | 0.36 ± 0.00 | 0.45 ± 0.00 | 0.36 ± 0.02 | 0.27 ± 0.01 |

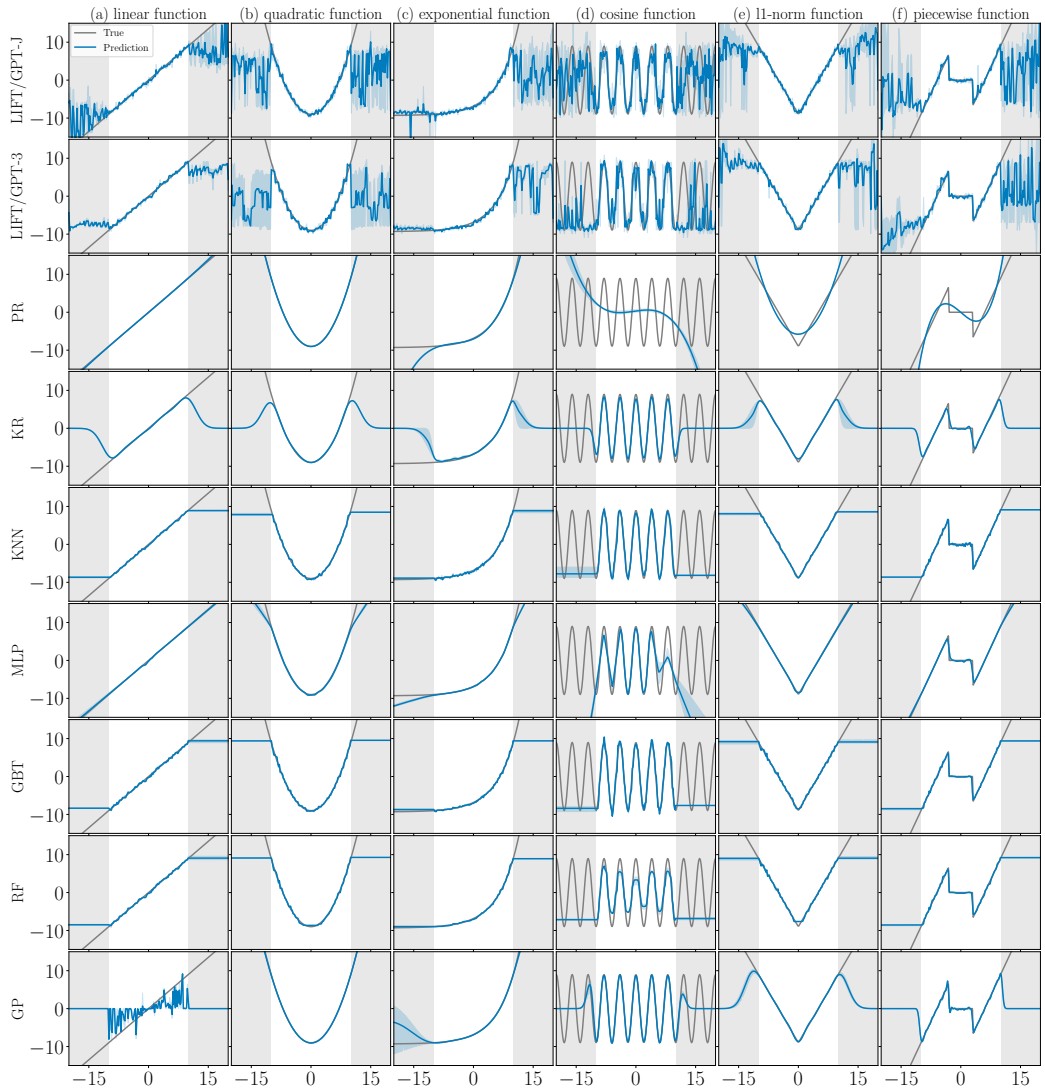

Figure 24: **Comparison of the extrapolation performance of LIFT and various baselines on synthetic regression tasks of approximating six functions** $f$**.** Each algorithm is trained by 200 samples $(x, y)$ where the input $x$ is drawn from interval $[-10, 10]$ and the output is defined as $y = f(x)$. We test the how each algorithm perform regression for $x \notin [-10, 10]$.

### D.1.2    How Many Samples Does LIFT Need?

Fig. 25a and Fig. 25b provide the sample complexity comparisons between evaluated methods in the classification and regression settings.

### D.1.3    Can We Understand the Inductive Biases of Language Models via LIFT?

Continuing from Sec. 3.4, here we provide more experiments with detailed measurements that quantify the similarity between decision boundaries.

**Visualizing decision boundary.**    We construct datasets with various classification complexities to investigate the adaptability of LIFT. In particular, we construct three datasets: a binary classification dataset, a 3-class and a 5-class dataset (shown in the first column of Fig. 26a, Fig. 26b, and Fig. 26c). We call these datasets *neural-net-based synthetic datasets* since we generate them using a 2-layer

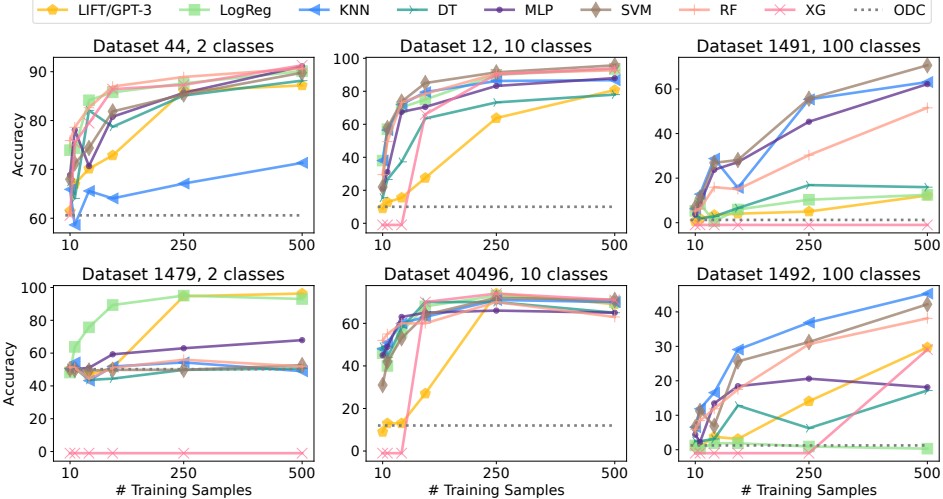

(a) Classification tasks on OpenML tabular datasets. The classification performance is measured in terms of accuracy (↑). *Here, ODC denotes optimal deterministic classifier, which is identical to the majority class classifier (MCC) in the main paper.*

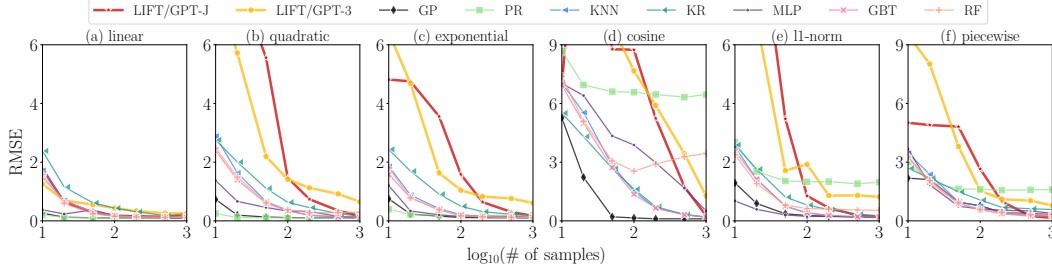

(b) Regression tasks (function approximation). The regression performance is reported in RMSE (↓).

Figure 25: **Sample complexity evaluations on classification and regression tasks.** Each figure presents the comparison of performance evaluated on LIFT/GPTs and baselines varying numbers of training samples (10–500 for classification and 10–1000 for regression). LIFT needs a slightly larger sample complexity to start achieving similar performances to the best baseline methods. For regression tasks, we note that LIFT achieves competitive or even better performance when around 1000s of samples are given, especially for the discontinuous functions, *e.g.*, piecewise function.

neural network. See Fig. 14 and Appendix C.1 for detailed explanations of how we generated these datasets. Note that Fig. 26a is the same as Fig. 6 in Sec. 3.4, which we put for completeness here.

Fig. 26 visualizes the decision boundaries of models trained on the neural-net-based synthetic data. In addition, we also visualize the decision boundaries of models trained on the *label-corrupted* versions of three binary classification datasets, with the corruption probabilities being 5% and 20% (see details in Sec. D.1.4), shown in Fig. 26d and Fig. 26e. Specifically, we consider the binary classification tasks and flip the training data labels with the provided probabilities. Overall, the same observation of Sec. 3.4 also holds for the 3-class and 5-class datasets — both GPT-J and GPT-3 models fine-tuned with LIFT can adapt well to different boundaries. They can capture the rough shapes of the decision boundaries in all three settings.

Besides, when the level of corruption increases, decision trees and XGboost are the most affected baselines. While roughly capturing the boundary, LIFT/GPT-J also shows more noisy predictions. In contrast, LIFT/GPT-3 displays great robustness against the corrupted labels while capturing the correct boundary shapes. Nevertheless, this experiment indicates the different behaviors of LIFTs from the baseline algorithms and their adaptability to different types of decision boundary.

One interesting observation here is that LIFT's decision boundaries are axis-parallel and show a lot of fractals. The axis-parallel boundary looks similar to the boundary of tree-based classifiers, and the

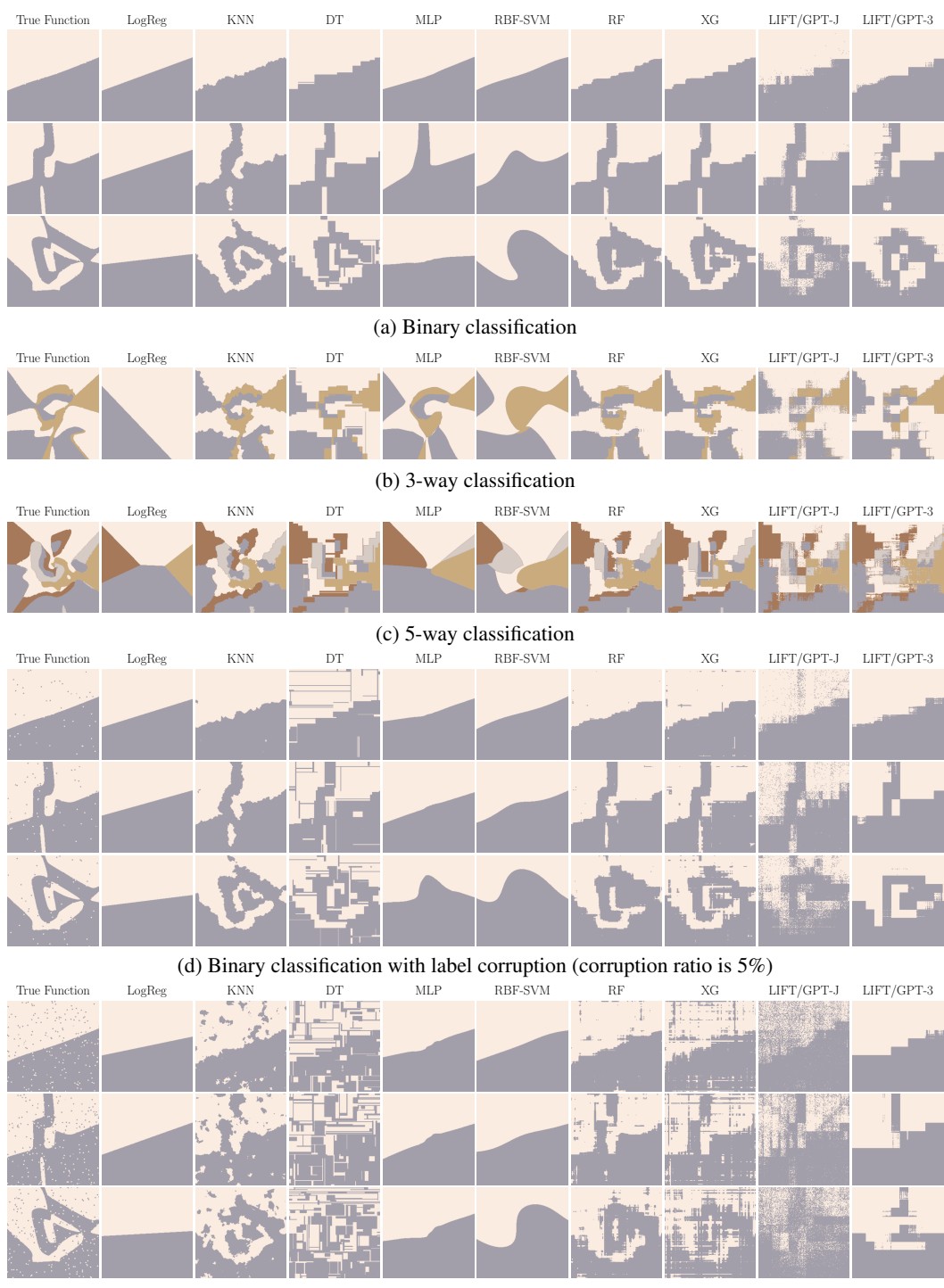

(a) Binary classification

(b) 3-way classification

(c) 5-way classification

(d) Binary classification with label corruption (corruption ratio is 5%)

(e) Binary classification with label corruption (corruption ratio is 20%)

Figure 26: **Classification decision boundary visualizations on neural-net-based synthetic datasets.** The first column of each row shows the decision boundary of the training datasets. In (a), (b), and (c), we visualize the decision boundaries of models trained on datasets with two, three, and five classes. In (d) and (e), we consider the label-corrupted version of the binary-class datasets, with corruption probabilities of 5% and 20%. We find that LIFT/GPTs adapt well and roughly estimate the true decision boundaries. The shapes of LIFT's decision boundary are likely to be axis-parallel and show multiple fractals.

Table 27: **Quantifying the similarity between decision boundaries of LIFT/GPT-3 and those of various baselines.** We use different settings of the baselines, where their hyperparameters are given with the baseline name, and the selected values of hyperparameters are specified in the second line. Each column reports the matching accuracy (↑) between the predictions of LIFT/GPT-3 with those of the baseline. Each score is a percentage similarity of both LIFT/GPT-3 and the baseline classifying a point with the same class. For example, a score of 100 for model A signifies that LIFT/GPT-3 classified all sampled test points in the same manner as model A, regardless of their true dataset accuracy. The last row reports the average matching accuracy. We highlight the highly matched algorithms, namely RBF-SVM, MLP (W=100), and Random Forest (E=100).

| Similarity \ Method | SVM (kernel) | | | LogReg | KNN ($k$) | | | DT (depth $D$) | | MLP (width $W$) | | | XG | RF (# estimators $E$) | | |
|---|---|---|---|---|---|---|---|---|---|---|---|---|---|---|---|---|
| Dataset (ID) | poly | rbf | sigmoid | | K=1 | K=3 | K=5 | D=3 | D=5 | W=10 | W=100 | W=200 | | E=20 | E=50 | E=100 |
| 9clusters (1) | 100.00 | 100.00 | 100.00 | 100.00 | 100.00 | 100.00 | 100.00 | 76.00 | 100.00 | 100.00 | 100.00 | 100.00 | 97.50 | 100.00 | 100.00 | 100.00 |
| blobs (2) | 98.50 | 97.50 | 92.00 | 97.50 | 94.00 | 95.50 | 96.00 | 94.50 | 91.00 | 97.00 | 97.00 | 97.00 | 93.50 | 94.50 | 94.00 | 94.00 |
| circles (3) | 54.00 | 93.50 | 48.00 | 51.00 | 85.50 | 88.50 | 88.50 | 67.00 | 80.00 | 84.00 | 92.50 | 92.50 | 87.50 | 85.00 | 87.50 | 89.00 |
| moons (4) | 90.50 | 97.50 | 74.50 | 87.50 | 99.00 | 99.00 | 99.00 | 91.00 | 98.50 | 92.50 | 98.50 | 98.50 | 97.50 | 95.50 | 94.50 | 96.00 |
| two circles (6) | 63.00 | 62.50 | 48.50 | 62.50 | 59.50 | 58.00 | 59.00 | 57.50 | 59.50 | 60.50 | 64.00 | 65.00 | 58.50 | 64.00 | 63.50 | 65.00 |
| CMC (23) | 59.50 | 61.00 | 68.00 | 63.50 | 50.50 | 52.50 | 51.00 | 65.00 | 72.00 | 62.50 | 53.50 | 53.50 | 63.00 | 56.00 | 55.50 | 59.00 |
| Pollen (871) | 66.00 | 74.50 | 66.00 | 67.50 | 60.00 | 63.00 | 64.50 | 69.50 | 66.00 | 60.00 | 59.50 | 58.00 | 63.50 | 64.50 | 62.50 | 67.00 |
| Climate (1467) | 100.00 | 100.00 | 100.00 | 94.50 | 92.50 | 98.50 | 100.00 | 94.50 | 91.00 | 98.00 | 97.50 | 97.50 | 94.50 | 99.50 | 99.50 | 100.00 |
| LED (40496) | 88.00 | 87.00 | 82.50 | 91.50 | 74.00 | 76.50 | 84.50 | 69.00 | 84.50 | 85.50 | 94.50 | 91.50 | 92.00 | 91.50 | 91.50 | 90.50 |
| Average | 79.94 | **85.94** | 75.5 | 79.5 | 79.4 | 81.28 | 82.50 | 76.00 | 82.50 | 82.22 | **84.11** | 83.72 | 83.06 | 83.39 | 81.17 | **84.5** |

fractal shapes of LIFT's boundaries are similar to the observations on the decision boundaries of some convolution neural networks [49]. However, the main reason why LIFT's decision boundary has such patterns seems to be due to the way it interprets numbers. Since we rely solely on the language interface, there are some artifacts due to the decimal numeral system. For instance, $0.98$ and $0.99$ are only one-character different, but $0.99$ and $1.00$ are three-characters different. We believe that such an artifact is the reason behind axis-parallel decision boundaries and fractal-like patterns.

**Quantifying the similarity of decision boundaries.** To further verify whether LIFTs behave similarly to any standard algorithm, we quantify the similarity between the decision boundaries of LIFT/GPT-3 and those of the baselines. Specifically, the similarity score is the percentage of the exact classification matches between LIFT/GPT-3 and the compared method. We randomly sample two sets of 200 data points from the original dataset for training and evaluation, respectively, and the results of all methods are reported in Table 27. Based on the similarity score, while we observe no similar discernible pattern between LIFT/GPT-3 and the baselines, we find that LIFT/GPT-3 appear to share the most similar behavior pattern to RBF-SVM, random forest (E=100), and MLP (W=100).

### D.1.4 How Robust Is LIFT?

**Robustness to outliers in training data.** Fig. 28 visualizes the outlier robustness results discussed in Sec. 3.5.

**Robustness to label corruption.** We choose a subset of samples and corrupt the label of the chosen samples, using two corruption schemes: (1) *random errors* (randomly select another label with an equal probability for all labels) and (2) *systematic errors* [163] (replace a label with its next label in the target label list, *e.g.*, $0 \rightarrow 1, 1 \rightarrow 2, 2 \rightarrow 0$ for a 3-way classification). As shown in Fig. 29 , LIFT/GPT-3 can perform well under label corruption; it follows the general trend of other baselines, not outperforming or underperforming. Note that LIFT/GPT-3 almost always displays greater robustness than KNN.

Table 30 and Table 31 extend the results reported in Fig. 29. These additional datasets follow a similar trend to what was discussed in Sec. 3.5.

**Robustness to class-imbalance of training data.** We evaluate LIFT on class-imbalanced classification tasks (OpenML datasets `Pizza`, `Climate`, and `Customers` having IDs 1444, 1467, and 1511), shown in Table 32. We use additional metrics: F1, precision, and recall (higher scores indicate better performance), which are considered as better measurements for the imbalanced data than the accuracy. The higher values of the MCC's accuracy imply the higher levels of imbalance in the data

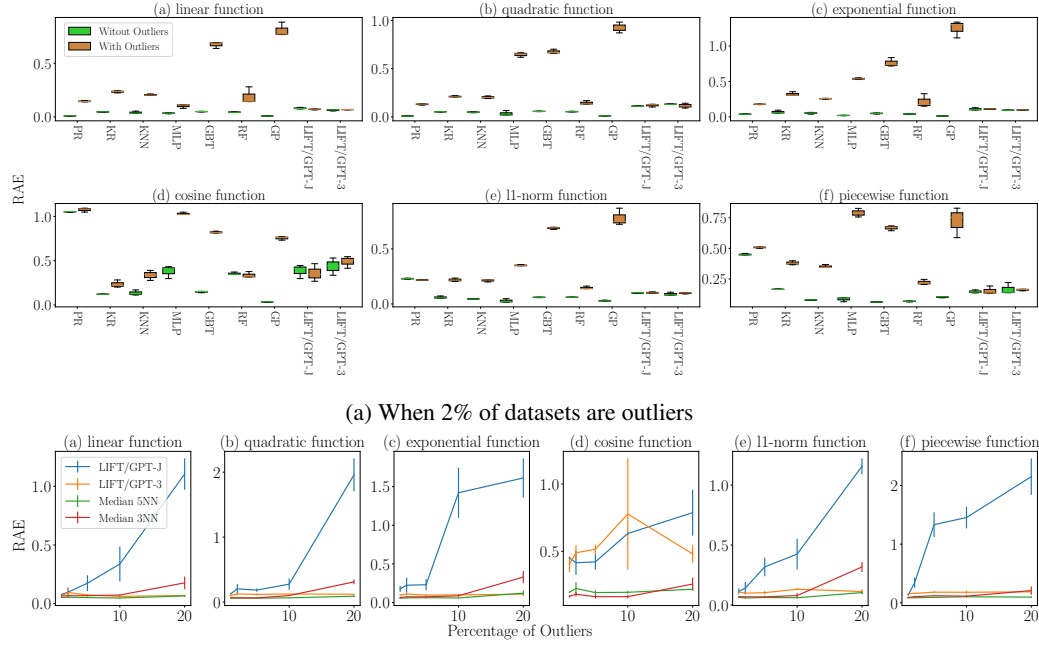

(a) When 2% of datasets are outliers

(b) For various portions of outliers (1%, 2%, 5%, 10%, 20%)

Figure 28: **Comparing robustness of methods against outliers on regression tasks when the datasets contain (a) 2% outliers and (b) various portions of outliers.** We report each algorithm's regression error measured by Related Absolute Error (RAE). *(a)* When training datasets contain 2% outliers, all LIFT models are highly robust against outliers compared with baselines. *(b)* When we increase the fraction of outliers (up to 20%), LIFT/GPT-3 is comparable to the strong baseline (median KNN), while LIFT/GPT-J fails.

(50% shows the perfect balance). For the reference, we report the performances of the deterministic classifiers that always return the label of class 0 (DC-0) and class 1 (DC-1).

Though evaluated datasets all have high class-imbalance ratios, we find that LIFT can perform well, achieving high F1, precision, and recall scores across the tasks. For instance, on the `Customers` dataset (the class-imbalance ratio is nearly 8), MCC gets 0 for both precision and recall as all predicted labels of MCC are 0 (*that is*, the major class), while LIFT/GPT-J achieves the best recall ($82.61 \pm 7.10$) and F1 scores ($84.43 \pm 1.43$). Here, the 0 value of precision and recall in MCC means that MCC classifies all samples as negative, which is the major class in the training dataset.

**Robustness to feature corruption on test data.** Here we provide detailed experiment setting and experiment results on random noise perturbation. Given a perturbation budget $\epsilon \geq 0$, we consider two types of perturbation $\boldsymbol{\delta}$ with $\|\boldsymbol{\delta}\|_\infty \leq \epsilon$: random noise and adversarial perturbation [164]. For random noise $\boldsymbol{\delta}$, we test on two types: (1) random Gaussian noise $\boldsymbol{\delta} \sim \mathcal{N}(\mathbf{0}, \mathbf{I}_p)$ scaled to satisfy $\|\boldsymbol{\delta}\|_\infty = \epsilon$, and (2) signed constant noise $\boldsymbol{\delta}$ where each element $\delta_i$ has magnitude $\epsilon$ and random sign. For adversarial perturbation $\boldsymbol{\delta}$, we test on the transfer attack [51], *i.e.*, we generate an adversarial example $(\boldsymbol{x} + \boldsymbol{\delta}, y)$ for a source neural network (that we can access) with constraint $\|\boldsymbol{\delta}\| \leq \varepsilon$, and test whether the target network correctly classifies the adversarial examples.

Table 33 shows the results of LIFT and baselines for the MNIST classification problem. We test on random noise (Gaussian and signed constant) and PGD attacks transferred from LeNet-5 and MLP. We compare the results for three networks: LeNet-5, MLP (having 2 hidden layers, each with 300 neurons and 100 neurons), and LIFT/GPT-3. LIFT/GPT-3 is observed to tolerate random noise (both Gaussian and signed constant) for small perturbation radius $\epsilon = 0.01$.

We do not include the result for LIFT/GPT-J since it is not even robust against simple noise. Please refer to Section 5.2 to check the vulnerability of LIFT/GPT-J against test-time noise and how data augmentation improves the robustness of LIFT/GPT-J.

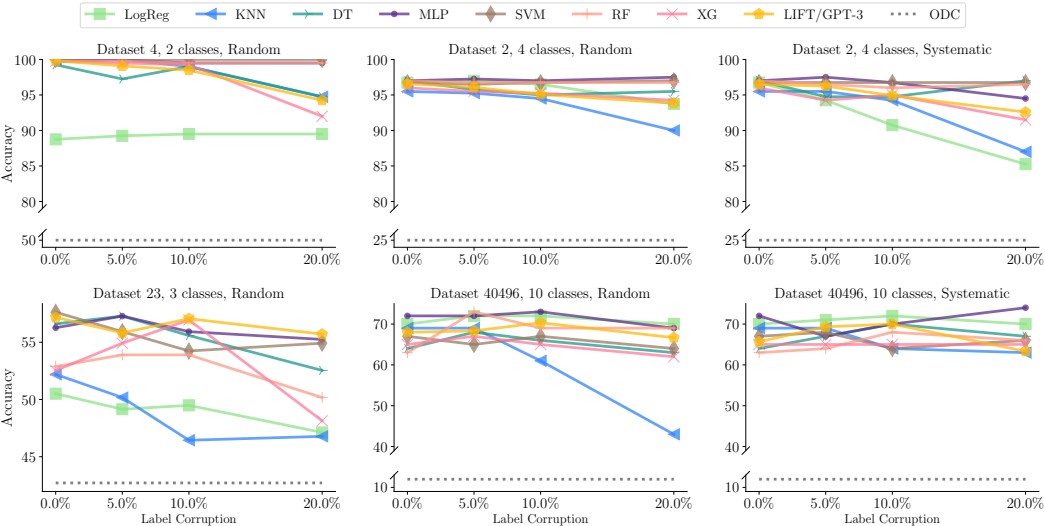

Figure 29: **Robustness against label corruption.** Each figure presents classification accuracies (↑) evaluated under different percentages of corruption in the training data (0% – 20%). We use synthetic data `Blobs` and `Moons` (ID 2 & 4) and real OpenML datasets `CMC` and `LED` (ID 23 & 40496). We simulate *random errors* (the first two columns) and *systematic errors* (the last column). LIFT/GPT-3 displays robustness across the datasets. *Here, ODC denotes optimal deterministic classifier, which is identical to the majority class classifier (MCC) in the main paper.*

### D.1.5 Does LIFT Need Large-Scale Models Pretrained on Natural Language Data?

Continuing from Sec. 3.6, we provide the detailed setup of this experiment. We obtain the Gibberish model by fine-tuning the entire GPT-J model (rather than LoRA [24]) on the Gibberish dataset [55] for 10 epochs at learning rate 0.1. For LIFT/Code-Gen, LIFT/CodeParrot, and LIFT/Gibberish, we follow the same e as LIFT/GPT-J, which we have discussed in Appendix C. To fine-tune LIFT/Rand-GPT-J for specific tasks, we set the temperature as 1 instead, as we observed the 0 temperature consistently gives us poor performances. Note that only 10%–15% of LIFT/Rand-GPT-J's outputs of are valid. The accuracies listed in the table are computed among valid outputs. Other settings of LIFT/RAnd-GPT-J are the same as LIFT/GPT-J.

### D.2 Results for LIFT-Specific Learning Properties (Section 4)

### D.2.1 Does LIFT Benefit from Incorporating Feature Names?

Continuing from Sec. 4.1, we provide more details of experiment settings, further evaluations with GPT-J models (Table 34), and results on regression tasks (Table 35) here.

**Prompt templates.** We design five prompts templates to assess how incorporating feature names affects the performance of LIFT. For instance, consider a data sample "$x =$ (`English speaker`, $23, 3$, `summer`, $19$), $y = 3$" from `TAE` dataset where the feature names are "`native speaker, instructor, course, semester, class size`", and the target attribute is `teaching performance`. We can incorporate the contextual information by either simply replacing the "$x_i$" in the prompts with the corresponding feature names or converting this sample into a coherent sentence. Meanwhile, we also investigate how shuffled feature names affect the performance of LIFT by designing the prompts accordingly. For illustration purposes, we provide the example of the five prompt templates as below.

- (W/O Names) "When we have $x_1 = 1, x_2 = 23, x_3 = 3, x_4 = 1, x_5 = 19$, what should be $y$ value?"
- (Correct-Names I)  "When we have `native speaker=English speaker, course instructor=23, course=3, semester=summer, class size=19`, how is the `teaching performance`?"

Table 30: **Accuracy(↑) comparison of various methods fitted to *randomly* corrupted classification labels.** In this regime, we corrupt a sample by assigning it another random label in the label space.

| Dataset (ID) | Corruption | MCC | LogReg | KNN | DT | MLP | SVM | RF | XG | LIFT/GPT-3 | LIFT/GPT-J |
|---|---|---|---|---|---|---|---|---|---|---|---|
| Blobs (2) | 0% | 25.00 | 96.75 | 95.50 | 97.00 | 97.00 | 96.75 | 97.00 | 96.00 | 96.58 | 96.17 |
| | 5% | 25.00 | 97.00 | 95.25 | 95.75 | 97.25 | 96.50 | 96.75 | 95.50 | 96.08 | 94.83 |
| | 10% | 25.00 | 96.50 | 94.50 | 95.00 | 97.00 | 96.75 | 96.75 | 95.25 | 95.08 | 91.38 |
| | 20% | 25.00 | 93.75 | 90.00 | 95.50 | 97.50 | 97.00 | 96.75 | 94.25 | 93.83 | 83.12 |
| Moons (4) | 0% | 50.00 | 88.75 | 100.00 | 99.25 | 99.75 | 100.00 | 99.75 | 99.75 | 99.83 | 99.58 |
| | 5% | 50.00 | 89.25 | 100.00 | 97.25 | 100.00 | 100.00 | 99.75 | 99.50 | 99.08 | 96.50 |
| | 10% | 50.00 | 89.50 | 99.00 | 99.00 | 99.50 | 100.00 | 99.50 | 99.25 | 98.50 | 94.00 |
| | 20% | 50.00 | 89.50 | 94.75 | 94.75 | 99.50 | 100.00 | 99.50 | 92.00 | 94.25 | 79.88 |
| CMC (23) | 0% | 42.71 | 50.51 | 52.20 | 56.61 | 56.27 | 57.63 | 52.88 | 52.54 | 57.18 | 49.83 |
| | 5% | 42.71 | 49.15 | 50.17 | 57.29 | 57.29 | 55.93 | 53.90 | 54.92 | 55.82 | 50.28 |
| | 10% | 42.71 | 49.49 | 46.44 | 55.59 | 55.93 | 54.24 | 53.90 | 56.95 | 57.06 | 48.47 |
| | 20% | 42.71 | 47.12 | 46.78 | 52.54 | 55.25 | 54.92 | 50.17 | 48.14 | 55.71 | 45.42 |
| TAE (48) | 0% | 35.48 | 51.61 | 61.29 | 67.74 | 58.06 | 61.29 | 77.42 | 64.52 | 50.54 | 61.29 |
| | 5% | 35.48 | 54.84 | 61.29 | 67.74 | 45.16 | 67.74 | 64.52 | 74.19 | 45.16 | 53.76 |
| | 10% | 35.48 | 41.94 | 45.16 | 54.84 | 32.26 | 32.26 | 51.61 | 54.84 | 52.69 | 46.24 |
| | 20% | 35.48 | 29.03 | 48.39 | 48.39 | 32.26 | 45.16 | 48.39 | 45.16 | 47.31 | 35.48 |
| Pollen (871) | 0% | 50.00 | 49.09 | 46.88 | 48.96 | 49.22 | 51.56 | 45.97 | 48.31 | 49.57 | 50.39 |
| | 5% | 50.00 | 51.43 | 48.18 | 49.22 | 50.26 | 49.74 | 48.44 | 46.62 | 50.65 | 48.61 |
| | 10% | 50.00 | 48.70 | 48.70 | 47.27 | 50.00 | 46.49 | 51.17 | 47.01 | 48.96 | 48.66 |
| | 20% | 50.00 | 50.39 | 49.22 | 50.52 | 47.01 | 50.52 | 47.14 | 49.61 | 50.74 | 50.82 |
| Climate (1467) | 0% | 91.67 | 89.81 | 89.81 | 91.67 | 91.67 | 87.96 | 91.67 | 90.74 | 91.67 | 87.04 |
| | 5% | 91.67 | 89.81 | 91.67 | 87.04 | 91.67 | 91.67 | 90.74 | 87.96 | 91.36 | 85.49 |
| | 10% | 91.67 | 90.74 | 88.89 | 90.74 | 88.89 | 91.67 | 91.67 | 88.89 | 91.67 | 83.80 |
| | 20% | 91.67 | 90.74 | 81.48 | 83.33 | 88.89 | 91.67 | 89.81 | 87.04 | 91.67 | 76.39 |
| LED (40496) | 0% | 12.00 | 70.00 | 69.00 | 64.00 | 72.00 | 67.00 | 63.00 | 65.00 | 68.00 | 67.33 |
| | 5% | 12.00 | 72.00 | 69.00 | 68.00 | 72.00 | 65.00 | 73.00 | 67.00 | 68.33 | 60.33 |
| | 10% | 12.00 | 72.00 | 61.00 | 66.00 | 73.00 | 67.00 | 69.00 | 65.00 | 70.33 | 56.00 |
| | 20% | 12.00 | 70.00 | 43.00 | 63.00 | 69.00 | 64.00 | 69.00 | 62.00 | 66.67 | 47.00 |

Table 31: **Accuracies(↑) of various methods fitted to *systematically* corrupted classification labels.** In this regime, we corrupt a label by assigning all corrupted labels of one class to a single label.

| Dataset ID | Corruption | MCC | LogReg | KNN | DT | MLP | SVM | RF | XG | LIFT/GPT-3 | LIFT/GPT-J |
|---|---|---|---|---|---|---|---|---|---|---|---|
| Blobs (2) | 0% | 25.00 | 96.75 | 95.50 | 97.00 | 97.00 | 96.75 | 97.00 | 96.00 | 96.50 | 96.17 |
| | 5% | 25.00 | 94.25 | 95.50 | 94.75 | 97.50 | 96.75 | 96.50 | 94.25 | 96.25 | 94.75 |
| | 10% | 25.00 | 90.75 | 94.25 | 94.75 | 96.75 | 96.75 | 96.00 | 95.00 | 94.92 | 90.17 |
| | 20% | 25.00 | 85.25 | 87.00 | 97.00 | 94.50 | 96.75 | 96.50 | 91.50 | 92.58 | 81.07 |
| LED (40496) | 0% | 12.00 | 70.00 | 69.00 | 64.00 | 72.00 | 67.00 | 63.00 | 65.00 | 65.67 | 67.33 |
| | 5% | 12.00 | 71.00 | 69.00 | 67.00 | 67.00 | 68.00 | 64.00 | 65.00 | 69.33 | 58.00 |
| | 10% | 12.00 | 72.00 | 64.00 | 70.00 | 70.00 | 64.00 | 68.00 | 65.00 | 70.00 | 55.67 |
| | 20% | 12.00 | 70.00 | 63.00 | 67.00 | 74.00 | 66.00 | 66.00 | 65.00 | 63.33 | 53.00 |

Table 32: **Comparing accuracy (↑), F1 (↑), Precision (↑), and Recall (↑) on imbalanced datasets in OpenML (`Pizza`, `Climate`, `Customers`).** All datasets are for binary classification and are highly imbalanced. The class-imbalance ratio (Imb. Ratio) is defined as the ratio of the number of samples in the majority class and that in the minority class. Here, DC-0 and DC-1 refer to deterministic classifiers that constantly predict all samples as class 0 and 1 respectively. MCC refers to the majority class classifier that returns the major class learned from the training dataset. LIFT/GPTs achieve comparably high scores across the three tasks. For instance, LIFT/GPT-J achieves the best F1 on datasets `Pizza` and `Customers`.

| Dataset (ID) | Imb. Ratio | | MCC | DC-0 | DC-1 | LogReg | KNN | DT | MLP | RBF-SVM | RF | XG | LIFT/GPT-J | LIFT/GPT-3 |
|---|---|---|---|---|---|---|---|---|---|---|---|---|---|---|
| Pizza (1444) | 7.36 | Accuracy | 88.04 | 88.04 | 11.96 | 86.92±0.23 | 87.56±0.68 | 87.24±0.60 | 86.28±1.37 | 88.04±0.00 | 88.04±1.04 | 88.04±0.68 | 83.89±0.45 | 85.17±1.35 |
| | | F1 | 0.00 | 0.00 | 21.37 | 10.77±2.74 | 16.84±5.26 | 9.01±12.74 | 11.77±3.43 | 0.00±0.00 | 15.79±6.45 | 35.50±3.68 | 35.83±3.61 | 24.52±1.78 |
| | | Precision | 0.00 | 0.00 | 11.96 | 28.97±3.41 | 42.86±10.10 | 13.89±19.64 | 32.69±12.26 | 0.00±0.00 | 51.67±23.21 | 52.21±7.28 | 38.84±5.24 | 32.41±6.55 |
| | | Recall | 0.00 | 0.00 | 100.00 | 6.67±1.89 | 10.67±3.77 | 6.67±9.43 | 8.00±3.27 | 0.00±0.00 | 9.33±3.77 | 28.00±5.66 | 33.33±2.36 | 20.00±0.00 |
| Climate (1467) | 11.00 | Accuracy | 91.67 | 8.33 | 91.67 | 88.89±0.76 | 90.74±0.76 | 88.89±2.27 | 91.67±0.00 | 87.96±0.00 | 91.36±0.44 | 89.51±0.87 | 87.04±2.27 | 91.67±0.00 |
| | | F1 | 95.65 | 0.00 | 95.65 | 94.00±0.43 | 95.13±0.40 | 94.04±1.27 | 95.65±0.00 | 93.47±0.00 | 95.48±0.24 | 94.37±0.48 | 94.51±1.08 | 95.65±1.00 |
| | | Precision | 91.67 | 0.00 | 91.67 | 93.07±0.06 | 91.85±0.43 | 92.26±1.20 | 91.67±0.00 | 93.00±0.00 | 91.64±0.04 | 92.83±0.43 | 92.10±0.36 | 91.67±0.00 |
| | | Recall | 100.00 | 0.00 | 100.00 | 94.95±0.82 | 98.65±0.48 | 95.96±2.86 | 100.00±0.00 | 93.94±0.00 | 99.66±0.48 | 95.96±0.82 | 97.08±2.57 | 100.00±0.00 |
| Customers (1511) | 2.14 | Accuracy | 68.18 | 68.18 | 31.82 | 87.12±0.54 | 88.64±0.00 | 85.98±0.53 | 86.36±1.86 | 86.36±0.00 | 85.23±0.00 | 85.23±0.00 | 85.23±1.61 | 84.85±1.42 |
| | | F1 | 0.00 | 0.00 | 48.28 | 79.76±0.89 | 80.51±0.36 | 78.60±1.00 | 77.80±2.79 | 78.82±0.35 | 76.64±0.39 | 76.91±0.39 | 84.43±1.43 | 75.28±2.60 |
| | | Precision | 0.00 | 0.00 | 31.82 | 79.79±1.23 | 88.64±1.61 | 76.40±0.38 | 81.00±4.65 | 77.94±0.90 | 77.14±0.91 | 76.50±0.91 | 87.57±7.11 | 78.18±1.97 |
| | | Recall | 0.00 | 0.00 | 100.00 | 79.76±1.68 | 73.81±1.68 | 80.95±1.68 | 75.00±2.91 | 79.76±1.68 | 76.19±1.68 | 77.38±1.68 | 82.61±7.10 | 72.62±3.37 |

- (Correct-Names II) "In the course 3 offered in the summer semester, there was a native English-speaking teaching assistant and an instructor whose ID is 23. How is the teaching performance?"

Table 33: **Accuracies (↑) of LIFT and baselines (LeNet-5, MLP) under the perturbation on the input feature of MNIST data.** Given the perturbation budget $\varepsilon \in [0, 1]$, we test on four types of perturbations within $L_\infty$ ball of radius $\varepsilon$. (1): adding random Gaussian noise that is scaled to reach the $L_\infty$ ball, (2): adding *signed constant* noise vector where each element has magnitude $\epsilon$ and random sign, (3) & (4): adversarial examples generated from a source network (LeNet-5 & MLP, respectively) using PGD attack [165] from foolbox [166]. For small perturbation radii ($\epsilon = 0.01$), LIFT/GPT-3 maintains high accuracy for random noise, both for Gaussian and signed constant noise types. When $\epsilon = 0.01$ or $\epsilon = 0.1$, the performance of LIFT/GPT-3 for random noise and transferred adversarial attacks have significant gap, showing that the adversarial examples generated at LeNet-5 and MLP are transferred to LIFT/GPT-3.

| Source | **Random noise (Gaussian)** | | | **Random noise (signed const.)** | | | **PGD attack on LeNet-5** | | | **PGD attack on MLP** | | |
|---|---|---|---|---|---|---|---|---|---|---|---|---|
| Target | LeNet-5 | MLP | LIFT/GPT-3 | LeNet-5 | MLP | LIFT/GPT-3 | LeNet-5 | MLP | LIFT/GPT-3 | LeNet-5 | MLP | LIFT/GPT-3 |
| $\varepsilon = 0$ | 99.22 | 98.09 | 98.15 | 99.22 | 98.09 | 98.15 | 99.22 | 98.09 | 98.15 | 99.22 | 98.09 | 98.15 |
| $\varepsilon = 0.01$ | 99.25 | 98.05 | 98.28 | 99.26 | 98.08 | 88.05 | 97.27 | 97.77 | 44.88 | 99.15 | 96.89 | 44.46 |
| $\varepsilon = 0.1$ | 99.20 | 97.70 | 88.38 | 99.06 | 97.39 | 68.80 | 26.80 | 93.99 | 33.66 | 96.98 | 23.12 | 23.62 |
| $\varepsilon = 0.3$ | 98.01 | 87.69 | 54.80 | 79.80 | 74.20 | 29.68 | 0.00 | 36.62 | 20.31 | 41.51 | 0.00 | 20.29 |

- (Shuffled-Names I) "When we have semester=English speaker, class size=$23$, semester=$3$, course instructor=summer, native speaker=$19$, how is the teaching performance?"
- (Shuffled-Names II) "In the course summer offered in the 3 semester, there was a 19 teaching assistant and an instructor whose ID is summer. How is the teaching performance?"

We note that the sentence generated using the (Shuffled-Names II) template can be incoherent.

**Settings.** *(Datasets)* Among the OpenML datasets evaluated in Table 4, we select three datasets: CMC, TAE, and Vehicle (with IDs being 23, 48, and 54) whose all provided feature names are meaningful and relevant to the prediction task and the response values. *(Baselines)* We compare our target model LIFT when feature names are correctly incorporated (Correct-Names I, II) with the versions of LIFT when feature names are incorrectly incorporated with randomly shuffled orders (Shuffled-Names I, II) and when feature names are not included (W/o Names). Also, we compare all models with the simple baseline MCC and the strong baseline XGBoost.

**Classification.** We provide additional evaluation of LIFT/GPT-J on three datasets used in the main paper. Table 34 presents our result with the same settings in the main paper. We can see that correctly using feature names helps improve the performance of LIFT from the models without feature names or the models with randomly shuffled feature names. This finding is consistent with the finding in the main papers on the usefulness of incorporating the feature names.

Table 34: **The effect of using feature names on LIFT/GPT-J.** We compare the classification accuracy (↑) of LIFT/GPT-J when feature names *are and are not* incorporated in prompts.

| Dataset (ID) | MCC | XGBoost | LIFT/GPT-J | | | | |
|---|---|---|---|---|---|---|---|
| | | | W/o Names | Shuffled-Names I | Shuffled-Names II | Correct-Names I | Correct-Names II |
| CMC (23) | 42.71 | 52.43±0.42 | 49.49±0.56 | **51.30±1.05** | **51.30±2.51** | 48.82±3.12 | 50.39±1.05 |
| TAE (48) | 35.48 | 66.67±8.05 | 60.22±4.02 | 63.44±6.08 | 58.06±7.90 | 60.21±10.64 | **65.59±8.47** |
| Vehicle (54) | 25.88 | 73.14±0.28 | 64.31±2.37 | 66.87±1.54 | 65.49±1.69 | **69.02±3.67*** | |

**Regression.** To investigate whether incorporating feature names in prompts improves the regression performance of LIFT, similar to the datasets selection process of classification tasks, we evaluate the effect of feature names on the datasets Insurance and Student, whose tasks can be helped by common knowledge. To be more specific, while the task of Insurance dataset is to predict the insurance costs, the key features of Insurance dataset are age, body mass index, and smoke or not, which are intuitively closely related to the task. For the Student dataset, the task is to predict students' grades based on their weekly study time, previous grades, etc. Therefore, the features and task of Student are also highly correlated. Table 35 presents our evaluation of regression tasks. We find that fine-tuning with feature names does not necessarily help with the regression tasks.

Table 35: **Investigating if incorporating feature names to LIFT improves sample efficiency in regression tasks.** The experiments are conducted on `Insurance` and `Student` datasets. The second column indicates the fraction of samples used for training the model. We observe no significant improvements in the performance when feature names are properly included.

| Dataset | Frac. | RF | LIFT/GPT-3 | | | | |
|---|---|---|---|---|---|---|---|
| | | | W/O Names | Shuffled-Names I | Shuffled-Names II | Correct-Names I | Correct-Names II |
| insurance | 0.2 | **0.31 ± 0.00** | 0.89 ±0.03 | 0.76 ±0.11 | 0.59 ±0.09 | 0.59 ±0.11 | 0.89 ±0.03 |
| | 0.4 | 0.26 ± 0.00 | 0.42 ±0.15 | 0.30 ±0.02 | **0.20 ±0.03** | 0.35 ±0.10 | 0.21 ±0.01 |
| | 0.6 | 0.26 ± 0.00 | 0.30 ±0.10 | 0.24 ±0.03 | **0.19 ±0.02** | 0.30 ±0.12 | 0.22 ±0.08 |
| | 0.8 | 0.27 ± 0.00 | 0.31 ±0.07 | 0.19 ±0.04 | 0.18 ±0.03 | 0.14 ±0.01 | **0.11 ±0.02** |
| | 1.0 | 0.26 ± 0.00 | 0.14 ±0.05 | 0.17 ±0.03 | 0.19 ±0.01 | 0.17 ±0.04 | **0.10 ±0.03** |
| student | 0.2 | 0.40 ± 0.00 | 0.32 ±0.01 | 0.32 ±0.01 | 0.34 ±0.02 | **0.31 ±0.01** | **0.31 ±0.01** |
| | 0.4 | 0.36 ± 0.00 | 0.32 ±0.02 | 0.31 ±0.01 | **0.30 ±0.00** | 0.32 ±0.01 | 0.35 ±0.01 |
| | 0.6 | 0.36 ± 0.00 | 0.31 ±0.01 | 0.31 ±0.01 | 0.31 ±0.01 | 0.31 ±0.01 | **0.30 ±0.00** |
| | 0.8 | 0.38 ± 0.00 | 0.28 ±0.01 | **0.27 ±0.01** | 0.29 ±0.02 | 0.28 ±0.01 | 0.28 ±0.00 |
| | 1.0 | 0.35 ± 0.00 | **0.27 ±0.01** | 0.28 ±0.01 | 0.28 ±0.01 | 0.28 ±0.01 | 0.35 ±0.02 |

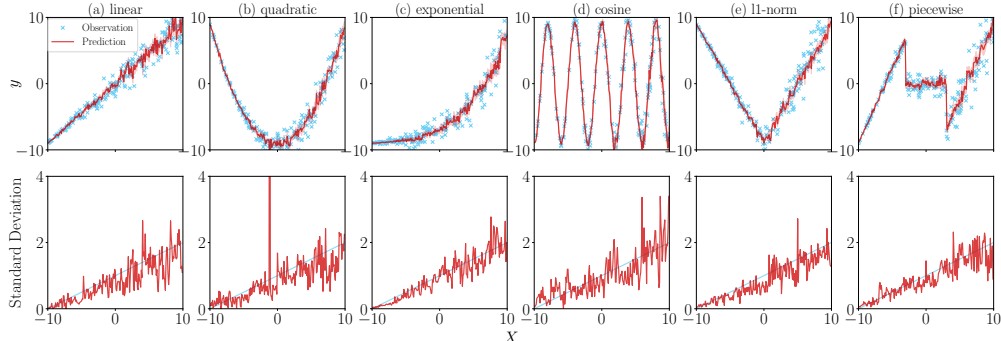

Figure 36: **Investigating calibrated prediction effect of LIFT.** Prediction standard deviations of LIFT/GPT align well to the observations (top), across datasets, implying the well calibration.

### D.2.2 Is LIFT Calibrated?

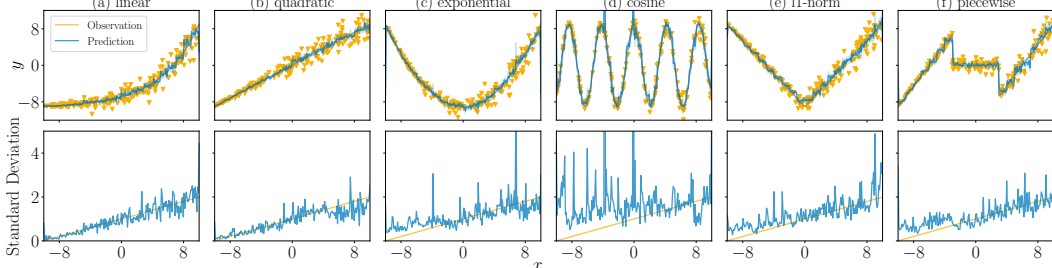

Figure 37: **Visualization of LIFT/GPT-3 predictions under varying noise levels.** The predictions are made on grid datasets consisting of 103 eveny-spaced samples in $[-10, 10]$. The standard deviation of LIFT/GPT-J predictions are computed based 20 repeated random predictions. We observe that the standard deviations of predictions from LIFT/GPT-3 aligns well with that of noisy training samples (observations), implying that LIFT/GPT-3 can reflect the confidence, *i.e.*, LIFT/GPT-3 is calibrated.

Continuing the discussion in Sec. 4.2, Fig. 37 indicates that LIFT/GPT-3 is calibrated.

### D.2.3 Can we use LIFT for Generation?

Here we provide the detailed experiment setting of Sec. 4.3.

**Data pre-processing.** We preprocess the MNIST dataset as below. First, we crop each $28 \times 28$ image at the center to make an $18 \times 18$ image. Then, we represent the cropped image as a sequence of 324 pixel values, where each pixel is an integer in $\{0, 1, \cdots, 255\}$.

The training process for both tasks (i) and (ii) are identical, detailed as below. Given an image in the training set, we put the digit of the image into a prompt and put the sequence of 324 pixel values into the completion, as in the example shown below.

- (Train Prompt) "`Generate an image of digit 9.###`"

- (Train Completion)  "0 0 0 0 0 0 0 0 0 0 0 0 0 0 0 0 0 0 0 0 0 0 0 0 0 0 0 1 90 255 254 255 96 0 0 0 0 0 0 0 0 0 0 0 24 139 253 253 253 253 247 95 0 0 0 0 0 0 0 0 0 24 211 253 231 134 12 125 251 175 0 0 0 0 0 0 0 0 0 196 253 215 26 0 0 0 63 47 0 0 0 0 0 4 61 21 48 239 211 28 0 0 0 0 14 59 181 206 19 0 0 26 181 64 230 249 63 0 0 0 0 165 253 253 253 36 0 0 0 0 166 253 98 0 0 0 0 33 198 252 210 242 253 36 0 0 0 0 234 253 31 0 0 13 63 250 253 207 44 235 253 36 0 0 0 0 234 253 162 6 137 221 253 250 169 25 79 253 224 21 0 0 0 0 148 253 253 253 253 243 161 42 0 0 100 253 186 0 0 0 0 0 6 81 155 155 111 41 0 0 0 0 182 253 186 0 0 0 0 0 0 0 0 0 0 0 0 18 223 253 84 0 0 0 0 0 0 0 0 0 0 0 0 0 0 32 253 246 54 0 0 0 0 0 0 0 0 0 0 0 0 0 129 253 233 0 0 0 0 0 0 0 0 0 0 0 0 135 253 201 0 0 0 0 0 0 0 0 0 0 0 0 0 0 186 253 129 0 0 0 0 0 0 0 0 0 0 0 0 0 0 239 253 59 0 0 0 0 @@@"

The testing process is different for tasks (i) and (ii). For task (i), we put the prompt as above and check the image drawn by pixel values generated in the completion. For task (ii), the test prompt is set as follows. First, we randomly sample an image with the target digit from the MNIST set. We then include its digit number and the top half pixels (first 162 pixels) in the prompt. Once this test prompt is given, we observe the remaining 162 pixel values generated in the completion, and check how the entire 324 pixels look like. Examples of the test prompt are given below.

- (Test Prompt for task (i)) "`Generate an image of digit 9.###`"

- (Test Prompt for task (ii)) "Generate an image of digit 9.### 0 0 0 0 0 0 0 0 0 0 0 0 0 0 0 0 0 0 0 0 0 0 1 90 255 254 255 96 0 0 0 0 0 0 0 0 0 0 0 24 139 253 253 253 253 247 95 0 0 0 0 0 0 0 0 0 24 211 253 231 134 12 125 251 175 0 0 0 0 0 0 0 0 0 196 253 215 26 0 0 0 63 47 0 0 0 0 0 4 61 21 48 239 211 28 0 0 0 0 14 59 181 206 19 0 0 26 181 64 230 249 63 0 0 0 0 165 253 253 253 36 0 0 0 0 166 253 98 0 0 0 0 33 198 252 210 242 253 36 0 0 0 0 234 253 31 0 0 13 63 250 253 207 44 235 253 36 0 0"

**Experiment setup.**  For GPT-3, we fine-tune with the pre-processed MNIST training set for 15 epochs. For GPT-J, we fine-tune for three epochs. We use the learning rate $10^{-5}$. In the inference phase, as described above, we use different prompts for the two tasks and collect the first 324 and 162 white space-separated tokens in the output for tasks (i) and (ii), respectively. In addition, for task (ii), we concatenate the provided 162 tokens in the prompt and the 162 tokens from the model output into a sequence of 324 tokens. The collected 324 tokens are converted to an $18 \times 18$ image if all of them are valid numbers in $\{0, 1, \cdots, 255\}$. We assess the performance of LIFT as a generator under different temperatures $\in [0, 0.3, 0.5, 0.7, 0.9, 1]$. Under a fixed temperature, we generate images of each digit 5 times. We use the standard perplexity score to evaluate the performance of the obtained generative models. Here perplexity is a standard metric for assessing the performance of a generative model, which reflects the inverse of the probability of the given data samples being produced by the model. In this experiment, we assess the generalization performance of models by comparing the perplexity scores between training and test sets. If the gap between the training perplexity and the test perplexity is small, it indicates that the model generalizes well. [167].

**Results.**  Fig. 38 visualizes the output of LIFT in generating the images of different digits under different temperatures. We observe that LIFT is able to generate reasonable images when both the digit number and top half pixels are given, as shown in Fig. 38c and Fig. 38d. In contrast, Fig. 38a and Fig. 38b show that LIFT is able to work only under high temperatures (*e.g.*, `Temp = 0.9, 1`). This might be because, under low temperatures, the model is likely to generate output associated with the highest probability. In contrast, a higher temperature gives chances to generate outputs associated with slightly lower probability, which introduces more variety and creativity. It has been observed in

compositional text generation tasks such as completing stories, high temperature leads to better and more creative performance [168].

Table 39: **Efficacy of LIFT as generative models.** Perplexity ($\downarrow$) is a metric for measuring the probability of the sample produced by the model on a dataset. We report the average perplexity of LIFT trained for generating MNIST images. Note that the difference between the average test perplexity and average training perplexity is small, implying the good generalizability of LIFT as generative models.

|  | LIFT/GPT-J | LIFT/GPT-3 |
| --- | --- | --- |
| Avg Training Perplexity ($\downarrow$) | $3.56 \pm 1.42$ | $3.58 \pm 1.46$ |
| Avg Test Perplexity ($\downarrow$) | $3.57 \pm 1.44$ | $3.62 \pm 1.51$ |

## D.3 Results for Improving Techniques of LIFT (Section 5)

### D.3.1 Two-Stage Intermediate Fine-Tuning for LIFT

We provide an additional evaluation of the two-stage intermediate fine-tuning with LIFTGPT-J on four more datasets. For any given dataset, we first generate two pretext tasks with simple synthetic Gaussian samples (discussed in C.1). We fine-tune the GPT with pretext tasks for a few (2–3) epochs, then fine-tune the newly fine-tuned GPT with the target (given) dataset. Here, due to the black-box API of GPT-3, we currently can neither keep the order of samples unchanged (pretext, target) during the fine-tuning stage nor fine-tune the model twice. Hence, we only use GPT-J in this experiment.

Fig. 40 presents results from eight datasets, including a regression task, three OpenML tasks, and other synthetic tasks. We see that two-stage intermediate LIFT helps to improve the original fine-tuning, especially when the number of training samples is small. Its effect is more clearly shown in synthetic classification datasets (Blobs, Circles, Moon, and Two Circles). We also observe that besides the number of features and number of classes, the pretexts do not need to represent any other characteristics of the target dataset, such as the linear/non-linear correlation or the relevance of features. This makes it simpler to generate the pretexts.

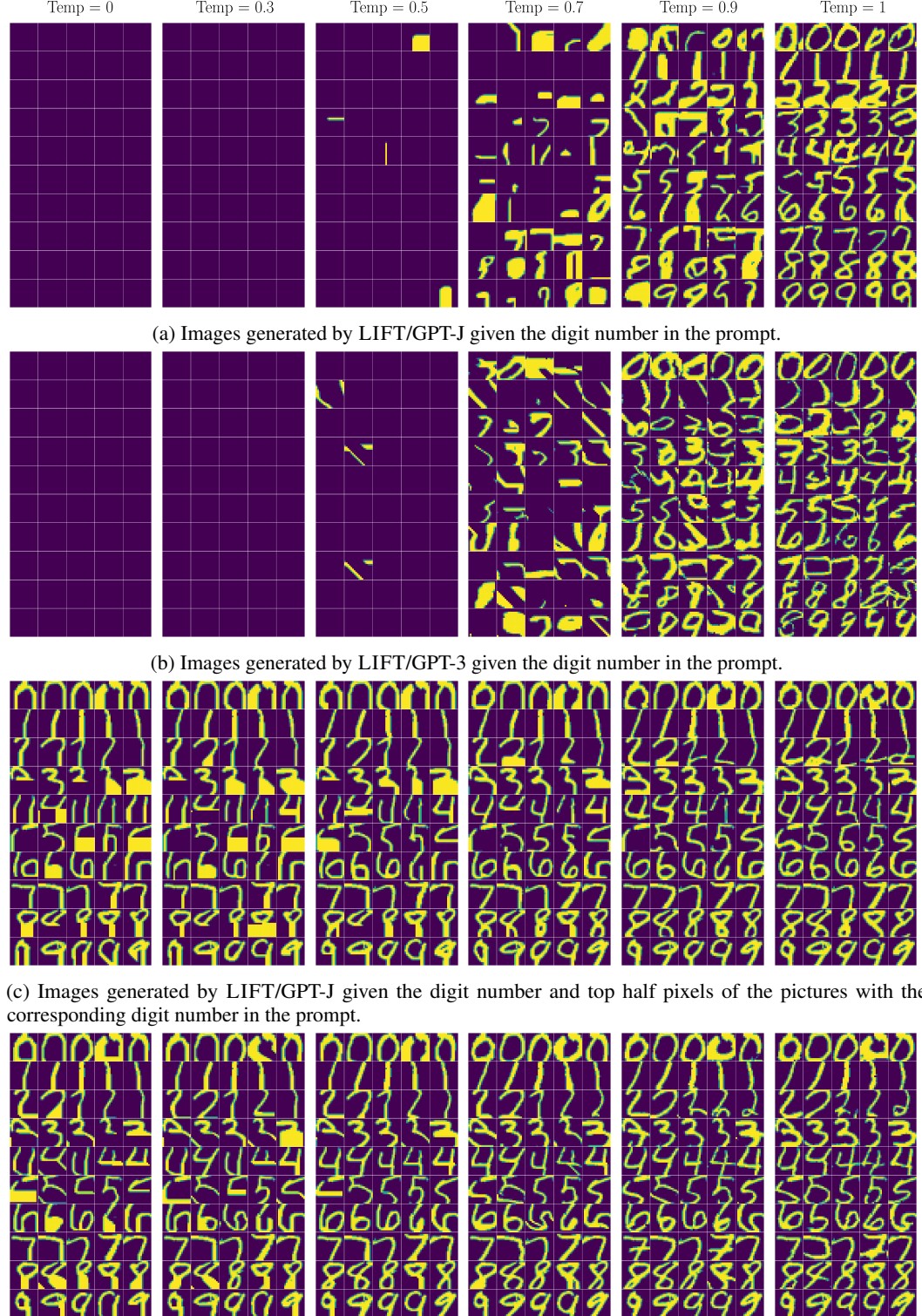

(a) Images generated by LIFT/GPT-J given the digit number in the prompt.

(b) Images generated by LIFT/GPT-3 given the digit number in the prompt.

(c) Images generated by LIFT/GPT-J given the digit number and top half pixels of the pictures with the corresponding digit number in the prompt.

(d) Images generated by LIFT/GPT-3 given the digit number and top half pixels of the images with the corresponding digit number in the prompt.

Figure 38: **Output of LIFT as generative models.** We apply LIFT to generate new MNIST images. Each figure contains six subfigures, where each subfigure visualizes the output of LIFT when different temperatures $\in [0, 0.3, 0.5, 0.7, 0.9, 1]$ of LMs are chosen. We generate five images for each digit by using LIFT to make the prediction five times. Task (i): when only the digit number is given, we observe that the LIFT can generate reasonable images under high temperatures. Task (ii): when both digit number and top half pixels are given, LIFT can generate images of comparably high quality under different temperatures.

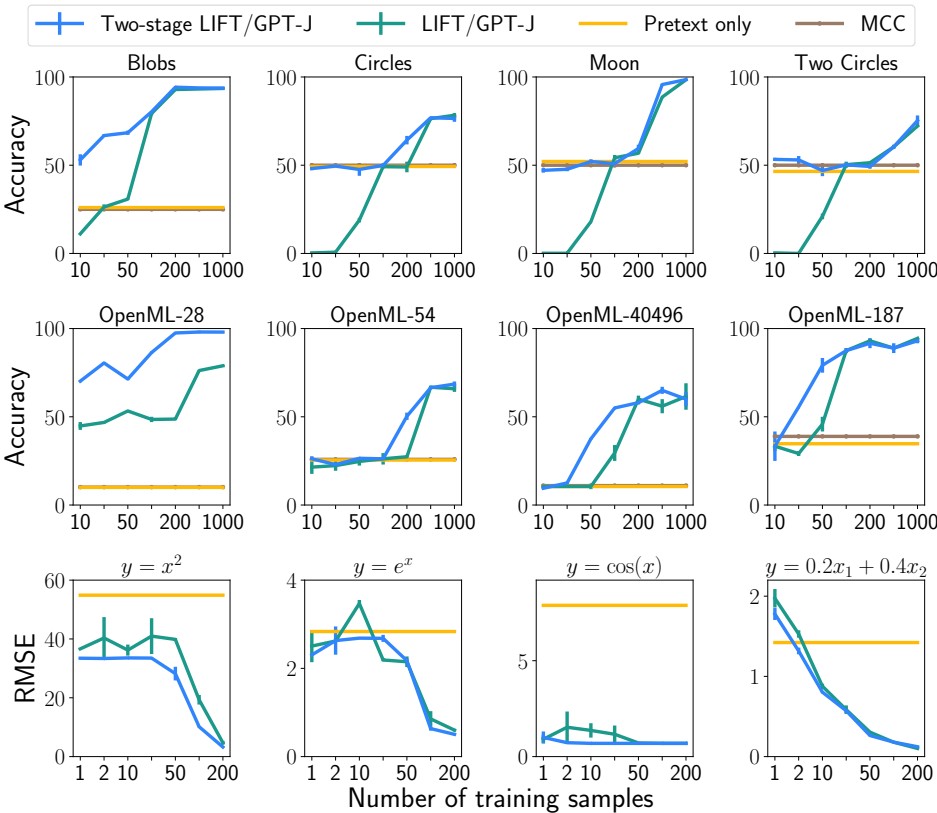

Figure 40: **Two-stage fine-tuning with LIFT/GPT-J**. We apply LIFT in two consecutive stages: first on synthetic pretext data, then on the target data. We evaluate on classification tasks (with accuracy) and a regression task (RMSE error). The two-stage fine-tuning LIFT(blue) outperforms the original fine-tuning LIFT (green) when the number of training samples is small, across all tasks.

# E  Additional Experiments and Findings (NOT Discussed in the Main Paper)

Here, we provide results of additional experiments that have not been discussed in the main manuscript. To be more specific, we study the effect of replacing the input or output layers (E.1), effect of large LMs (E.2), quantitative classification evaluation on neural-net-based synthetic datasets (E.3), and the ability of LIFT performing the ridge regression (E.4). We also provide the visualization of LIFT's training curve in Section E.5.

## E.1  What Is the Effect of Replacing the Input or Output Layers?

In this experiment, we assess the performance of transformer fine-tuned with replaced input/output layer, following the methods used in Frozen Pretrained Transformers (FPT) [18]. We consider vanilla FPT and its two variants. Specifically, for vanilla FPT, we reinitialized a trainable input layer and a trainable output layer, with frozen pretrained GPT-J transformer architectures in the middle. As in [18], the input dimension equals to the number of features and the output dimension equals to the number of classes, which varies depending on the tasks. Table 41 compares the result of our method LIFT/GPT-J, FPT and the two variants of FPT: (i) FPT (Output Only) which only replaces the output layer, and (ii) FPT (Input Only), which only replaces the input layer. We observe that both FPT and FPT (Output Only) perform slightly better than LIFT on almost all tested cases, while FPT (Input Only) performs the worst. Our justification for this observation is that training an output layer is similar to training a linear classifier, which might be easier than training an input layer as an encoder.

Table 41: **Accuracies(↑) of LIFT, Frozen Pretrained Transformer (FPT) [18] and its two variants — FPT (Output Only) and FPT (Input Only).** We employ GPT-J in this experiment. We observe that replacing the output layer can slightly improve the performance of LIFT, while only replacing the input layer performs the worst.

| Dataset (ID) | LIFT/GPT-J | FPT/GPT-J | FPT (Output Only)/GPT-J | FPT (Input Only)/GPT-J |
|---|---|---|---|---|
| Blobs (2) | 96.17±0.59 | 96.75±0.00 | 96.67±0.12 | 96.75±0.00 |
| Two Circles (6) | 75.92±1.65 | 74.33±0.31 | 76.33±2.49 | 69.83±1.31 |
| Iris (61) | 96.67±0.00 | 96.67±0.00 | 97.78±1.57 | 81.11±3.14 |
| Customers (1511) | 85.23±1.61 | 87.88±0.54 | 88.26±0.54 | 86.74±1.42 |
| Wine (187) | 93.52±1.31 | 100.00±0.00 | 99.07±1.31 | 92.59±3.46 |
| LED (40496) | 65.33±0.47 | 73.00±2.94 | 71.67±1.25 | 68.67±1.89 |

Table 42: **The effects of larger LMs under different classification settings**. Recall that our previous results on GPT-3 are based on the smallest model `Ada`. Here we use larger GPT-3 versions (`Babbage`, `Curie`, `Davinci`) as the pretrained LMs in our framework and evaluate the classification accuracy (↑) of them in three settings: classification *without* feature name, classification *with* feature name and in-context classification. For the setting of classification *with* feature names, we incorporate names of features (columns) into the input prompts (see more details in Sec. 4.1). For in-context learning, the OpenML dataset ID and number of prompts are written together at each column, *e.g.*, `TAE (48)/50` means that we run experiments on the OpenML dataset `TAE` having ID 48, by using 50 input prompts. For the first two settings when LIFT is applied, larger LIFT/GPT-3 models (`Babbage`, `Curie`, `Davinci`) perform better than the smaller models LIFT/GPT-3-`Ada` and LIFT/GPT-J, but the performance gains are not always consistent and significant with model sizes. For the in-context classification (LIFT is not used), we observe more consistent improvement by using larger models.

| Tasks | | LIFT Classification W/O Feat. Names | | | | | LIFT Classfication W/ Feat. Names | | |
|---|---|---|---|---|---|---|---|---|---|
| Dataset (ID) | | Customers (1511) | Texture (1493) | Margin (1491) | TAE (48) | Vehicle (54) | TAE (48) | CMC (23) | Vehicle (54) |
| **LIFT/GPT-J** | | 93.97±1.00 | 50.32±2.18 | 50.23±1.33 | 61.29±6.97 | 64.31±2.37 | 67.74±11.48 | 48.36±0.97 | 69.02±3.67 |
| **LIFT/GPT-3** | **Ada** | 95.39±0.67 | 67.50±1.42 | 59.37±0.92 | 65.59±6.63 | 70.20±2.73 | 67.74±2.63 | 57.48±1.14 | 72.16±2.00 |
| | **Babbage** | 96.81±0.07 | 62.19±1.80 | 67.50±3.87 | 61.29±6.97 | 72.06±3.82 | 64.52±6.97 | 57.06±2.15 | 70.00±1.44 |
| | **Curie** | 95.21±0.06 | 62.50±0.97 | 61.88±1.48 | 66.67±6.09 | 74.27±0.73 | 65.59±4.02 | 55.42±0.84 | 70.66±2.28 |
| | **Davinci** | 96.81±0.41 | 57.19±0.70 | 58.13±2.50 | 64.52±9.50 | 71.47±0.88 | 65.59±6.63 | 56.31±0.04 | 68.16±1.69 |

| Tasks | | In-context Classifction | | | | | |
|---|---|---|---|---|---|---|---|
| Dataset (ID) / #Prompts | | TAE (48)/50 | Breast (13)/35 | LED (40496)/32 | Customers (1511)/28 | Vehicle (54)/42 | Hamster (893)/13 |
| **GPT-J** | | 34.33±1.47 | 56.90±19.51 | 10.00±0.82 | 56.06±17.14 | 25.49±0.55 | 48.89±3.14 |
| **GPT-3** | **Ada** | 37.64±4.02 | 62.07±1.41 | 8.00±1.63 | 60.61±1.42 | 28.82±2.10 | 57.78±6.29 |
| | **Babbage** | 47.31±3.04 | 71.26±0.81 | 11.00±0.00 | 53.79±12.07 | 24.32±0.56 | 53.33±5.44 |
| | **Curie** | 32.26±0.00 | 70.69±0.00 | 20.67±4.78 | 67.80±0.53 | 26.28±2.22 | 53.33±0.00 |
| | **Davinci** | 49.46±4.02 | 67.82±4.06 | 20.67±6.60 | 68.94±0.54 | 26.28±2.22 | 55.55±3.14 |

Table 43: **Comparison of LIFT on different LMs across regression tasks.** The regression performance is measured by RAE (↓). In general, LIFT/GPT-3 with `Davinci` model performs the best, but the gaps to other models are not always significant.

| Method / Function | LIFT/GPT-J | LIFT/GPT-3 | | | |
|---|---|---|---|---|---|
| | | Ada | Babbage | Curie | Davinci |
| linear | 0.08±0.01 | 0.06±0.01 | 0.06±0.00 | 0.06±0.01 | 0.06±0.00 |
| quadratic | 0.11±0.00 | 0.13±0.00 | 0.11±0.02 | 0.10±0.01 | 0.09±0.00 |
| exponential | 0.11±0.02 | 0.09±0.00 | 0.09±0.01 | 0.08±0.00 | 0.08±0.00 |
| cosine | 0.38±0.08 | 0.44±0.10 | 0.41±0.06 | 0.38±0.01 | 0.38±0.05 |
| L1-norm | 0.10±0.00 | 0.09±0.01 | 0.10±0.01 | 0.08±0.01 | 0.09±0.01 |
| piecewise | 0.15±0.01 | 0.17±0.05 | 0.15±0.02 | 0.15±0.01 | 0.14±0.01 |

## E.2 Does LIFT Benefit from Larger LMs?

In this experiment, we apply LIFT to different pretrained LMs to verify whether LIFT benefits more from larger LMs. Together with previously used GPT-J and GPT-3 (the version named `Ada`), we consider three bigger versions of GPT-3, namely `Baggage`, `Curie`, and `Davinci` (in the ascending

order of the number of parameters). We compare all models on several classification tasks in Table 42 and regression tasks in Table 43. Overall, we find that the performance gain of using larger LMs is not consistently significant for LIFT. Although larger LMs outperform smaller LMs in many cases, the improvements are relatively small.

**Verifying the capability of large LMs, when LIFT is not used.** We first verify if larger LMs are more helpful for the evaluated downstream tasks. We evaluate LMs in the in-context classification when no fine-tuning (LIFT) is involved. Table 42 shows consistent improvements in classification performance when the size of LMs increases across all the tasks. Thus, larger LMs, with larger embedded knowledge, are more useful for these downstream tasks.

**When LIFT is used.** Both Table 42 and Table 43 show that using larger LMs may positively affect LIFT in several tasks and settings compared to the smaller LMs. However, the performance gains from replacing the smaller LMs with larger LMs are not consistent across the settings. For instance, in the classification settings without feature names, `Davinci` performs better than GPT-J on four datasets and worse than on one dataset. For the setting with feature names, GPT-J performs better than `Davinci` on two out of three tasks. Furthermore, the performance gains of large LMs over the smaller models are not relatively significant. We note that LIFT always outperforms the in-context learning using the same pretrained LMs in most cases. The regression results shown in Table 43 further confirm that the improvement from utilizing larger LMs is relatively small.

### E.3 Quantitative Classification Evaluations on Neural-Net-Based Synthetic Datasets

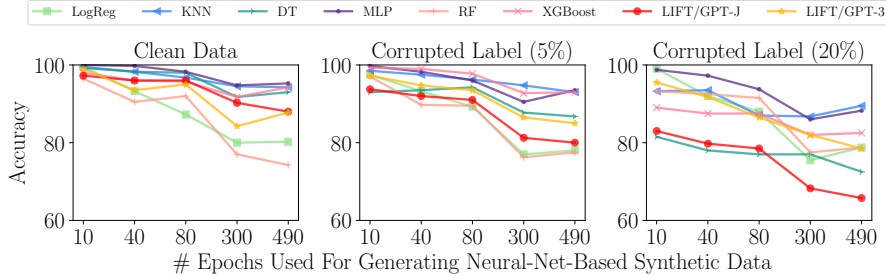

Figure 44: **How accuracy (↑) changes as the target classification problem becomes more complex, i.e., the ground-truth decision boundary becomes more complex.** The x-axis shows the number of epochs we used to train the neural network on the `Rolls` dataset. Note that the network becomes more complex as the number of epochs increases. Thus the classification problem also gets challenging. We measure the performances on three cases: (left) clean data, (middle) label-corrupted data with corruption probability 5% and (right) 20%.

In Sec. 3.4 and Appendix D.1.3, we assess how well LIFT/GPTs adapt to different shapes of decision boundaries on neural-net-based synthetic datasets. We now provide the test accuracies for all models. *For binary-class datasets,* Fig. 44 shows the accuracies on three binary-class datasets when the difficulty of classification tasks varies by using different network checkpoints at different epochs. As the difficulty increases or the level of corruption increases, all methods tend to decrease classification accuracy. We provide three settings of training data: clean data, 5% label corruption, and 20% label corruption. We observe that LIFT/GPT-3 outperforms logistic regression and decision tree, especially when the label corruption is up to 20%. However, LIFT/GPT-J is performing worse than other baselines in the data corruption scenarios. *For 3-class and 5-class datasets,* both LIFT/GPT-J and LIFT/GPT-3 achieve approximately 90%, while the best baselines (MLP and XGBoost) obtains approximately 92% and 91% for the 3-class and 5-class data, respectively.

### E.4 Can LIFT Perform Ridge Regression via Data Augmentation?

As shown in Fig. 2, LIFT can perform linear regression. We take one step further and study whether LIFT can perform Ridge regression. Note that this is a non-trivial task as the LIFT framework does not allow any changes to the loss function. Consider a standard ridge regression problem solving the optimal $w$ with $p$ parameters so that $\|y - Xw\|_2^2 + \lambda\|w\|_2^2$ is minimized. Note that this problem is equivalent to minimizing $\|[y^T, 0]^T - [X^T, \sqrt{\lambda}I]^T w\|_2^2$. Therefore, if we add $p$ additional training

Table 45: **Performance of LIFT on Ridge regression.** We measure the RAE (↓) of LIFT corresponding to Linear Regression (LR) and Ridge Regression. The RAEs indicate that LIFT does not perform well on the Ridge regression problem.

| $p$ | $\lambda$ | LIFT/GPT-J | | LIFT/GPT-3 | |
|---|---|---|---|---|---|
| | | LR | Ridge | LR | Ridge |
| 1 | 0 | $0.000 \pm 0.000$ | $0.000 \pm 0.000$ | $0.915 \pm 0.000$ | $0.000 \pm 0.000$ |
| 1 | 10 | $0.000 \pm 0.000$ | $0.016 \pm 0.000$ | $0.915 \pm 0.000$ | $0.016 \pm 0.000$ |
| 1 | 50 | $0.000 \pm 0.000$ | $0.403 \pm 0.000$ | $0.915 \pm 0.000$ | $0.402 \pm 0.000$ |
| 1 | 100 | $0.000 \pm 0.000$ | $1.691 \pm 0.000$ | $0.915 \pm 0.000$ | $1.690 \pm 0.000$ |
| 1 | 1000 | $0.000 \pm 0.000$ | $170.406 \pm 0.000$ | $0.915 \pm 0.000$ | $170.612 \pm 0.000$ |
| 10 | 0 | $0.532 \pm 0.000$ | $0.532 \pm 0.000$ | $0.915 \pm 0.000$ | $0.521 \pm 0.000$ |
| 10 | 10 | $0.374 \pm 0.000$ | $0.369 \pm 0.000$ | $0.915 \pm 0.000$ | $0.504 \pm 0.000$ |
| 10 | 50 | $0.417 \pm 0.000$ | $0.523 \pm 0.000$ | $0.915 \pm 0.000$ | $0.563 \pm 0.000$ |
| 10 | 100 | $0.365 \pm 0.000$ | $1.307 \pm 0.000$ | $0.915 \pm 0.000$ | $1.539 \pm 0.000$ |
| 10 | 1000 | $0.414 \pm 0.000$ | $114.042 \pm 0.000$ | $0.915 \pm 0.000$ | $111.357 \pm 0.000$ |
| 50 | 0 | $0.688 \pm 0.000$ | $0.688 \pm 0.000$ | $0.915 \pm 0.000$ | $1.064 \pm 0.000$ |
| 50 | 10 | $0.628 \pm 0.000$ | $0.635 \pm 0.000$ | $0.915 \pm 0.000$ | $0.909 \pm 0.000$ |
| 50 | 50 | $0.553 \pm 0.000$ | $0.732 \pm 0.000$ | $0.915 \pm 0.000$ | $1.296 \pm 0.000$ |
| 50 | 100 | $0.774 \pm 0.000$ | $1.857 \pm 0.000$ | $0.915 \pm 0.000$ | $2.311 \pm 0.000$ |
| 50 | 1000 | $0.970 \pm 0.000$ | $118.241 \pm 0.000$ | $0.915 \pm 0.000$ | $133.122 \pm 0.000$ |

samples $\sqrt{\lambda}I$, one can perform ridge regression via data augmentation. Inspired by this, we study whether one can perform ridge regression via data augmentation within the framework of the LIFT framework. The results of LIFT on Ridge regression are reported in Table 45.

We observe that LIFT fails to perform Ridge regression. This is expected, as LIFT is shown to be robust to outliers (in Sec. 3.5 and Appendix D.1.4).

## E.5 LIFT's Training Curve

We report the learning curves of LIFT in terms of LM-loss and accuracies/RAE for several classification and regression tasks. We observe a decrease in training loss over the tasks and datasets. We select the best models based on the validation criteria (accuracy for classification and RAE for regression) on the validation sets. Fig. 46 visualize the accuracy and loss of LIFT/GPT-J in the training and validation process for classification tasks. For the regression task, Fig. 47 shows that the decrease in RAE does not necessarily imply a decrease in loss. Furthermore, we observe that LIFT only requires a few epochs to achieve good performance.

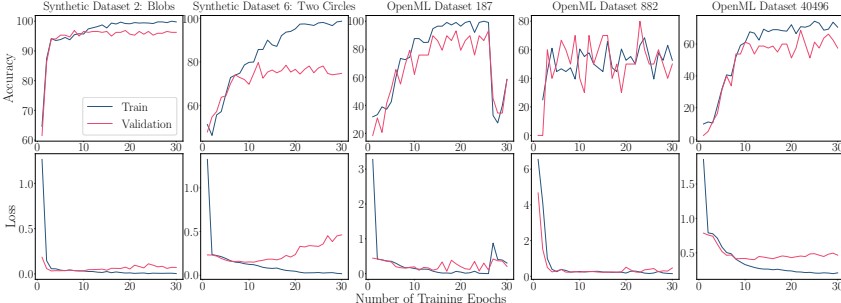

Figure 46: **Learning curves of LIFT/GPT-J on several synthetic/OpenML classification datasets.** We plot the accuracy (top row) and the loss (bottom row) of LIFT varying the number of training epochs.

## F   Additional Discussion

Continuing from Sec. 7, here we elaborate on the difficulty of regression tasks and the broader impact of LIFT, and discuss the limitation on classification tasks and other open questions.

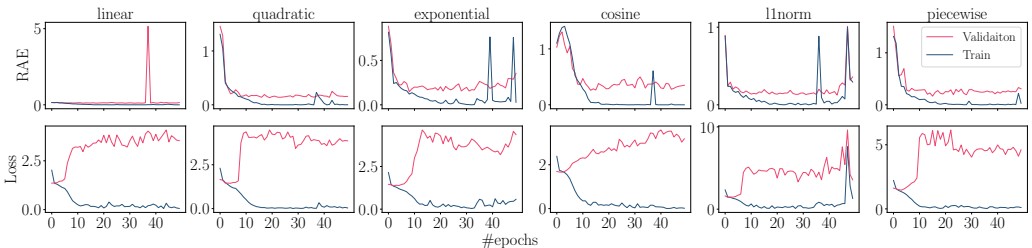

Figure 47: **Regression RAE and Loss curves of LIFT/GPT-J on the synthetic regression datasets.** We observe that LIFT/GPT-J only requires a few epochs to achieve good performance.

## F.1 Limitations and Open Questions

**The difficulty of regression tasks.** For regression tasks, in addition to poor performance on high-dimensional functions, some interesting phenomena observed in the classification tasks are not consistently observed in the regression tasks. For example, incorporating feature names in the prompts does not consistently improve LIFT in the regression tasks (see Sec. D.2.1).

As previously discussed in Sec. 3.1, the difficulty of regression tasks on LIFT may come from the classification loss function used in LMs. Due to the adoption of the classification loss function, two different predictions will lead to the same loss, even if one of the predictions is closer to the true $y$ value. As a result, we also observe that a reduction in RAE does not necessarily imply a reduction in LM loss (see Fig. 47). Therefore, we use RAE as the criterion for model selection. Moreover, how LIFT understands numerical values may also limit the regression performance of LIFT. Recent works [169, 134, 170, 171] have illustrated the difficulty and failures of the existing LMs in understanding the numbers because two numbers with close values can have very different tokenizations [134]. Recent attempts [132, 172, 173, 174, 175] propose new encoding schemes of numbers to improve the LMs' numerical capabilities, probably helping LIFT in the regression tasks.

A promising method for improving LIFT on regression is *level encoding*. The idea of level encoding is to discretize the continuous values of the output $y$ to better utilize the classification loss of LMs. Assuming that the range of $y$ is known, we can partition this range into a finite number of bins and represent all values in the same bin by a unique canonical representation in a way that the number of mismatched bits between the representations of two values is proportional to their absolute difference. For instance, for all real-value $y \in [0, 3]$, we can define three bins as $\{[0, 1), [1, 2), [2, 3]\}$ with the canonical representations being $00, 01, 11$. With these bins, $0.3$ and $0.7$ are represented as $00$, and $1.5$ and $1.1$ are represented as $01$. The distance between representations of $0.3$ and $1.1$ is only $1$ bit, which is proportional to their absolute distance of $0.8$. For the training of LIFT, we convert all output values in the original training dataset into the level-encoding canonical representation and use them as the target values. By using the level encoding technique, the loss function of LMs can better capture the distance between the prediction and the true values, thus potentially improving the generalization of LIFT on regression tasks. We leave this as one of the interesting directions for our future investigation.

**The limitation of LIFT on classification tasks.** We observe that LIFT does not perform comparably well on classification tasks when the number of classes is large. For instance, Table 4 shows that the accuracies of LIFT/GPT-3 are lower than RBF-SVM and XGboost on the datasets with 100 classes. Another limitation is that the dimension of features LIFT can handle is upper bounded due to the limited context length of LMs. This limitation may be mitigated by using LMs with a more memory-efficient variant or implementation of transformer models, *e.g.*, see [176].

**Other open questions.** In addition to previously discussed questions of improving LIFT for regression and classification tasks, our pioneering work on LIFT is also expected to open up interesting research questions on generalist models. First, can generalist LMs (*e.g.*, GPTs) play a leading role in developing universal models that can adapt well to any data? Second, can we apply LIFT to different generalist models, such as GATO [89]?

## F.2   Broader Impact

LIFT greatly simplifies the machine learning pipeline that requires only the reformatting of training datasets of the target task. This simplicity helps enable *no-code ML* for the masses, where general users without prior knowledge of the ML frameworks can use LIFT for their target non-language tasks by properly designing the input/output prompt format. Therefore, LIFT can apply to a wide range of applications and areas, such as credit loaning, disease diagnosis, and criminal sentencing. This is closely related to the line of automated machine learning research [90, 91], which aims to automate the standard machine learning methods pipeline.

Employing LIFT without careful justification or understanding will lead to undesired outcomes, such as discrimination. Since most existing language models (LMs) are pretrained on a large amount of human-annotated data, LIFT could exhibit discrimination against different demographic groups (*e.g.*, gender, race, ethnicity) due to the bias existing in the training datasets. In other words, LIFT may prefer certain groups while making decisions in downstream tasks, especially when feature names and different demographic contexts are fed at training and inference time. This effect is exacerbated by the use of large pretrained LMs (*i.e.*, GPT-J and GPT-3), which have been known to inherently contain bias [177]. The bias in the pretraining data for these large language models adds an opaque layer to regression and classification tasks beyond bias within the downstream data. Therefore, adopting LIFT in tasks that consider demographic information requires more consideration to avoid discrimination. To further remove the bias, users can combine LIFT with the existing fairness-aware reweighting mechanisms [178, 179] or data augmentation and parameter-efficient fine-tuning techniques [180].

Finally, we emphasize that more model evaluation steps are required when applying LIFT instead of using it as a panacea for all applications. We believe our work can significantly benefit society by providing a simple tool for handling various tasks with proper justification.