# OpenReview forum: "LIFT: Language-Interfaced Fine-Tuning for Non-language Machine Learning Tasks"
_NeurIPS.cc/2022/Conference — NeurIPS 2022 Accept_

### Official Review · Reviewer_igVi · 2022-06-17

**Rating:** 5
**Confidence:** 5
**Soundness:** 2 fair
**Presentation:** 1 poor
**Contribution:** 2 fair

**Summary:**

This paper aims to answer whether pre-trained language models (PLMs) can be used for fine-tuning on non-language downstream tasks without changing the input and output format of the original PLM. To answer the above question, this paper introduces a framework, Language-Interfaced FineTuning (LIFT), which formats non-language classification and regression tasks into natural language-like sentences by predefined templates. By extensive experiments on various datasets, they show that fine-tuning autoregressive PLMs on those downstream tasks can achieve relatively well performance. This paper also presents a series of analyses on the properties of LIFT.

**Questions:**

### Questions
1. Do the authors manually change the style file or change the font sizes of the table captions?
2. The sentence in Line 109 "For **tabular data**, ... non-language tasks" seems odd.
3. During inference, how is the answer generated? By argmaxing or sampling? If it is sampled, what is the sampling scheme? When will the generation end?
4. I am afraid that I will have to disagree with the claim on Line 219 that "LIFT/GPT outperforms in-context learning". This is because, in Table 10, the variances are very big. Considering the variance of both in-context learning and LIFT/GPT, a would rather say that there is no statistical significance between the performance of the two methods.
5. Why can the experiment results in Section 5.1 imply that LIFT "may" perform Bayesian inference?
6. There is an important reference that is missing: Kao and Lee (2021) [4] have already fine-tuned BERT on non-language downstream tasks as standard fine-tuning on natural language downstream tasks and show that the performance is decent.


[4] Wei-Tsung Kao and Hung-yi Lee. 2021. Is BERT a Cross-Disciplinary Knowledge Learner? A Surprising Finding of Pre-trained Models’ Transferability. In Findings of the Association for Computational Linguistics: EMNLP 2021, pages 2195–2208, Punta Cana, Dominican Republic. Association for Computational Linguistics.

**Limitations:**

The authors have addressed their limitations and potential social impacts.

**Strengths And Weaknesses:**

Update after rebuttal
===========
I have read all the comments by the authors, and they mostly clear up my questions on the experiment settings and formatting issues. While I agree that this paper has contribution to the ML community, I cannot lean strongly toward acceptance due to its massive content. I think the content is more suitable for a journal instead of a 10-page conference paper since a lot of details are placed in the appendix, making the main content not very self-contained. Also, I still feel like this paper is more of an empirical report and it is hard to know how LIFT can be used in real-life use cases.
***
***

### Strengths
1. This paper provides some interesting findings on directly fine-tuning PLMs on non-language downstream tasks.
2. The amount of downstream tasks evaluated in this paper is impressive.

### Weaknesses
(If I am misunderstanding any parts of the paper in the following parts, please let me know.)
1. **It is unclear whether the benefit of LIFT is just because the model is gigantic**
    * Since the goal of the paper is to answer "can fine-tuning PLMs work on non-language downstream tasks", it should be verified whether the exceptional performance of LIFT is really because *"LIFT fine-tunes a model **pre-trained on human language**"*. Chances are the good performance actually stems from *"LIFT uses a gigantic transformer model with billions of parameters, and it doesn't matter whether it is pre-trained on human language or not."* To exclude the latter case, the paper should include an experiment to train a transformer model with the same architecture of GPT-3/GPT-J from scratch on the downstream task, instead of initializing the weights from a PLM. Since the trained from-scratch model is not pre-trained with human language, if this model indeed fails to perform well on those non-language downstream tasks, we can rule out the hypothesis that "LIFT performs well simply because it uses a gigantic model". Next, one should also make sure that it is because of pre-training on **human language** that makes the success of LIFT. For example, one can use artificial datasets to train GPTs and fine-tune them on non-human language downstream tasks, as in [1], [2], and [3]. If training from scratch and fine-tuning from a model pre-trained with artificial datasets both result in bad non-language downstream performance, one will be able to arrive at the conclusion that "PLM fine-tuning" solves non-language downstream tasks.
2. I am **not convinced by the motivation** of "why one will want to use a PLM to fine-tune on non-language downstream tasks".
    * According to the paper, it seems that the goal is to achieve "no-code machine learning with language models". But I cannot see why that would be beneficial for two reasons. First, no-code machine learning can be achieved by the AutoML (Vertex AI by Google) without relying on PLM. It is unclear why one would want to rely on a bulky PLM to achieve no-code machine learning. Second, as illustrated in the experiment results, the performance of LIFT is still largely inferior to other simpler and smaller machine learning models; this will refrain LIFT from being used in real-world cases. It is unclear why one will resort to a larger and not always better model while one can just use smaller and better models (for example, always use logistic regression for tabular data and always use MLP for other classification tasks).
3. The input length of LIFT is upper-bounded by the maximum sequence length of the PLM (2048 for GPT-J and GPT3), making it hard to be used for high-dimensional data. Moreover, considering that the input of LIFT is not only the raw feature but also the words in the templates (
$\mathbf{x}_1 = v_1 $
what should be the
$\mathbf{y}$
), the actual acceptable feature length will be significantly less than 2048. It will be even lesser if each feature $v_i$ is tokenized to multiple tokens by the model. For example, considering MNIST, which has 784 features per image, assuming that each feature will only be tokenized to one token. Then when encoding an image using the template $\mathbf{x}_i=v_i$,  it will require at least $784\times4=3136$ tokens, which is significantly longer than the maximum input length of the GPT models used in the paper. It is unclear how this is handled in the paper.
4. The memory of LIFT is quadratic to the input length, indicating that LIFT is highly memory inefficient compared to other models.
4. This paper tries to include too many things in the main content, making the snippets included in the main content seem fragmented, incomplete, not self-contained, and hard to follow. The following lists are pieces of evidence from the paper to justify the above claim.
    1. The "non-deterministic predictions" that appeared in Line 18 and Line 70 do not appear in the rest of the paper, and it is unclear what "non-deterministic predictions" refers to? Is it related to the Bayesian prediction in Section 5.1?
    2. How is the "increase the generation randomness" in Line 137 done? Perhaps by changing the temperature for sampling?
    3. What do the "inductive biases of language models" refer to in Section 4.2?
    4. What is a median kNN in Line 241?
    5. The paragraph of "Robustness to adversarial samples" in Line 244 includes no experiment results, making the paper not very self-contained.
    6. It is unclear how the mixup is done in Section 6.2, the details of this experiment is not presented neither in the main content nor the appendix.
    7. It is unclear how the in-context learning is done in Section 4.3. For example, how is the training/testing split determined? Is there an evaluation set? Where does the variance shown in Table 10 come from?
5. This paper is poorly formatted, and some of the figures are extremely small for reading.
    1. Ill-formatted parts
        * According to the formatting instructions for NeurIPS 2022, "the table number and title always appear before the table"; all caption of the table in the paper appears **after** the table.
        * The size of the caption is sometimes significantly smaller than what should be expected, for example, Table 3, Table 5, Table 6, and Table 7. I am not sure whether this violates the guideline of "not changing the font sizes except for the reference".
        * The bottom of Figure 7 on page 8 is occupying part of Line 329 on page 9.
        * The figures in Table 3 are extremely small.
        * The legends in Figure 4 and Figure 5 are too small to read when printed on A4 paper.
        * According to the formatting instructions for NeurIPS 2022, "Place one line space before the table title, one line space after the table title, and one line space after the table". But the margin of Table 3 is quite odd and seems to violate the instructions.


[1]Papadimitriou, Isabel, and Dan Jurafsky. "Learning Music Helps You Read: Using Transfer to Study Linguistic Structure in Language Models." Proceedings of the 2020 Conference on Empirical Methods in Natural Language Processing (EMNLP). 2020.

[2] Chiang, Cheng-Han, and Hung-yi Lee. "On the transferability of pre-trained language models: A study from artificial datasets." arXiv preprint arXiv:2109.03537 (2021).

[3] Ryokan Ri and Yoshimasa Tsuruoka. 2022. Pretraining with Artificial Language: Studying Transferable Knowledge in Language Models. In Proceedings of the 60th Annual Meeting of the Association for Computational Linguistics (Volume 1: Long Papers), pages 7302–7315, Dublin, Ireland. Association for Computational Linguistics.

---

> ### Author Response · Authors · 2022-08-02
> **To Reviewer igVi**
>
> We would like to thank the Reviewer for your constructive comments as well as the appreciation of our interesting findings and the impressive amount of downstream tasks in experiments.
>
> **`Q. It is unclear whether the benefit of LIFT is just because the model is gigantic.`**
>
> As per your great question, we ran extensive experiments to better explain the reasons for the good performance of LIFT. Please find our response to this question in our response to Q2 in the response to all reviewers & AC. In short, we observe that gigantic language models that are not pretrained in human language do not necessarily lead to as good performance as language models that are pretrained in human languages. Therefore, we conclude that the gigantic model itself is not the only reason for the good performance of LIFT.
>
>
> **`Q. Why one will want to use a PLM to fine-tune on non-language downstream tasks.`**
>
> Thanks for your question. Please find our response to this question in our response to Q1 in the response to all reviewers & AC.
>
> **`Q. Dependency of LIFT on the input length of LMs.`**
>
>
> Thanks for the great question. In the updated version, we acknowledge this limitation and the dependency of LIFT on the chosen language model. We provide two techniques for mitigating this issue.
> 1. **Subsampling the features and keeping less number of digits.** For example, to handle MNIST, we (i) crop each image and keep the center with a size of 18x18, and (ii) use the integer representation of pixel (from 0 to 255). So, each image totally has 384 pixels and fits in the context length of the input.
> 2. **Leveraging long-range Transformer-based language models.**
> There are LMs with long context lengths based on the long-range transformer architectures, such as Transformer-XL (Dai et al., 2019), Longformer (Beltagy et al., 2020), BigBird (Zaheer et al., 2020), or Perceiver-IO (Jaegle et al., 2021).
>
> *References:*
> * Zaheer, M., Guruganesh, G., Dubey, K. A., Ainslie, J., Alberti, C., Ontanon, S., ... & Ahmed, A. (2020). Big bird: Transformers for longer sequences. Advances in Neural Information Processing Systems, 33, 17283-17297.
> * Beltagy, I., Peters, M. E., & Cohan, A. (2020). Longformer: The long-document transformer. arXiv preprint arXiv:2004.05150.
> * Dai, Z., Yang, Z., Yang, Y., Carbonell, J., Le, Q. V., & Salakhutdinov, R. (2019). Transformer-xl: Attentive language models beyond a fixed-length context. arXiv preprint arXiv:1901.02860.
> * Jaegle, Andrew and Borgeaud, Sebastian and Alayrac, Jean-Baptiste and Doersch, Carl and Ionescu, Catalin and Ding, David and Koppula, Skanda and Zoran, Daniel and Brock, Andrew and Shelhamer, Evan and others. Perceiver io: A general architecture for structured inputs and outputs. arXiv:2107.14795, 2021.
>
> **`Q: The memory of LIFT is quadratic to the input length, indicating that LIFT is highly memory inefficient compared to other models.`**
>
> We thank the reviewer for the detailed comment. The reviewer is correct that the memory of LIFT is quadratic to the input length, given the use of the attention mechanism in GPT-J and GPT-3. To mitigate the memory issues of transformers, we can make use of LMs with a more memory-efficient variant or implementation of transformer models, e.g., Flash Attention (Dao et al., 2022) or with efficient or sparse attention methods (Linformer (Wang et al., 2020), BigBird (Zaheer et al., 2020) or ETC (Ainslie, et al., 2020) ) to reduce the quadratic dependency of models on the input length to linear.
>
> *References*:
> * Zaheer, M., Guruganesh, G., Dubey, K. A., Ainslie, J., Alberti, C., Ontanon, S., ... & Ahmed, A. (2020). Big bird: Transformers for longer sequences. Advances in Neural Information Processing Systems, 33, 17283-17297.
> * Ainslie, J., Ontanon, S., Alberti, C., Cvicek, V., Fisher, Z., Pham, P., ... & Yang, L. (2020). ETC: Encoding long and structured inputs in transformers. arXiv preprint arXiv:2004.08483.
> * Wang, S., Li, B. Z., Khabsa, M., Fang, H., & Ma, H. (2020). Linformer: Self-attention with linear complexity. arXiv preprint arXiv:2006.04768.
> * Dao, T., Fu, D. Y., Ermon, S., Rudra, A., & Ré, C. (2022). FlashAttention: Fast and Memory-Efficient Exact Attention with IO-Awareness. arXiv preprint arXiv:2205.14135.

---

> > ### Author Response · Authors · 2022-08-02
> > **To Reviewer igVi: Continued**
> >
> > **`Q. The main content seems fragmented, incomplete, not self-contained, and hard to follow.`**
> >
> > 1. The "non-deterministic predictions" that appeared in Line 18 and Line 70 do not appear in the rest of the paper, and it is unclear what "non-deterministic predictions" refers to. Is it related to the Bayesian prediction in Section 5.1?**
> >
> >     Yes, the "non-deterministic predictions" refers to the uncertainty of predictions evaluated in Section 5.1. We show that if we allow non-deterministic predictions of LIFT by setting non-zero [`temperature`](https://beta.openai.com/docs/quickstart/add-some-examples), the prediction variance reflects its prediction confidence. We will reword the terminology into "calibration", which is a more precise terminology for this property.
> >
> >
> > 2. How is the "increase the generation randomness" in Line 137 done? Perhaps by changing the temperature for sampling?**
> >
> >     The reviewer is correct that we control the generation randomness by changing the [`temperature`](https://beta.openai.com/docs/quickstart/add-some-examples) for sampling.
> >
> > 3. What do the "inductive biases of language models" refer to in Section 4.2?
> >
> >     The inductive biases of language models refer to the preference bias (axis-parallel decision boundaries) and behavior of language models in making the prediction.
> >
> >
> > 4. What is a median kNN in Line 241?
> >
> >     The standard kNN makes predictions by taking the *mean* of the sample's $k$ nearest neighbors' predictions. Median kNN here refers to predicting with the *median* instead of *mean*, which is more robust.
> >
> > 5. The paragraph "Robustness to adversarial samples" in Line 244 includes no experiment results, making the paper not very self-contained.
> >
> >
> >     In the revised manuscript, we moved the results in Section C.1.1 of the Appendix to the main body.
> >
> >
> > 6. It is unclear how the mixup is done in Section 6.2, the details of this experiment are not presented neither in the main content nor the appendix.
> >
> >     We apply mixup on LIFT for binary classification problems as below. Given each sample $(x,y)$ having feature $x$ and label $y \in \{0,1\}$, we convert it to $(x, \tilde{y})$ for $\tilde{y} = 10y$, so that the label is either 0 or 10. Given $(x_1, \tilde{y}_1)$ and $(x_2, \tilde{y}_2)$, we make mixup sample $(x^{\text{mix}}, y^{\text{mix}}) = (\lambda x_1 + (1-\lambda )x_2, \lambda \tilde{y}_1 + (1-\lambda )\tilde{y}_2)$, where the convex combination coefficient $\lambda \in \{0, 0.1, 0.2, \cdots, 1.0\}$. In this way, we set the mixup label to be integer, \textit{i.e.,} $y^{\text{mix}} \in \{0, 1, 2, \cdots, 10\}$. We fine-tuned GPT using the input-output pair $(x^{\text{mix}}, y^{\text{mix}})$. We included this detail in the revised manuscript.
> >
> >
> >
> >
> > 6. It is unclear how the in-context learning is done in Section 4.3. For example, how is the training/testing split determined? Is there an evaluation set? Where does the variance shown in Table 10 come from?
> >
> >     We used the same split of training/validation/testing all the time for all datasets used in different tasks. In in-contexting learning, we randomly select input-output pairs from the training dataset and concatenated them in front of each testing sample. We used a fixed group of 3 random seeds and run in-context learning on each dataset 3 times. We took average and STD on all 3 runs. We did not use the validation dataset in this case and all results reported were based on the testing dataset.
> >
> > We have added the clarification above to our updated manuscript.
> >
> > **`Q. This paper is poorly formatted, and some of the figures are extremely small for reading.`**
> >
> > We acknowledge the formatting issues in the last few pages of our paper. We have revised the paper to make it properly formatted.
> >
> >
> > **`Q: Do the authors manually change the style file or change the font sizes of the table captions?`**
> >
> > No, we didn't change the style file. We only used some latex commands (e.g.,`\wrapfigure`) to save the space and `\resize` the tables to fit them horizontally on a page. Table captions are below the table because we used the `\caption` command after the tabular data. We updated them in the new version.
> >
> > **`Q. The sentence in Line 109 "For tabular data, ... non-language tasks" seems odd.`**
> >
> > We revise this sentence to be more clear, as "To the best of our knowledge, we are one of the first works studying the adaptation of LMs for non-language tasks on tabular data".
> >
> > **`Q.During inference, how is the answer generated? By argmaxing or sampling? If it is sampled, what is the sampling scheme? When will the generation end?`**
> >
> > How the answer is generated depends on the LM we are using. For GPT, the answer is generated via a sampling mechanism, while the generation randomness is controlled by [`temperature`](https://beta.openai.com/docs/quickstart/add-some-examples). GPT will generate `max_tokens` tokens and then stop.

---

> > > ### Author Response · Authors · 2022-08-02
> > > **To Reviewer igVi: Continued**
> > >
> > > **`Q. I am afraid that I will have to disagree with the claim on Line 219 that "LIFT/GPT outperforms in-context learning".`**
> > >
> > > We agree that the provided result shows that 'subset-LIFT' (LIFT performed on the subset of data) has no significant statistical improvements with in-context learning.
> > > However, compared to in-context learning, the performance of LIFT on the full data is almost always staticially better.
> > > Here, we provide the updated result with in-context learning to support this claim.
> > >
> > >  We include three training schemes:
> > > * `In-context learning`: few-shot prompting;
> > > * `LIFT/Subset`: LIFT on the subset of data used in the corresponding in-context learning setting;
> > > * `LIFT/Full-data`: LIFT on the full dataset.
> > >
> > > | Dataset (ID)     | #Prompts | ODC   | GPT-J       | GPT-J       | GPT-J          | GPT-3      | GPT-3       | GPT-3          |
> > > |------------------|----------|-------|-------------|-------------|----------------|------------|-------------|----------------|
> > > |                  |          |       | In-Context  | LIFT/ Subset | LIFT/ Full-data | In-Context | LIFT/ Subset | LIFT/ Full-data |
> > > | Breast (13)      | 35       | 70.69 | 56.90±19.51 | 58.62±2.44  | 64.94±11.97    | 62.07±1.41 | 70.69±0.00  | 71.26±1.62     |
> > > | TAE (48)         | 50       | 35.48 | 34.33±1.47  | 32.26±9.50  | 61.29±4.56     | 37.64±4.02 | 33.33±1.52  | 65.59±6.63     |
> > > | Vehicle (54)     | 14       | 25.88 | 25.49±0.55  | 26.04±1.69  | 64.31±2.37     | 28.82±2.10 | 23.73±2.27  | 70.20±2.73     |
> > > | Hamster (893)    | 43       | 53.33 | 48.89±3.14  | 60.00±10.88 | 55.55±16.63    | 57.78±6.29 | 53.33±0.00  | 53.33±0.00     |
> > > | Customers (1511) | 29       | 68.18 | 56.06±17.14 | 59.85±2.84  | 85.23±1.61     | 60.61±1.42 | 63.26±6.96  | 84.85±1.42     |
> > > | LED (40496)      | 33       | 68.67 | 10.00±0.82  | 13.04±3.27  | 65.33±0.47     | 8.00±1.63  | 11.33±2.62  | 69.33±2.05     |
> > >
> > > **`Q. There is an important reference that is missing: Kao and Lee (2021) [4] have already fine-tuned BERT on non-language downstream tasks as standard fine-tuning on natural language downstream tasks and show that the performance is decent.`**
> > >
> > > Thanks for bringing this work to our attention. We note that this work adds a linear layer on top of pretrained BERT for fine-tuning, which requires architectural changes that are not needed for LIFT. We have added this work to the related work.
> > >
> > > -----
> > >
> > > **Final notes:** We hope that our responses and new experimental results on (i) LIFT/Rand-GPT-J, (ii) LIFT/`LM pretrained on non-human languages`, and (iii) improving the performance of LIFT by utilizing feature names can help you better appreciate our work as well as considering increasing your score and supporting accepting our paper. We agree that LIFT has some limitations, but with its novelty, significance, and potential for future research, we believe LIFT can have a good impact on the machine learning society. Thanks again for your careful reading!

---

> > > > ### Comment · Reviewer_igVi · 2022-08-04
> > > > **Update score to 5**
> > > >
> > > > I have read all the comments by the authors, and they mostly clear up my questions on the experiment settings and formatting issues. While I agree that this paper has contribution to the ML community, I cannot lean strongly toward acceptance due to its massive content. I think the content is more suitable for a journal instead of a 10-page conference paper since a lot of details are placed in the appendix, making the main content not very self-contained. Also, I still feel like this paper is more of an empirical report and it is hard to know how LIFT can be used in real-life use cases.

---

> > > > > ### Author Response · Authors · 2022-08-08
> > > > > **Re: Update score to 5**
> > > > >
> > > > > We thank the Reviewer for increasing the score.
> > > > > We appreciate your valuable feedbacks for our work and we will happily keep improving our paper.
> > > > > Best,

---

### Official Review · Reviewer_iBBJ · 2022-06-24

**Rating:** 7
**Confidence:** 4
**Soundness:** 3 good
**Presentation:** 4 excellent
**Contribution:** 3 good

**Summary:**

This paper studies an interesting and important topic: can we use large pre-trained language models to machine learning tasks where the inputs are unstructured real vectors? Through a series of experiments on real/synthetic classification/regression tasks, the author concludes that
- fine-tuning LLMs have performance on par with many traditional algorithms.
- LLM is performing bayesian prediction
- two-stage fine-tuning improves the performance
- feature name helps LLM perform better.

**Questions:**

- Table 2: why does the paper claim that LIFT performs relatively well compared to other methods, even if in many or even most cases it cannot outperform 3-layer MLP? It does not seem that the algorithm is successfully making use of the LLM internal representation.
- Table 3 caption tries to conclude that larger models generally perform better -- but it seems that GPT-3 Davinci is overall worse than GPT-3 Curie, and the performance increase w.r.t. scaling is extremely unclear.
- Section 5.1 tries to claim that LLM is performing bayesian inference. I thought it meant "LLM has a prior about what the function is look like and it performs posterior inference based on true labels". However, the paper only experimented with "trying sampling a lot of time and see whether the variance is correlated with noise level". This only implies that LIFT is calibrated (where prediction logits/samples reflects the confidence), rather than that LLM is performing bayesian inference.
- Table 5 tries to claim that LLM can successfully use the feature name -- for only 2 out of 3 classification dataset this seems to hold, and when it holds, the margin is very small (compared to the noise) and it is unclear whether they are noise. In particular, what is dataset 48 and 54, and what are their intuitions of LIFT can help? One experiment you could potentially run is to find a dataset where we have very strong. prior notion of what features are good and test the models' behavior on that as well.


**Limitations:**

Yes, the author successfully addressed their limitations.

**Strengths And Weaknesses:**

Strength:
 - The paper is generally very well-written, with clear figures and a lot of experimental details in the appendix. It also addresses its scope of study and societal impact properly.
- The topic is interesting, potentially impactful, and novel.
- There are a lot of experimental results, which could help practitioners gain intuitions.

Weakness:
- The performance by LIFT is not impressive.
- Most of the important claims in the paper are not well-supported by the empirical evidence. See questions below.

**I am very willing to increase the score to 8 if the author can either 1) justify why their claims are correct, or 2) re-position the paper as a "negative empirical finding", rather than advertising "LIFT" as a very promising approach for real-value-input ML tasks.**

minor:
- Non-language can also refer to image, sound, or haptic information, and I was confused what non-language task the paper was discussing when reading the title and the abstract. Probably rename "Non-Language Machine Learning Tasks" to "Real-Valued Machine Learning Tasks" (or something similar). For example, Flamingo (from DeepMind) also uses LM as an interface, but they used it for image + text.

---

> ### Author Response · Authors · 2022-08-02
> **To Reviewer iBBJ**
>
> We thank  Reviewer iBBJ for the detailed review and very helpful suggestions.
> We thank the Reviewer for appreciating the importance, novelty, and impact of our research problem, extensive experimental results, and our efforts in making a well-written paper.
> Our responses are detailed below.
>
>
> **``Q. The performance of LIFT is not impressive. Why does the paper claim that LIFT performs relatively well compared to other methods?``**
>
> We wrote "LIFT performs well" because we found that it is ranked within the top three approaches among all the tested baselines on average. That being said, we agree with the reviewer that we could tone down our claim by making claims more precise and specific.
>
> Besides, we would like to note that achieving impressive performance is not the only goal of LIFT. We believe that LIFT has a lot of potentials to be useful -- please find our response in Q1 of the common response.
>
> **`Q. Can LM successfully use the feature names?`**
>
> We thank the reviewer for the sharp question. To better support our claim, we have expanded the corresponding experiments. We compare LIFT when feature names are correctly incorporated (`Correct-Names`) with the versions of LIFT when feature names are incorrectly incorporated with randomly shuffled orders (`Shuffled-Names`).
> Also, we evaluate the performance of LIFT without feature names specified (`Without Names`). (In fact, we tried two different templates for each method in our revision to reduce the unwanted effect of prompt designs -- please see the revision for more detailed comparisons.) The results are reported in the Table below.
>
> | Dataset \ Schemes | `Without Names` |  `Shuffled-Names` | `Correct-Names` |
> |:-----------------:|:-----------:|:-----------:|:-----------:|
> |CMC| $\mathbf{57.74 \pm 0.89}$  | $57.06 \pm 4.24$ | $57.40 \pm 1.09$ |
> |TAE| $65.59 \pm 6.63$ | $64.52 \pm 8.53$ | $\mathbf{69.89 \pm 9.31}$ |
> |Vehicle| $70.20 \pm 2.73$ | $69.22 \pm 2.72$ | $\mathbf{75.29 \pm 2.04}$ |
>
> The new experiment results support our claim that LM can successfully use the feature names.
>
> **`Q. Table 3 caption tries to conclude that larger models generally perform better.`**
>
>
> Thanks for pointing this out. To clarify, our purpose was to show that: compared to the results with GPT-J and GPT-3-Ada (smallest version) used throughout the paper, the use of larger versions of GPT-3 (3 models) may improve the performance of LIFT. We did not mean to claim the comparison between the three larger versions of GPT-3. Based on the reviewer's comment, we have revised the caption to make it more clear.
>
> **`Q.  Why does LM perform bayesian inference? ... This only implies that LIFT is calibrated.`**
>
> We thank the Reviewer for pointing this out. We have replaced the conclusion that "LM is performing Bayesian inference" to "LIFT is calibrated" as per the reviewer's comment.
> Also, we have replaced the term "Bayesian inference" with "calibration" in our paper.
>
> **`Q. Rename "non-language machine learning tasks" to "real-valued machine learning tasks"?`**
>
> Thanks for engaging so deeply! Note that the datasets LIFT handles are not necessarily real-valued. For example, the categorical attributes are not real-valued, and we do not process the categorical attributes to be real-valued. To resolve the reviewer's concern, we have elaborated on what non-language tasks are earlier in the paper, including both the abstract and introduction.
>
> -----
> **Comment. '`I am very willing to increase the score to 8 if the author can either 1) justify why their claims are correct, or 2) re-position the paper as a "negative empirical finding", rather than advertising "LIFT" as a very promising approach for real-value-input ML tasks.`'**
>
> Thanks for the suggestion and your willingness to update the score based on our response. In the revision, we have toned down most claims with more fair and accurate claims. We believe that with our modification of writing and our new experimental results, we have successfully justified the correctness of all our updated claims. Moreover, we will add "negative empirical findings" on regression tasks and emphasize various limitations of LIFT. We hope that our revision can help you reconsider your score and support accepting our paper. Thanks again for your careful reading of our work and for providing valuable feedback!

---

> > ### Comment · Reviewer_iBBJ · 2022-08-03
> > **I support acceptance after revision; updated my score to 7.**
> >
> > Thanks for incorporating the comment and tuning down the claims. I support acceptance after revision and updated my score to 7.

---

> > > ### Author Response · Authors · 2022-08-08
> > > **RE: I support acceptance after revision; updated my score to 7.**
> > >
> > > We thank the Reviewer and really appreciate your support for our paper.

---

### Official Review · Reviewer_GQen · 2022-07-12

**Rating:** 6
**Confidence:** 4
**Soundness:** 3 good
**Presentation:** 2 fair
**Contribution:** 3 good

**Summary:**

This paper fine-tunes pre-trained language models to perform non-language tasks, like regression of functions and classification. They test a large variety of tasks, and test against several standard algorithms ( like KNN, SVM, RF). Additionally, they investigate sample efficiency, ability to extrapolate, and robustness of these models.

**Questions:**

* Did the authors try other templates? How did they perform? What was the sensitivity of the model performance on different prompts?
* From figure 4, it looks like this might not work for few-shot prompting. Did you run that experiment?
* I would include additional information about how adversarial examples were generated.

**Limitations:**

Currently the social impacts section says that "LIFT might have embedded bias targeting certain social groups." This seems possible to measure. In particular, for tasks where text is used in the prompt, the authors could compare LIFT with all other models they compare with to see if the examples that are mis-classified are systemically biased in some way. Additionally, this could be an interesting experimental setup to evaluate bias in the pre-trained model.

**Strengths And Weaknesses:**

### Strengths

Section 5.2 is the most interesting part of this paper to me, and to me, this is one of the best motivations for fine-tuning a large language model to perform this task. I haven't seen this idea before, and it's intriguing as a method for understanding the pre-training dataset and also harnessing contextual information. I'd like to see more focus directed here and maybe additional tasks explored or an expansion on experiments in this direction.


* I appreciate the summary listed in table 1!
* I appreciated figure 4 (comparing # training samples vs. accuracy for different classes of tasks).

### Weaknesses
I don't doubt the results of this work, but after having read the paper, I'm still unsure what advantage we gain by finetuning a pre-trained language model to perform non-language tasks. It currently reads like the paper makes the argument that finetuning pre-trained language models can perform well on non-language tasks. However, these models are orders of magnitude more costly to train and run inference on, so when and why should we prefer it? Could the authors better motivate this?

Additionally, please explain what non-language tasks are earlier in the paper. Specifically, also say that the paper will test both classification and regression tasks. Please give an example of each in the abstract and also in the intro.
Also, the sentence: "We note that though achieving tremendous success with natural data, deep learning still faces difficulties with standard machine learning tasks, especially on tabular data..." seems to be at the core of this paper. I would include this idea earlier in the paper (abstract or intro).


This paper covers a lot of ground. I think it could be more focused by moving additional experiments to the appendix (for example, sec 6.2).

### Additional Feedback
* Suggestion to use "what should the y be" as opposed to "what should be the y" in the template. Language models are still fairly sensitive to prompts, and wording may change results.
* Include the sizes of GPT-J (6b) and GPT-3 (175b) in the section on "pretrained language models" as a reference for the reader.
* Figure 4 text should be larger
    * Also table 3 should highlight interesting results as the text is too small to read
* It would make the paper easier to read if the authors could include the names of the tasks (in addition to the IDs) in the text to better understand the context of the task. For example, in section 5.2, which DID is the Medical Insurance dataset?

---

> ### Author Response · Authors · 2022-08-02
> **To Reviewer GQen**
>
> We thank  Reviewer GQen for the detailed review and very helpful suggestions. We appreciate that Reviewer GQen finds our idea and the use of feature names novel and interesting.
> Making use of feature names with LMs is exactly our very first motivation for this work.
> We also thank the Reviewer for appreciating our presentation in Table 1 and Figure 4.
>
> **`Q. It's intriguing as a method for understanding the pre-training dataset and also harnessing contextual information. I'd like to see more focus and additional tasks explored or an expansion on experiments on using the feature names with LMs.`**
>
> As per your great suggestion, we have expanded this section with more experiment results and provide stronger evidence that including feature names in the prompts indeed improves the performance of LIFT. We compare LIFT when feature names are correctly incorporated (`Correct-Names`) with the version of LIFT when feature names are incorrectly incorporated with randomly shuffled orders (`Shuffled-Names`). Also, we evaluate the performance of LIFT without feature names specified (`Without Names`). (In fact, we tried two different templates for each method in our revision to reduce the unwanted effect of prompt designs -- please see the revision for more detailed comparisons.)
>
>
> | Dataset (DID) \ Schemes | `Without Names` |  `Shuffled-Names II` | `Correct-Names` |
> |:-----------------:|:-----------:|:-----------:|:-----------:|
> |CMC (23)| $\mathbf{57.74 \pm 0.89}$ |  $57.06 \pm 4.24$ | $57.40 \pm 1.09$ |
> |TAE (48)| $65.59 \pm 6.63$ | $64.52 \pm 8.53$ | $\mathbf{69.89 \pm 9.31}$ |
> |Vehicle (54)| $70.20 \pm 2.73$ |  $69.22 \pm 2.72$ | $\mathbf{75.29 \pm 2.04}$ |
>
>
> As we can see from the table, the performance of LIFT is maximized when feature names are correctly specified. We plan to expand this analysis and include a more extensive analysis in the final version.
>
>
> **`Q. "what should the y be" vs "what should be the y"`**
>
> | Dataset (DID) \ Schemes | `Original template` (what should the y be) | `New template` (what should be the y) |
> |:-----------------:|:-----------:|:-----------:|
> |CMC (23)| $57.74 \pm 0.89$ |$57.40 \pm 1.37$ |
> |TAE (48)| $65.59 \pm 6.63$ | $66.67 \pm 5.48$ |
> |Vehicle (54)| $70.20 \pm 2.73$ |$71.96 \pm 3.09$ |
>
> We thank the reviewer for an interesting suggestion. The reviewer is absolutely correct that small changes in prompt design can make a large difference in the overall performance when using large language models. We tried your particular suggestion, and we saw some (yet marginal) differences in our experiments. Inspired by your suggestion, we plan to further investigate the effect of prompt design on the performance of LIFT and include a more thorough analysis in the final version.
>
>
>
> **`Q. The motivation of LIFT: when and why should we prefer it.`**
>
> Please find our response to Q1 in our common response.
>
>
> **`Q. Suggestions on the writing.`**
>
> Thanks for your helpful suggestions, which aided us greatly in improving the clarity of our writing. We have properly made corresponding edits to address each of your comments. We (1) added an early explanation of non-language tasks and the core idea, (2) added detailed descriptions and pretrained language models and keep only concise and important methods in figures, and tables to make them more readable, (3) added the dataset names, and (4) moved a lot of less significant experiments to the appendix.

---

> > ### Author Response · Authors · 2022-08-02
> > **To Reviewer GQen: continued**
> >
> > **`Q. From figure 4, it looks like this might not work for few-shot prompting. Did you run that experiment?`**
> >
> > Thanks for a sharp observation. Yes, we observe that few-shot prompting also does not work well in these cases. In the following table, we report the performance of three schemes:
> > * `In-context learning`: few-shot prompting;
> > * `LIFT/Subset`: LIFT trained with few shots;
> > * `LIFT/Full-data`: LIFT trained with the full dataset.
> >
> > | Dataset (ID)     | # Shots | ODC   | GPT-J       | GPT-J       | GPT-J          | GPT-3      | GPT-3       | GPT-3          |
> > |------------------|----------|-------|-------------|-------------|----------------|------------|-------------|----------------|
> > |                  |          |       | In-Context  | LIFT/ Subset | LIFT/ Full-data | In-Context | LIFT/ Subset | LIFT/ Full-data |
> > | Breast (13)      | 35       | 70.69 | 56.90±19.51 | 58.62±2.44  | 64.94±11.97    | 62.07±1.41 | 70.69±0.00  | 71.26±1.62     |
> > | TAE (48)         | 50       | 35.48 | 34.33±1.47  | 32.26±9.50  | 61.29±4.56     | 37.64±4.02 | 33.33±1.52  | 65.59±6.63     |
> > | Vehicle (54)     | 14       | 25.88 | 25.49±0.55  | 26.04±1.69  | 64.31±2.37     | 28.82±2.10 | 23.73±2.27  | 70.20±2.73     |
> > | Hamster (893)    | 43       | 53.33 | 48.89±3.14  | 60.00±10.88 | 55.55±16.63    | 57.78±6.29 | 53.33±0.00  | 53.33±0.00     |
> > | Customers (1511) | 29       | 68.18 | 56.06±17.14 | 59.85±2.84  | 85.23±1.61     | 60.61±1.42 | 63.26±6.96  | 84.85±1.42     |
> > | LED (40496)      | 33       | 68.67 | 10.00±0.82  | 13.04±3.27  | 65.33±0.47     | 8.00±1.63  | 11.33±2.62  | 69.33±2.05     |
> >
> > As one can see, the performance of few-shot prompting is similar to (or slightly worse than) the performance of LIFT trained with the same number of examples.
> >
> > **`Q. How are adversarial examples generated? `**
> >
> > We generated adversarial examples using a proxy network (LeNet-5 or MLP) and transferred them. Please find Section C.1.1 of the Appendix for more details.
> >
> >
> >
> > **`Q. Measuring the embedded bias in LIFT.`**
> >
> > Thanks for engaging so deeply and providing such a good suggestion! The reviewer is correct that language models are known to be biased, so the classifier obtained with LIFT could be also biased. In fact, we also observed a few cases where LIFT generates biased predictions (and explanations as well) in our own experiments. We plan to further explore this bias issue of LIFT in future work while leaving a remark on this issue in the current paper.
> >
> > ----
> > **Final notes**: We want to thank you again for such exemplary and detailed comments. We are excited that you find our idea novel and hope that our response makes you consider increasing your score and further support accepting our paper.

---

### Official Review · Reviewer_evpg · 2022-07-18

**Rating:** 7
**Confidence:** 4
**Soundness:** 3 good
**Presentation:** 4 excellent
**Contribution:** 3 good

**Summary:**

The paper investiagtes the application of pre-trained language models on the task of classifying non-textual data, without any architecture changes.
The input features are linearized into a text-like sequence and given as context, the output is then collected as a language model prediction.
While it doesn'g perform best overall, it does achieve surprisingly competitive performance.

**Questions:**

^

**Limitations:**

Yes

**Strengths And Weaknesses:**

The paper is well written and clear.
The direction of investigation is interesting.
There are many experiments presented, analysing different aspects of the set-up.
The findings are interesting and non-intuitive. For example, the model performing well on MNIST by describing each pixel in a textual sequence.
While the setup is clearly not optimal for the tasks, the findings could lead to some interesting future research.

Previous work is mentioned, which replaces the input and output layers of the LM, whereas the current work keeps entirely the same architecture.
The performance of that system should be reported as well.
At the moment it is unclear whether keeping the original input or output layers actually helps or hurts the performance on different tasks.

Furthermore, the replacement of the input and output layerst should be investigated separately.
While giving the input as text might actually have some benefits (as experiments in Section 5.2 also indicate), using only language model predictions as output could be hurting performance. Even models that work with natural language generally fine-tune pre-trained language models by replacing the LM output layer with a task-specific version.

An open question left by the paper is whether these results are down to the architecture (and the large number of parameters) or the fact that the model is pre-trained on large amounts of textual data.

There should also be a discussion on the difference in the number of parameters.
As I understand, all the baseline models are quite simple, with perhaps a few hundred trainable parameters. In contrast, GPT-3 has billions of parameters.
How much does model capacity affect performance? Are there no baselines with more comparable numbers of parameters that could have been also reported?

---

> ### Author Response · Authors · 2022-08-02
> **To Reviewer evpg**
>
> We thank the Reviewer evpg for the detailed review and constructive suggestions. We appreciate your acknowledgement that the paper is well-written and clear with interesting investigation directions and findings. We also share with Reviewer the belief that our findings could lead to interesting future research. Please find our answers to your comments and questions as follows.
>
> **`Q. Effect of replacing the input/output layers?`**
>
> **==Updated results on the effect of input layer and output layer replacements (last 2 columns)==**
> | Dataset (ID) |     LIFT/GPT-J    |FPT/GPT-J|FPT(Output Only)/GPT-J|FPT(Input Only)/GPT-J
> |:------------------:|:------------:|:------------:|:------------:|:------------:|
> |Blobs (2)|96.17±0.59|96.75±0.00|96.67±0.12|96.75±0.00
> |Two Circles (6)|75.92±1.65|74.33±0.31|76.33±2.49|69.83±1.31
> |Iris (61)|96.67±0.00|96.67±0.00|97.78±1.57|81.11±3.14
> |Customers (1511)|85.23±1.61|87.88±0.54|88.26±0.54|86.74±1.42
> |Wine (187)|93.52±1.31|100.00±0.00|99.07±1.31|92.59±3.46
> |LED (40496)|65.33±0.47|73.00±2.94|71.67±1.25|68.67±1.89
>
> Thanks for your suggestions. We ran additional experiments by changing input/output layers of GPT-J, following the methods used in Frozen Pretrained Transformers (FPT)[https://arxiv.org/pdf/2103.05247.pdf]. Specifically, we reinitialized a trainable input layer and a trainable output layer, with frozen pretrained GPT-J transformer architectures in the middle. Same as in the FPT paper, the input dimension equals to number of features and the output dimension equals to number of classes, which varies depending on the tasks. We reported the results in the table above and compared them with our own method LIFT/GPT-J. As expected, FPT performs slightly better than LIFT on almost all tested cases.
>
> As per the reviewer's suggestion, we are also working on investigating the effect of having trainable output layer only (or input layer only). Once our experiments are completed, we will report additional results here.
>
> **Updated**: We added experiments to test the effect of input layer and output layer separately. Both FPT and FPT with only output layer perform slightly better than LIFT on almost all tested cases. FPT with only input layer performs the worst. Our justification for these observations is  that training an output layer is similar to training a linear classifier, which might be easier than training an input layer as an encoder.
>
> **`Q. Open question: Reasons for good performance of LIFT.`**
>
> As per your great question, we ran extensive experiments to better explain the reasons for the good performance of LIFT. Please find our response to this question in our response to Q2 in the response to all reviewers & AC. In short, it is important to use large language models that are pretrained on human language data.
>
>
> **`Q. Are there baselines with more comparable numbers of parameters? What is the effect of model capacity?`**
>
> As per your suggestion, we added a new comparison with baseline models with larger capacities. (Though they are still much smaller compared to GPT models!) In particular, we provide the results of deep neural network models based on architectures specifically designed for tabular data: TabNet and TabTransformer.
> * TabNet (Arik and Pfister, 2021) is one of the SoTA deep network architectures specifically designed for tabular data.
> * TabTransformer (Huang et al., 2020) is a deep network architecture specifically designed for tabular data with the use of Transformer architectures. To run TabTransformer, one has to specify the categorical features (with the number of unique values) and continuous features. For the simple evaluation, we consider all features as continuous features and use the TabTransformer directly.
>
> As shown in the table below, these deep-learning-based baselines perform similarly to XGBoost and Random Forest.
>
> | Dataset (DID)    | ODC   | LIFT/GPT-3      | LIFT/GPT-J      | TabNet      | TabTransformer |
> |------------------|-------|------------|------------|-------------|----------------|
> | Blobs (2)        | 25.00 | 96.67±0.24 | 96.17±0.59 |  96.75±0.00  | 50.00±0.00     |
> | Two Circles (6)  | 50.00 | 81.42±0.82 | 75.92±1.65 | 74.25±12.39 | 49.25±1.29     |
> | Iris (61)        | 33.33 | 97.0±0.00  | 96.67±0.00 | 97.78±1.92  | 72.22±5.09     |
> | Customers (1511) | 68.18 | 84.85±1.42 | 85.23±1.61 | 85.22±3.93  | 87.12±0.66     |
> | Wine (187)       | 38.89 | 92.59±1.31 | 93.52±1.31 |  94.44±5.56  | 90.74±13.70    |
> | LED (40496)      | 11.0  | 69.33±2.05 | 65.33±0.47 | 11.68±4.44  | 23.46±13.85 | 67.00±2.46  | 41.00±12.49    |
>
> In addition to these experiments, we are planning to compare the performance of LIFT with a larger range of model sizes. This can be done by comparing the performance of LIFT with various open-sourced language models. In specific, Facebook's OPT model size varies from 125M to 175B. We could not complete the experiments during the rebuttal period, but we will keep you posted once we see any new results.

---

> > ### Author Response · Authors · 2022-08-02
> > **To Reviewer evpg: Continued**
> >
> > *References:*
> > * Arik, S. Ö., & Pfister, T. (2021, May). Tabnet: Attentive interpretable tabular learning. In Proceedings of the AAAI Conference on Artificial Intelligence (Vol. 35, No. 8, pp. 6679-6687).
> > * Huang, X., Khetan, A., Cvitkovic, M., & Karnin, Z. (2020). Tabtransformer: Tabular data modeling using contextual embeddings. arXiv preprint arXiv:2012.06678.
> > * OPT: https://github.com/facebookresearch/metaseq/tree/main/projects/OPT
> >
> > -----
> > **Final notes**: We are excited that you find our work interesting and potentially impactful. We will integrate these answers into our new version. Thanks again for your careful reading of our work and for providing encouraging feedback!

---

### Author Response · Authors · 2022-08-02
**To AC and All Reviewers**

## To AC and All Reviewers

We thank the Reviewers for their insightful feedback and constructive comments and for providing suggestions that would improve our paper.

First of all, we are encouraged that the reviewers found that:
(i) the direction of investigation is interesting (R-evpg, R-GQen, R-iBBJ, R-igVi)), novel (R-GQen, R-iBBJ), and potentially impactful (R-iBBJ);
(ii) the amount of evaluated downstream tasks and results is impressive (R-evpg, R-iBBJ, R-igVi), with interesting and non-intuitive findings on different aspects of the set-up (R-evpg, R-igVi), which could help practitioners gain intuitions (R-iBBJ) and lead to some interesting future research (R-evpg);
(iii) our paper is generally well written and clear (R-evpg, R-iBBJ) with clear tables, figures (R-GQen, R-iBBJ), and details of the appendix, scope of the study, and societal impact (R-iBBJ).

As for the concerns/questions raised, we believe that we successfully addressed every single one, as replied to each reviewer. **We integrated most of the answers and new results in the newly updated version (attached).**

In particular, we found that there are two major questions raised by the reviewers. To answer them, we ran extensive additional experiments, and summarized below are the two most important questions and our responses to them.

**Q1. (paraphrased) Why should one use LIFT? Isn't it too big/slow/expensive compared to small models? It supports "No-code-ML", but it alone does not seem sufficient to justify its employment in practice.**

While we mostly emphasized the "no-code-ML" property of LIFT, indeed it has a lot of potentials to be more useful and powerful than many of the current ML models. Particularly, we believe that LIFT can bring a completely novel approach to enable (i) explainability and (ii) updatability via information retrieval. We will incorporate these additional benefits in the final revision.


(i) **Explainability**. Most ML models cannot explain their own predictions. While there are specific algorithms developed to enable the explainability of such models, their efficacy is still in question.

On the other hand, LIFT, which is based on a large pretrained language model, can be made to explain its prediction using its reasoning capabilities (Wei et al., 2022), (Kojima et al., 2022). While we have not explored this aspect of LIFT in the current paper, we provide a small toy example to demonstrate the explainability of LIFT below.

Consider the German-credit dataset, whose goal is to predict whether the bank should approve or decline loan applications. After asking GPT3 if one should approve/decline a loan application via LIFT, one can also ask GPT3 to explain its own prediction result. This can be implemented by making two consecutive inference calls as shown in the following table. Note that we provide five different responses generated with different random seeds.

---

> ### Author Response · Authors · 2022-08-02
> **Q1 Motivation: Continued (Part I)**
>
> |  | Prompt or Generated Text |
> |:-----------------:|:-----------:|
> |Raw input|Checking Account: less than 200 DM; Loan Duration: 48 months; Credit History: no credits/paid back duly; Loan Purpose: education; Number of applying for credits: 18424; Savings Account: less than 100 DM; Employment Time: 1~4 years; Installment Rate: 1%; Sex: Female; Other Debtors: No; Property: building society savings agreements/life insurance; Age: 32; Housing: self-owned house; Job: highly qualified employee/ self-employed; Foreigner: no|
> |First prompt (soliciting prediction)| A 32-year-old female is applying for a loan of 18424 credits for 48 months for other purposes. She has a checking account with less than 200 Deutsche Mark and a savings account with less than 100 Deutsche Mark. She had no credits or paid back all credits duly. The installment rate of this applicant in percentage is 1%. She is employed for more than one year but less than four years and is recognized as a highly qualified employee or is self-employed. She has lived in a self-owned house. She owns some building society savings agreements or life insurance. Given such information, will the bank accept or decline this loan application? |
> |First response|accept|
> |Second prompt (soliciting explanation)|A 32-year-old female is applying for a loan of 18424 credits for 48 months for other purposes. She has a checking account with less than 200 Deutsche Mark and a savings account with less than 100 Deutsche Mark. She had no credits or paid back all credits duly. The installment rate of this applicant in percentage is 1%. She is employed for more than one year but less than four years and is recognized as a highly qualified employee or is self-employed. She has lived in a self-owned house. She owns some building society savings agreements or life insurance. Given such information, will the bank accept or decline this loan application? **\n The bank will accept this loan application. Let’s think logically. This is because**|
> |Second response (random seed 1)|the applicant is employed and is qualified in a high level. Since she is employed, she is able to make all the payments. Furthermore, she has a self-owned house and owns some building society savings agreements or life insurance. She has less than 2000 Deutsche Mark in checking and savings account. She had no credits or paid back all credits duly. She is able to pay the loans. All of these are positive factors to this loan. The bank will accept this loan.
> |Second response (random seed 2)| the applicant has a high income. She has lived in a self-owned house. She has some savings agreements or life insurance. She is a highly qualified employee. She has no negative credits. All of these facts show that she is not in danger of not being able to pay back the loan. Therefore, the bank will accept this loan application.|
> |Second response (random seed 3)|the applicant has a checking account with less than 200 Deutsche Mark and savings account with less than 100 Deutsche Mark. Hence she is a low-risk customer. Furthermore, she has no credits or paid back all credits duly. And the installment rate of this applicant is 1%. She is employed for less than four years and is recognized as a highly qualified employee. She has lived in a self-owned house. She owns all the building society savings agreements or life insurance. Hence she is considered a good customer. So the bank will accept her loan application |
> |Second response (random seed 4)| the applicant is employed with a high salary and is economically stable. She is not likely to file for bankruptcy. She can pay back the loan. All in all, we can see that the bank will accept this loan application.|
> |Second response (random seed 5)| the bank will accept all applications with a very low risk of default. There is a very low probability that she will be unemployed in the near future. She can afford to pay off this loan. Furthermore, this applicant has a high annual income.|
>
> Note that these explanations are generated without seeing *any* reasoning examples, i.e., these are solely based on the zero-shot reasoning capability of language models. While these explanations are not even close to perfect, they are still very reasonable. Improving the few-/zero-shot reasoning capabilities of language models is a fast-growing research field, so incorporating new techniques will further improve the explainability of LIFT. Given its inherent compatibility with the zero-shot reasoning capability of language models, we strongly believe that it is worth sharing our LIFT framework with the research community.

---

> > ### Author Response · Authors · 2022-08-02
> > **Q1 Motivation: Continued (Part II)**
> >
> > (ii) **Updatability via information retrieval**. Another drawback of current ML models is that it is difficult to "update" models when distribution shift occurs. Handling this distribution shifts in a compute-efficient manner has recently become one of the most active research areas in the field. Recently, researchers have come up with a very efficient way of updating language models by augmenting them with a retrieval mechanism. With such a retrieval mechanism equipped, language models can be efficiently updated as one can simply update its associated database. One can also connect language models with the Internet. See Retrieval-Enhanced Transformer (Retro) (Borgeaud et al., 2021), SPALM (Yogatama et al., 2021), FiD (Izacard et al., 2021), or $EMDR^2$ (Singh et al., 2021) for recent advances.
> >
> > While we only used language models that are not equipped with a retrieval mechanism in this paper, it is very straightforward to apply our framework to other language models that can retreive information from database or the Internet, i.e., LIFT can support a compute-efficient update mechanism.
> >
> > *References*:
> > * Wei, J., Wang, X., Schuurmans, D., Bosma, M., Chi, E., Le, Q., & Zhou, D. (2022). Chain of thought prompting elicits reasoning in large language models. arXiv preprint arXiv:2201.11903.
> > * Kojima, T., Gu, S. S., Reid, M., Matsuo, Y., & Iwasawa, Y. (2022). Large Language Models are Zero-Shot Reasoners. arXiv preprint arXiv:2205.11916.
> > * Borgeaud, S., Mensch, A., Hoffmann, J., Cai, T., Rutherford, E., Millican, K., ... & Sifre, L. (2022, June). Improving language models by retrieving from trillions of tokens. In International Conference on Machine Learning (pp. 2206-2240). PMLR.
> > * Yogatama, D., de Masson d’Autume, C., & Kong, L. (2021). Adaptive semiparametric language models. Transactions of the Association for Computational Linguistics, 9, 362-373.
> > * Izacard, G., & Grave, E. (2020). Leveraging passage retrieval with generative models for open domain question answering. arXiv preprint arXiv:2007.01282.
> > * Singh, D., Reddy, S., Hamilton, W., Dyer, C., & Yogatama, D. (2021). End-to-end training of multi-document reader and retriever for open-domain question answering. Advances in Neural Information Processing Systems, 34, 25968-25981.

---

> > > ### Author Response · Authors · 2022-08-02
> > > **Q2: Does LIFT really need a large "pretrained" model on "natural language" data?**
> > >
> > > **Q2. (paraphrased) Does LIFT really need a large "pretrained" model on "natural language" data?**
> > >
> > > For better answering, we decompose this question into two parts.
> > >
> > > In Q.2.1, we first answer "Does LIFT really need a large **pretrained** model"? To answer this, we compare the performance of LIFT applied to pretrained GPT-J and that applied to randomly initialized GPT-J. We observed that LIFT applied to randomly intialized GPT-J does not work.
> > >
> > > In Q.2.2, we then answer "Does LIFT really need a large pretrained model on **natural language** data"? To answer this, we compare the performance of LIFT applied to natural-language-pretrained GPT-J and that applied to program-code-pretrained GPT. (Note that ideally, we wanted to compare it with GPT pretrained on gibberish, but we could not perform a large-scale pretraining on our own given a limited time and compute resource.) As a result, we found that there is a significant gap between the two cases, showing the necessity of our choice of **natural language models**.
> > >
> > >
> > >
> > > **Q.2.1 Does LIFT really need a large **pretrained** model?**
> > >
> > > The following table presents our results. Here, we obtain baselines by applying LIFT onto different pretrained LMs. ODC is the Optimal Deterministic Classifier which always outputs the major class as the classification. We use ODC as a simple baseline for comparison.
> > >
> > > | Dataset (DID)    | LIFT/GPT-J    | LIFT/Rand-GPTJ       |   ODC   |
> > > |------------------|-------|-------------|------------|
> > > | Blobs (2)        | 96.17±0.59  | 25.65±1.58  |  25.00 |
> > > | Two Circles (6)  | 75.92±1.65 | 49.88±5.01  | 50.00  |
> > > | Iris (61)        | 96.67±0.00  | 27.78±20.79 | 33.33 |
> > > | Customers (1511) | 85.23±1.61 | 52.47±7.15  | 68.18  |
> > > | Wine (187)       | 93.52±1.31 | 22.22±15.71 | 38.89 |
> > > | LED (40496)      | 65.33±0.47  | 11.68±4.44  | 11.0 |
> > >
> > >
> > > We compare our models with LIFT/Rand-GPT-J where Rand-GPT-J is the GPT-J model with weights being randomly initialized. More specifically, we randomly initialized a GPT-J model and fine-tuned the whole model (instead of LoRA). We tried various learning rates (0.1-0.5) and report the performance of the model that achieved the highest average validation accuracy. We also tuned the temperature as we observed the zero temperature consistently gave us poor performances. As a result, we used a temperature of 1. Even after we set the temperature to 1, only 10%~15% of outputs are valid. The accuracies listed in the table are calculated only among valid outputs.
> > >
> > > As one can see from the results, LIFT/Rand-GPT-J achieves much lower performance than LIFT/GPT-J. Thus, we believe that good performance with LIFT comes from pretraining, not just from the large model size.
> > >
> > > **Q.2.2 Does LIFT really need a large pretrained model on *natural language* data?**
> > >
> > > | Dataset (DID)    |  LIFT/GPT-J (pretrained on natural language)      |  LIFT/CodeParrot (pretrained on program code)  |  LIFT/CodeGen (pretrained on program code)    | ODC   |
> > > |------------------|-------|------------|------------|------------|
> > > | Blobs (2)        | 96.17±0.59 |  93.39±1.82  | 93.67±0.72 | 25.00 |
> > > | Two Circles (6)  | 75.92±1.65 |  50.08±2.47  | 53.02±0.66 | 50.00 |
> > > | Iris (61)        | 96.67±0.00 |  60.00±8.82  | 43.31±6.67 | 33.33 |
> > > | Customers (1511) | 85.23±1.61 |  43.11±3.34  | 45.96±8.96 | 68.18 |
> > > | Wine (187)       | 93.52±1.31 |  33.88±3.87  | 77.78±0.00 | 38.89 |
> > > | LED (40496)      | 65.33±0.47 |  23.46±13.85 | 11.00±4.00 | 11.0  |
> > >
> > > To answer the second question, we apply LIFT to models that are pretrained on non-human language data. CodeParrot (https://huggingface.co/codeparrot) and CodeGen (Nijkamp et al., 2022) are the language models pretrained on programming source code.
> > >
> > > As one can see from the table, LIFT applied with CodeGen or CodeParrot does not perform well in most cases. These results imply that LIFT works the best with language models that are trained on natural language, not just any data.
> > >
> > > *References*:
> > > * Nijkamp, E., Pang, B., Hayashi, H., Tu, L., Wang, H., Zhou, Y., ... & Xiong, C. (2022). A conversational paradigm for program synthesis. arXiv preprint arXiv:2203.13474.

---

### Meta-Review · Area_Chair_G26u · 2022-08-25

**Recommendation:** Accept
**Confidence:** Certain

**Metareview:**

The paper demonstrates that pre-trained language models can be competitive at classifying non-textual data, where the input features are linearized into a text-like sequence and used as the conditional prefix for the language model. While the method is still not competitive with supervised learning methods, the fact that LLMs are able to do the task is intriguing. During the rebuttal, the authors also provided convincing answers to when one might prefer this approach (which is computationally expensive) over traditional methods. This paper provides timely insights into empirical research on LLMs. Therefore, I recommend acceptance.

**Award:**

No

---

### Decision · Program_Chairs · 2022-09-14

Accept